# Interpretable Functional Koopman Learning
# with Non-Markovian Closure for Spatiotemporal Systems

**Wanfeng Lu** [* 1 2]   **He Ma** [* 1 2 3]   **Wei Lin** [1 2 3 4]   **Qunxi Zhu** [2]

## Abstract

Precise prediction of spatiotemporal dynamics over predictive horizons is constrained by the computational cost of high-fidelity solvers and the sparsity, noise, and irregularity of data. We introduce MERLIN, a Koopman-based framework that lifts dynamics to the evolution of learned *observation functionals* with near-linear progression, enabling full-field reconstruction at arbitrary resolutions. Theoretically, we develop a functional Koopman theory for PDEs and compensate for the loss of finite-dimensional linear invariance via the Mori–Zwanzig formalism, which augments the linear backbone with non-Markovian memory terms to improve predictive accuracy. Practically, MERLIN employs discretization-invariant *function encoders* that map partial, irregular observations to observables, and resolution-free *function decoders* that reconstruct states at arbitrary query points. Training under linear constraints yields an interpretable, low-dimensional model that captures principal modes and supports reduced-order modeling, while memory correction further enables stable long-horizon rollouts even in ultra-low-dimensional latent spaces. Our code is available at: https://github.com/RobinLufdu/MERLIN.

## 1. Introduction

Partial differential equations (PDEs) underpin spatiotemporal modeling in climate, turbulence, and biomechanics (Vallis, 2017; Pope, 2000; Fung, 2013). Yet long-horizon prediction remains difficult: high-resolution solvers are costly with respect to integration time and grid refinement (LeVeque, 2002; Brenner & Scott, 2008), and real data are often irregularly sampled, which hinders purely data-driven methods (Brunton & Kutz, 2022). To narrow this computation-data gap, two main paradigms have emerged: neural operators (e.g., DeepONet (Lu et al., 2021), FNO (Li et al., 2021)) and PINNs (Raissi et al., 2019). While neural operators approximate infinite-dimensional maps with finite parameters, they can raise concerns about truncation, stability, and interpretability, and often assume structured discretizations; PINNs, by contrast, require known governing equations. An equation-free, latent, function-level approach instead naturally accommodates inputs of arbitrary format: a discretization-invariant encoder maps any discrete sampling of a field to a fixed-dimensional latent, and a resolution-free decoder reconstructs the field at arbitrary query locations.

These considerations motivate an interpretable, dynamics-grounded paradigm for learning long-term evolutionary characteristics from random, partial observations (see Table 1 for the comparison between different models). The Koopman perspective provides the organizing principle: lift states to an observable space that evolves linearly (Brunton et al., 2022; Nakao & Mezić, 2020; Mauroy, 2021; Xu et al., 2025). However, despite successes in finite-dimensional systems (Takeishi et al., 2017; Lusch et al., 2018), a unified, practical framework for *infinite-dimensional* settings with random, partial, irregular observations remains elusive. To bridge this gap, we introduce a *functional Koopman learning* framework: measurements are treated as *discrete representations* of the underlying function states; a discretization-invariant *function encoder* learns *Koopman observation functionals* whose evolution is approximately linear; and a resolution-free *function decoder* reconstructs states at arbitrary points and resolutions. However, when the learned observable subspace fails to be invariant under the action of Koopman operator, the dynamics on its orthogonal complement, which corresponds to unresolved observables, feed back into the linear evolution of the resolved observables via an additional memory correction term, as prescribed by the classical Mori-Zwanzig formalism (Mori, 1965; Zwanzig, 1973). To account for both the functional Koopman learning and its corresponding Mori-Zwanzig mo-

---

[*]Equal contribution [1]School of Mathematical Sciences, Fudan University, China [2]Research Institute of Intelligent Complex Systems, Fudan University, China [3]Shanghai Artificial Intelligence Laboratory, China [4]State Key Laboratory of Medical Neurobiology and MOE Frontiers Center for Brain Science, Institutes of Brain Science, Fudan University, China. Correspondence to: Wei Lin <wlin@fudan.edu.cn>, Qunxi Zhu <qxzhu@fudan.edu.cn>.

*Proceedings of the 43rd International Conference on Machine Learning*, Seoul, South Korea. PMLR 306, 2026. Copyright 2026 by the author(s).

tivated memory correction, we present **MERLIN**, short for **M**emory-augmented **K**oopman **E**volution with a **R**esolution-free autoencoder for random partial observations in PDE **L**earn**IN**g, together with the ensuing technical contributions.

**Functional Koopman-Mori-Zwanzig theory for PDEs.** We generalize the classical Koopmanism from finite-dimensional systems to *infinite-dimensional* by extending observation functions to *observation functionals*, and demonstrate that the Koopman operator associated with PDEs admits a finite-dimensional matrix approximation whenever a finite-dimensional invariant subspace exists; when this linear invariance fails, we *theoretically* prove that a memory correction term should be incorporated, yielding an exact description of those observables via *non-Markovian* dynamics.

**Learning observation functionals from random partial observations.** We parameterize observation functionals which map *function states* to a low-dimensional latent manifold using a discretization-invariant *function encoder*. This design naturally (i) handles irregular/partial sampling, and (ii) remains discretization-invariant.

**Resolution-free function decoder.** Conditioned on the latents, we design a *function decoder* that reconstructs fields at arbitrary resolutions. A FourierNet-based realization supports any number of query points and irregular grids, enabling data completion and super-resolution.

**Interpretable Koopman learning with non-Markovian correction.** Building on our theory and functional autoencoder, we adopt a two-phase training pipeline: Phase I learns observation functionals that approximately linearize Koopman evolution; Phase II augments this linear backbone with learned memory terms to capture non-Markovian effects. The overall model is *interpretable*: the linear backbone approximates Koopman eigenfunctionals and yields an ultra-low-dimensional ROM (further reduced via projection heads), while memory terms improve long-horizon accuracy. In practice, as few as **8** observables capture most dynamical variance for the Navier–Stokes equation, and the memory-augmented model remains competitive on both synthetic and real datasets, *suggesting a parsimonious finite-dimensional closure for effectively infinite-dimensional dynamics within our Koopman-MZ framework.*

## 2. Preliminaries

### 2.1. Nonlinear Evolution Equations

**Nonlinear evolution equations.** Let $T > 0$, $\Omega \subset \mathbb{R}^d$ be a spatial domain with Lipschitz boundary $\partial\Omega$. Throughout this work, we consider the following time-dependent PDEs over $[0, T] \times \Omega$:

$$\partial_t u(t, x) = \mathcal{L}[u](t, x), \qquad (1)$$

with initial condition $u(0, x) = u_0(x)$ ($x \in \Omega$), and boundary condition $\mathcal{B}[u|_{\partial\Omega}](t) = 0$ ($t \in [0, T]$). Here, $u(t, \cdot)$ denotes the *state* (or *field variable*) of the system at time $t$, viewed as an element of a suitable function space $\mathcal{H}$ equipped with norm $\| \cdot \|_{\mathcal{H}}$ (e.g. $\mathcal{H} = L^2(\Omega)$ or $H_0^1(\Omega)$, depending on the regularity of the underlying system). The spatial operator $\mathcal{L}$ is possibly nonlinear and nonlocal, and $\mathcal{B}$ encodes boundary conditions (Dirichlet, periodic, etc.).

**Abstract Cauchy formulation and standing assumption.** We reformulate the spatiotemporal dynamics as the evolution of abstract function states $u(t, \cdot) \triangleq u_t \in \mathcal{H}$ and write

$$\dot{u}(t) = \mathcal{F}(t, u(t)), \qquad u(s) = u_s \in \mathcal{H}, \qquad (2)$$

where $\mathcal{F}(t, u) \in \mathcal{H}$ is given by $\mathcal{F}(t, u)(x) = \mathcal{L}[u](t, x)$ for $x \in \Omega$. We fix a realization of the spatial operator $\mathcal{L}$ on $\mathcal{H}$ whose domain $\mathcal{D}(\mathcal{L}(t)) \subset \mathcal{H}$ encodes the boundary conditions. We assume throughout that the abstract Cauchy problem (2) on $\mathcal{H}$ is autonomous and well posed, that is, it admits a unique solution with continuous dependence on the initial state. This well-posedness induces the semigroup $S_t$ defined by $S_t(u_s) = u(t + s)$ for $t, s > 0$. See Appendix A.1 for details.

### 2.2. Problem Setting

We model spatiotemporal dynamics from data. The training set comprises trajectories $\{u^{(i)}\}$ obtained by simulating spatiotemporal dynamics (1) from initial states $u_0^{(i)} \in \mathcal{H}$ and sampling randomly on time-varying, irregular grids $\mathcal{S}_t^{(i)} \subset \Omega$ (finite cardinality). Each trajectory is recorded as $\left\{ u^{(i)}(k\Delta t)|_{\mathcal{S}_{k\Delta t}^{(i)}} \right\}_{k=0}^{K}$ with sampling interval $\Delta t$. At test time, given $N_{\text{test}}$ short sequences with $l$ conditioning frames $\{v(0), \ldots, v((l-1)\Delta t)\}$ observed on random grids $\mathcal{S}_t'$, we roll out to $K'\Delta t$ and query predictions on a discretized grid $\Omega_d \subset \Omega$: $\{v_{\text{pred}}(k\Delta t)|_{\Omega_d}\}_{k=l}^{K'}$. We evaluate on the **test set** using two time windows. The **train-horizon loss** $\ell_{\text{train-t}}$ is the average pointwise error $\ell(\cdot, \cdot)$ on $\Omega_d$ over the part of the rollout that falls within the training horizon ($k = l, \ldots, K$), averaged across test trajectories. The **test-horizon loss** $\ell_{\text{test-t}}$ is the same error averaged over the extrapolation window beyond training ($k = K+1, \ldots, K'$) and across trajectories. See Appendix B for more details.

### 2.3. Overview

The overall framework is depicted in Figure 1. We establish a functional Koopman framework in Section 3.1 and its memory-augmented extension motivated from Mori-Zwanzig theory in Section 3.2. Leveraging these theoretical insights, we propose in Section 4 a two-phase training procedure that learns (i) the linear backbone dynamics and (ii) memory-augmented non-Markovian dynamics, respectively. We then present numerical experiments in Section 5.

*Table 1.* Comparison with other data-driven approaches for spatiotemporal modeling.

| Model | Interpretability | Reduced order modeling | Random partial observations | Irregular-grid evaluation | Stable time extrapolation | Optimization free evaluation |
|---|---|---|---|---|---|---|
| FNO (Li et al., 2021) | ✗ | ✗ | ✗ | ✗ | ✓ | ✓ |
| DeepONet (Lu et al., 2021) | ✗ | ✗ | ✗ | ✗ | ✗ | ✓ |
| KNO (Xiong et al., 2024) | ✓ | ✗ | ✗ | ✗ | ✓ | ✓ |
| PINN (Raissi et al., 2019) | ✓ | ✗ | ✓ | ✓ | ✗ | ✓ |
| DINo (Yin et al., 2023) | ✗ | ✓ | ✓ | ✓ | ✓ | ✗ |
| MERLIN (Ours) | ✓ | ✓ | ✓ | ✓ | ✓ | ✓ |

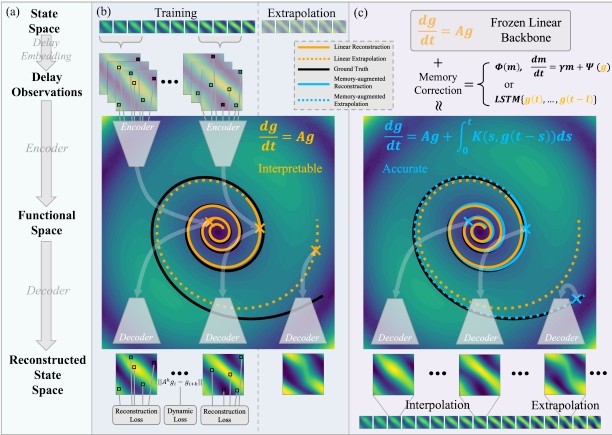

*Figure 1.* **An overview of the framework.** (b) Learning a linear backbone with Koopman observation functionals: (top) delay-embedded **random observations** are fed into an encoder (middle) to produce observables in the **functional space**; (bottom) a subsequent decoder maps latents to the **reconstructed state space**. (c) Memory correction atop the frozen linear backbone: (top) a non-Markovian dynamics model "——" is trained (middle) to adjust linear trajectories "——" toward the encoded latent trajectory "——", and (bottom) the result is rolled out in the state space.

## 3. Koopman Theory for PDEs

The classical Koopman viewpoint (Koopman, 1931; Koopman & v. Neumann, 1932) shifts attention from Poincaré's phase-space geometry (Strogatz, 2024) to the evolution of observables. Remarkably, although the underlying dynamics may be nonlinear, observables propagate *linearly* under an (often infinite-dimensional) operator semigroup, the Koopman operators, which enable data-driven approximation without linearizing the state space dynamics (Schmid, 2010; Williams et al., 2015; Lusch et al., 2018). Most Koopman learning in the machine-learning literature, however, implicitly assumes a finite-dimensional state space and hence observables as functions on $\mathbb{R}^n$. For PDEs the state lives in a function space $\mathcal{H}$, and observables are naturally *functionals* on $\mathcal{H}$ (usually nonlinear); while the formal operator-theoretic picture resembles the ODE case, the PDE setting is intrinsically different (e.g., infinite-dimensional state space, unbounded generators, and more delicate spectral structure). In what follows, we adopt this *functional* Koopman perspective and build on the seminal Koopman

theory for PDEs developed in Nakao & Mezić (2020).

### 3.1. Functional Observables and Koopman Operator for PDEs

Denote by $\mathcal{H}$ the state space, and let $g : \mathcal{H} \to \mathbb{R}$ be a (possibly nonlinear) *functional observable* of the field variable $u \in \mathcal{H}$. Let $\mathcal{O}$ denote the space of functionals on $\mathcal{H}$ (see Appendix A.1 for details). We consider the one-parameter semigroup $(S_t)_{t\geq 0}$ on $\mathcal{H}$. From the Koopman viewpoint, the evolution of the observable $g$[1] is described by the action of the Koopman operator $\mathcal{K}_t$, namely $\mathcal{K}_t g[u] = g[S_t u]$, where we use $g[\cdot]$ to denote the action of a functional on state variables. It is easy to verify that (i) $\mathcal{K}_0$ is an identity; (ii) $\mathcal{K}_t$ is a linear operator; (iii) $\mathcal{K}_t$ is commutable, and (iv) $\mathcal{K}_t f(g) = f(\mathcal{K}_t g)$, where $f$ is any function. Proof can be found in appendix A.2. Furthermore, denoting the observable at $t$, $\mathcal{K}_t g$, by $g(t)$, the infinitesimal evolution of observable $g(t) \in \mathcal{O}$ can be described following

$$\frac{\mathrm{d}}{\mathrm{d}t} (g(t)[u]) = \mathcal{A} (g(t)[u]), \qquad (3)$$

where the linear operator $\mathcal{A}$ is the infinitesimal generator (see appendix A.2 for details) of the Koopman operator $\mathcal{K}_t$ given by $\mathcal{A}g[u] = \lim_{\tau \to 0} \frac{\mathcal{K}_\tau g[u] - g[u]}{\tau}$. We refer to (3) as the *Koopman dynamics*. Using the generator $\mathcal{A}$, the action of the Koopman operator $\mathcal{K}_t$ on the observable $g[\cdot]$ can be expressed as $\mathcal{K}_t g[u] = \mathrm{e}^{\mathcal{A}t} g[u] = \sum_{k=0}^{\infty} \frac{1}{k!} t^k \mathcal{A}^k g[u]$, which also implies $\mathcal{A}$ and $\mathcal{K}_t$ are commutative.

Benefiting from the linearity of the Koopman operator $\mathcal{A}$, the observables evolve in a linear manner, as shown in (3). If we further assume the existence of a finite set of dominant observables $\{g_1, g_2, \cdots, g_D\}$ that spans a linear invariant subspace of $\mathcal{A}$[2], then we can truncate $\mathcal{A}$ via the *finite-rank operator* given by $\mathcal{A}_D \triangleq \mathcal{P}_D \mathcal{A} \mathcal{P}_D$, where $\mathcal{P}_D : \mathcal{O} \to \mathrm{span}\{g_1, \cdots, g_D\}$ is the projection. The resulted finite-rank operator is then characterized by a simple $D \times D$ matrix, as is the case in traditional Koopman learning methodologies. Consequently, we obtain a finite-

---

[1]By the evolution of $g$ we mean composition with the semigroup, $g \mapsto g \circ S_t$.

[2]Invariance means $\mathcal{A}\,\mathrm{span}\{g_1, g_2, \cdots, g_D\} \subset \mathrm{span}\{g_1, g_2, \cdots, g_D\}$.

dimensional linear dynamics for $\mathbf{g} = [g_1, g_2, \cdots, g_D]^\top$, that is $\dot{\mathbf{g}} = A\mathbf{g}$, where $A$ is the matrix representaion of $\mathcal{A}_D$.

## 3.2. The Emergence of Memory Beyond Finite-Dimensional Linear Invariance

Existing Koopman learning frameworks rely heavily on the existence of a non-trivial finite-dimensional invariant subspace $\mathcal{M} = \mathrm{span}\{g_1, \ldots, g_D\}$ (Colbrook et al., 2023). Nonetheless, even when such a subspace exists, identifying a finite set of observables that *exactly* closes the dynamics is notoriously difficult; moreover, for PDE-governed systems the evolution of observables typically unfolds in a richer (often infinite-dimensional) observable space $\mathcal{O}$, so exact linear closure is generally not available.

Fortunately, the classical Mori-Zwanzig (MZ) formalism (Mori, 1965; Zwanzig, 1973; 2001) addresses an analogous issue in non-equilibrium statistical mechanics: starting from a high-dimensional *full-order* microscopic model, it derives an exact reduced description for a chosen set of *resolved* (macroscopic) variables. The effect of *unresolved* variables on the resolved dynamics appears through (i) a time-nonlocal *memory* term and (ii) a *fluctuation/forcing* term determined by the orthogonal dynamics of unresolved variables and uncertain initial conditions. **In this work**, we transplant this operator-theoretic MZ viewpoint into the *functional Koopman* setting for PDEs. We take the Koopman dynamics (3) on the observable space $\mathcal{O}$ as the *full-order* linear evolution, and we declare an **arbitrary** finite family of observables $\{g_i\}_{i=1}^D$ as the *resolved* macroscopic variables. Let $\mathcal{M} = \mathrm{span}\{g_i\}_{i=1}^D \subset \mathcal{O}$ denote the resolved subspace; any components of the Koopman evolution not representable in $\mathcal{M}$ are treated as *unresolved*. The observables $g_i$ may be linear or nonlinear, hand-crafted or learned by a neural network (and need not be linearly independent). Our goal is to characterize the closed evolution of the resolved observable vector $\mathbf{g}_\mathcal{M}$ under the Koopman dynamics. This leads to the following generalized Langevin equation.

**Theorem** (Generalized Langevin equation). *Let* $\mathbf{g}_\mathcal{M}(t) = [g_1(t), \ldots, g_D(t)]^\top$ *denote the time evolution of a finite family of observables under the Koopman dynamics (3). Then* $\mathbf{g}_\mathcal{M}$ *admits a generalized Langevin representation of the form*

$$\frac{\mathrm{d}}{\mathrm{d}t}\mathbf{g}_\mathcal{M}(t) = \mathbf{M}\,\mathbf{g}_\mathcal{M}(t) + \int_0^t \mathbf{K}(s, \mathbf{g}_\mathcal{M}(t-s))\,\mathrm{d}s + \mathbf{F}(t),$$
(4)

*where* $\mathbf{M}$ *is a Markovian transition matrix acting on the resolved observables,* $\mathbf{K}$ *is a memory kernel capturing the influence of unresolved observables through time-nonlocal interactions, and* $\mathbf{F}(t)$ *is a fluctuation term induced by the orthogonal dynamics on the unresolved subspace.*

The above theorem yields an exact, closed evolution for the observables $\mathbf{g}_\mathcal{M}$ in the form of a generalized Langevin equation (GLE). Further discussions and its proof are given in Appendix A.2, and a detailed analysis of the scope and limitations of the theory is provided in Appendix A.3. To give intuition for the emergence of memory, we also examine a simple linear PDE example in Appendix A.3.

## 3.3. Learning Koopman Observation Functionals from Data

Based on our theory, we learn observables directly from data and augment their evolution with memory corrections. To minimize the contribution of possible memory effects, we seek observables that *approximately* linearize the underlying dynamics. Inspired by Koopman dictionary learning for ODEs (Takeishi et al., 2017; Lusch et al., 2018), we parameterize observation functionals $\mathbf{g} : \mathcal{H} \to \mathbb{R}^D$ with neural networks, treating $\mathbf{g}$ as a *function encoder* whose output is the latent. Since observables act on functions, we regard $u(k\Delta t)|_\mathcal{S}$ as discrete representations of $u(k\Delta t, \cdot)$ and enforce *discretization invariance*: different grids/resolutions/sensor layouts of the same field map to (approximately) the same latent. A resolution-free *function decoder* then reconstructs the field at arbitrary query locations.

In modern dynamical theory, delay observables plays a crucial rule in phase space reconstruction (Takens, 2006; Sauer et al., 1991), and underpin Hankel-DMD based spectral analysis (Arbabi & Mezic, 2017; Brunton et al., 2017), among other applications. Accordingly, we inherit these favor by first lifting $\mathcal{H}$ to $\mathcal{H}^{\otimes l}$ (the Cartesian product $\underbrace{\mathcal{H} \times \cdots \times \mathcal{H}}_{l \text{ times}}$) via delay embedding, i.e. $u(k\Delta t) \mapsto \big(u(k\Delta t), \cdots, u((k-l+1)\Delta t)\big)$. On the enlarged space, we choose integral operators $\boldsymbol{v}(\cdot) \mapsto \int \boldsymbol{\kappa}(\cdot, \xi, \boldsymbol{v}(\xi))\boldsymbol{v}(\xi)\,\mathrm{d}\xi$ to perform nonlinear transformations between function spaces of the form $\mathcal{H}^{\otimes d}$, which is analogous to how MLP layers transform vectors between Euclidean spaces $\mathbb{R}^d$. To further extract a finite–dimensional observable subspace, we append a similar integral projection $\boldsymbol{v}(\cdot) \mapsto \int \widetilde{\boldsymbol{\kappa}}(\xi, \boldsymbol{v}(\xi))\boldsymbol{v}(\xi)\,\mathrm{d}\xi \in \mathbb{R}^D$, akin to pooling layers that aggregate hidden information in CNNs. Overall, this yields the pipeline $\mathcal{H} \to \mathcal{H}^{\otimes l} \to \mathcal{H}^{\otimes d} \to \mathbb{R}^D$, whose composition defines the observation functionals $\mathbf{g} : \mathcal{H} \to \mathbb{R}^D$. Details of the parameterization of $\mathbf{g}$ are provided in the next section (also see Appendix D).

## 4. Method

A two-phase learning algorithm is adopted. In Phase I, the function encoder-decoder is optimized under reconstruction, latent linearity, and one-step prediction losses, so that the encoded latent define reconstructive observation functionals whose evolution is well approximated by a linear Koopman propagator. In Phase II, the encoder and the linear propagator are kept fixed, and the memory module is optimized

to model the residual dynamics not captured by the linear backbone. The two phases therefore separate the learning of functional Koopman observables from the subsequent Mori-Zwanzig motivated compensation for the loss of finite-dimensional linear invariance.

### 4.1. Phase-I: Approximated Koopman Learning for Linear Backbone

The learning goal of phase I is to find observation functionals that (i) linearize the dynamics; (ii) can be decoded back to $\mathcal{H}$. To satisfy the *discretization invariant* property which is inherent to our formulation, we instantiate our encoder on Galerkin Transformer (Cao, 2021), and use FourierNet (Yin et al., 2023; Fathony et al., 2021) as our decoder, which maps latent embeddings to reconstructed state functions $u(\cdot) \in \mathcal{H}$. Details of the encoder-decoder architecture are discussed in Section 4.3. As mentioned in Section 3.3, we use delay-embedded state as input to our function encoder $\mathcal{E}_\phi$, and feed the latent embeddings $z$ encoded by $\mathcal{E}_\phi$ together with the query locations $\boldsymbol{x}$ to the function decoder $\mathcal{D}_\theta$, obtaining reconstructed $\widehat{u}(\boldsymbol{x}) = \mathcal{D}_\theta(z)(\boldsymbol{x})$. Specifically, we stack $\{u((k-l+1)\Delta t, \cdot), \cdots, u(k\Delta t, \cdot)\}$ to form delay-embedded feature $\boldsymbol{u}_{\text{delay}}(k\Delta t)$, and we further encode $\boldsymbol{u}_{\text{delay}}(k\Delta t)$, obtaining $z_k = \mathcal{E}_\phi[\boldsymbol{u}_{\text{delay}}(k\Delta t)] \in \mathbb{R}^D$. Instead of parameterizing the Koopman propagator as a trainable matrix, we fit the linear map $(A^*, b^*)$ from latent pairs $(z_k, z_{k+1})$ via closed-form ridge regression and optimize

$$\begin{aligned} \mathcal{L}_\text{I}(\boldsymbol{\phi}, \boldsymbol{\theta}) = {} & \lambda_{\text{recon}}^{(1)} \|\mathcal{D}_\theta(z_k) - u(k\Delta t)\|_2^2 \\ & + \lambda_{\text{linear}}^{(1)} \|z_{k+1} - A^* z_k - b^*\|_2^2 \\ & + \lambda_{\text{pred}}^{(1)} \|\mathcal{D}_\theta(A^* z_k + b^*) - u((k+1)\Delta t)\|_2^2. \end{aligned}$$

The three terms respectively enforce field reconstruction, near-linear latent Koopman evolution, and one-step decoded prediction. After Phase I training, we save the learned encoder-decoder together with the latent linear propagator, which is maintained via an exponential moving average (EMA) of ridge-regression sufficient statistics and then fixed as the linear backbone in Phase II. Further details are provided in Appendix C.1.

### 4.2. Phase-II: Augmenting Linear Dynamics with Memory Correction

Grounded on our theoretical findings established in the previous section, we propose a non-Markovian correction term on the basis of linear backbone in Phase I. Roughly speaking, the temporal dynamics of the observables $z$ can be written as $z_{t+1} = A z_t + e_t$, where $A z_t$ represents the linear backbone and $e_t$ is the memory-correction term. Here, we adopt a discrete formulation of the Mori-Zwanzig formalism, since common spatiotemporal datasets are sampled at fixed regular intervals.

The following two modeling strategies of the memory correction term are adopted:

$$\begin{aligned} & (\textit{leaky memory}) \ e_t = r \odot \Phi_{\text{dec}}(m_t), \\ & \qquad \text{where } m_t = \gamma \odot m_{t-1} + \Phi_{\text{enc}}(z_t); \quad (5) \\ & (\textit{finite memory}) \ e_t = \text{LSTM}(z_t, \cdots, z_{t-1-l_{\text{mem}}}). \end{aligned}$$

Here, for the *leaky memory model* (LMM), $r$ functions as gate controlling the strength of memory correction, $m_t \in \mathbb{R}^{d_{\text{mem}}}$ denotes the hidden memory state, while $\gamma \in (0,1)^{d_{\text{mem}}}$ stands for the memory decay; as for the *finite memory model* (FMM), $l_{\text{mem}}$ is a pre-defined hyperparameter meaning the length of the memory. During Phase II, the encoder $\mathcal{E}_{\phi*}$ and the linear backbone are fixed, with the memory module trained with

$$\mathcal{L}_\text{II} = \lambda_{\text{dyn}}^{(2)} \|z_t - \alpha_t^*\|_2^2 + \lambda_{\text{pred}}^{(2)} \|\mathcal{D}_\theta(z_t) - u(t, \cdot)\|_2^2 + \lambda_{\text{corr}} \|e_t\|_2^2.$$

Here the first term aligns the rollout latent $\{z_t\}$ with the frozen encoder's trajectory $\{\alpha_t^*\}$, the second maintains predictive accuracy after decoding, and the last term regularizes the memory pathway so that the linear backbone remains dominant. Further details are included in Appendix C.2, the theoretical motivation and comparison of the two memory parameterizations are given in Appendix E, and empirical sensitivity to the memory parameterization is reported in Appendix I.2 Table 13.

### 4.3. Discretization-Invariant Encoder and Resolution-Free Decoder

**Encoder $\mathcal{E}_\phi$.** We adopt Galerkin Transformer (Cao, 2021) as the encoder backbone. Given delay-embedded samples $\{\boldsymbol{u}_{\text{delay}}(t, x_i)\}_{i=1}^N$ discretized over $\{x_i\}_{i=1}^N \subset \Omega$, a linear embedding produces $\boldsymbol{Y}_0 \in \mathbb{R}^{N \times d_1}$. A stack of $L$ *self-attention* layers with linear (Galerkin) attention $\text{Atten}(\boldsymbol{Y}) = \frac{1}{N} \boldsymbol{Q} \boldsymbol{K}^\top \boldsymbol{V}$ maps $\boldsymbol{Y}_0$ to $\boldsymbol{Y}_L \in \mathbb{R}^{N \times d_1}$. This

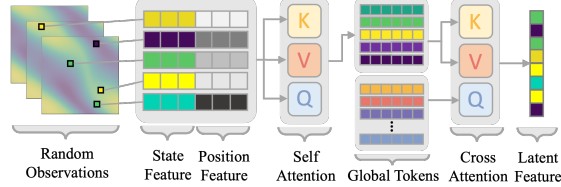

*Figure 2.* Attention-based function encoder.

attention mechanism can be interpreted as a quadrature approximation to a learnable integral operator on functions. Moreover, unlike Galerkin Transformer as in Cao (2021); Li et al. (2023b), which maintains location-wise features in $\mathbb{R}^{N \times d_1}$ throughout, we aggregate to a discretization invariant global embedding. Specifically, we initialize $K$ learnable [CLS]-style tokens $\boldsymbol{z}^{(0)} \in \mathbb{R}^{K \times d_1}$ and apply $L'$ layers of *cross-linear-attention* to *pool* the location-scaling representation $\boldsymbol{Y}_L$ into $\boldsymbol{z}^{(L')} \in \mathbb{R}^{K \times d_1}$, yielding latents

whose dimension is *independent* of the number of spatial locations $N$. This cross-attention aggregation turns the function encoder into a discretization-invariant map from sampled fields to compact global observables. A final linear head projects these tokens to $z_t \in \mathbb{R}^D$ with $D \ll N$. Viewed continuously, cross-attention implements a learnable integral transform $\boldsymbol{v}(\cdot) \mapsto \int \widetilde{\boldsymbol{\kappa}}(\xi, \boldsymbol{v}(\xi)) \, \boldsymbol{v}(\xi) \, d\xi \in \mathbb{R}^D$. Further details are discussed in Appendix D.1.

**Learning observation functionals.** The encoder in Fig. 2 actually realizes the map $\mathcal{H} \xrightarrow{\text{delay embedding}} \mathcal{H}^{\otimes l} \to \mathbb{R}^D$, i.e., it learns observation functionals for the functional Koopman setting. Self-attention acts as function→function integral operators, while cross-attention performs function→vector aggregation (pooling) on irregular grids, yielding compact, discretization-invariant observables.

**Decoder $\mathcal{D}_{\boldsymbol{\theta}}$.** While the encoder queries from delay-embedded state $\{\boldsymbol{u}_{\text{delay}}(t, x_i)\}_{i=1}^N$, the decoder retrieves these information at randomly-located query points $\{y_j\}_{j=1}^M$ from the latent embeddings $z$. We employ FourierNet (Fathony et al., 2021) as the backbone of our decoder. The forward recursion of $\mathcal{D}_{\boldsymbol{\theta}}$ can be summarized as:

$$h^{(i+1)}(y) = \phi(W_i y) \odot \left( A_i h^{(i)} + B_i z + \text{MLP}(z) + b_i \right), \quad (6)$$

where $h^{(i)}(y)$ denotes the hidden representation of the target function $u(\cdot) \in \mathcal{H}$, $\phi$ denotes $\sin - \cos$ Fourier basis functions, and $\odot$ stands for element-wise multiplication. The latent terms $B_i z$ and $\text{MLP}_i(z)$ provide FiLM-style conditioning that modulates the amplitudes of Fourier features, yielding an implicit function $\widehat{u}(y; z) = \mathcal{D}_{\boldsymbol{\theta}}(z)(y)$ that can be queried at arbitrary (possibly irregular) locations as well as on full grids. This latent-modulated Fourier parameterization directly represents functions and supports resolution-free evaluation. Related latent-conditioned decoders have

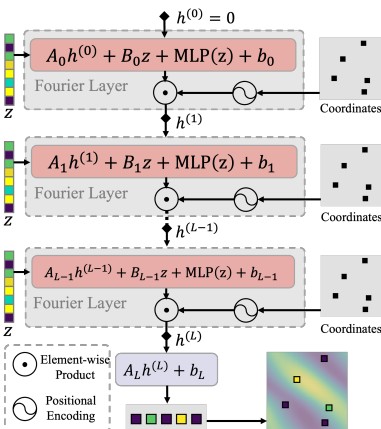

*Figure 3.* FourierNet-based function decoder architecture.

been employed in Yin et al. (2023) to obtain implicit neural representations. See Appendix D.2 for further properties of the decoder.

The encoder-decoder architecture, under the guidance of loss function (17), learns **nonlinear** observation functionals that span *approximately* **linear** invariant subspace of the Koopman operator.

### 4.4. Reduced-Order Modeling via Memory-augmented Koopman Learning

As a by-product of Koopman learning, we obtain a surrogate dynamics model for the underlying infinite-dimensional PDE (here we use leaky memory model as an illustration):

$$\begin{aligned} \text{encode: } & z_0 = \mathcal{E}_{\phi^*}(u(0, \cdot)), \\ \text{latent: } & \begin{cases} z_{t+1} = A z_t + r \odot \Phi_{\text{dec}}(m_t) \\ m_t = \gamma \odot m_{t-1} + \Phi_{\text{enc}}(z_t) \end{cases}, \quad (7) \\ \text{decode: } & u(t, x) = \mathcal{D}_{\theta^*}(z_t)(x). \end{aligned}$$

The effective dimensionality of the spatiotemporal dynamics is significantly reduced to $D$, the dimension of the latent state $z$. Moreover, on top of the pre-trained encoder-decoder, we can train a projection head $U \in \text{St}(d, D)$ (stiefel manifold (Edelman et al., 1998) consisting of matrices satisfying $U^* U = I_d$) to further reduce the dimensionality of the latent space (we provide the rationale for the Stiefel constraint in Appendix I.1). Once $U$ is learned, we train another memory-augmented model to capture the evolution of the projected latents $\beta_k = U^\top z_k$ in the reduced space $\mathbb{R}^d$. We cover the details of reduced order modeling (ROM) in Appendix C.3.

## 5. Experiments

### 5.1. Experiments Design

**Datasets.** We evaluate on two synthetic PDE datasets and one real-world dataset. (i) 2D wave equation: $\frac{\partial^2 u}{\partial t^2} = c^2 \Delta u$, where $u$ denotes the displacement field and $c > 0$ is the wave speed. (ii) 2D incompressible Navier–Stokes equations: $\partial_t \omega + u \cdot \nabla \omega = \nu \Delta \omega + f$, with the incompressibility constraint $\nabla \cdot u = 0$; here $\omega$ denotes the vorticity, $u$ the velocity field, $\nu > 0$ the viscosity, and $f$ a prescribed external forcing. (iii) Sea-surface temperature (SST): daily fields from the CMEMS Global Ocean Physics Reanalysis, an eddy-resolving global product at $1/12°$ horizontal resolution. Details for all datasets are deferred to Appendix F.

**Experiment designs.** We design four classes of experiments to extensively evaluate the performance of our proposed framework. **1. Long-term prediction with the memory-augmented model.** To assess long-term forecasting capability of the memory-augmented model (7), we construct prediction tasks on all datasets mentioned above. Following section 2.2 (also see Appendix B), we split trajectories into training and test sets with different initial conditions and random observation grids. We train the model via Phase I–II on the training set and then extrapolate on the test set given a short sequence of conditioning

*Table 2.* **Temporal prediction performance within and beyond the training horizon.** Using initial conditions unseen during training, we report MSE (↓) over the full inference interval, partitioned into (i) within the training horizon and (ii) rollout beyond the training horizon. For MERLIN, we report the performance of the linear backbone without memory augmentation, the leaky memory (augmented) model (LMM), and the finite memory (augmented) model (FMM) for comparison.

| Model | Wave | | Navier Stokes | | SST | |
|---|---|---|---|---|---|---|
| | Training horizon | Test horizon | Training horizon | Test horizon | Training horizon | Test horizon |
| FNO | **1.012e-5** | **5.426e-5** | **2.797e-5** | **3.967e-4** | 6.579e-2 | 2.693e-1 |
| KNO | 6.256e-4 | 4.623e-3 | 5.001e-4 | 2.859e-3 | 4.878e-2 | 1.331e-1 |
| DeepONet | 1.827e-2 | 3.136e-2 | 9.476e-3 | 1.224e-2 | - | - |
| UNO | 1.313e-3 | 3.817e-3 | 6.143e-4 | 2.891e-3 | 1.019e0 | 1.038e0 |
| GKT | 6.339e-4 | 1.829e-2 | 6.823e-2 | 5.521e-1 | 6.298e-2 | 1.573e-1 |
| UNet | 7.893e-4 | 6.024e-3 | 9.305e-4 | 5.858e-3 | - | - |
| SIREN | 1.783e-2 | 2.263e-2 | 4.392e-2 | 2.172e-1 | - | - |
| DINo | 1.361e-4 | 4.455e-4 | 9.329e-4 | 3.871e-3 | 5.001e-2 | 1.169e-1 |
| DeepKAE | 3.569e-2 | 5.291e-2 | 1.838e-2 | 4.510e-2 | 7.324e-2 | 1.173e-1 |
| **MERLIN** (linear) | 1.716e-4 | 6.150e-4 | 2.025e-3 | 9.527e-3 | 4.102e-2 | 1.025e-1 |
| **MERLIN** (LMM) | 6.403e-5 | 2.073e-4 | 4.675e-4 | 2.237e-3 | **2.923e-2** | **8.481e-2** |
| **MERLIN** (FMM) | **6.194e-5** | **1.659e-4** | **4.590e-4** | **2.035e-3** | **2.893e-2** | **7.510e-2** |

frames. We report the mean squared error (MSE) on the training horizon and on the test horizon of the test dataset. **2. Flexibility with random field observations.** Owing to the discretization-invariant encoder and resolution-free decoder, our model naturally handles random, partial observations. On synthetic datasets, we simulate random partial observations by subsampling the field at at different missing ratios $r$. We examine the model performance under $r \in \{25\%, 50\%, 75\%, 95\%\}$, with the same setup as in experiment 1. **3. Reduced-order modeling.** Koopman-learned observables (dynamic modes) are inherently interpretable and can be viewed as approximations to eigen-functionals of the Koopman operator. Furthermore, building on the linear backbone, we attach projection heads to reduce the latent dimension to $d$. We report results for the reduced dynamics with $d \in \{64, 32, 16, 8, 4, 2\}$ for NS ($d = 128$ for the original model used for Experiment 1 and 2), $d \in \{32, 16, 8, 4\}$ for wave ($d = 64$ for the original model). **4. Ablation studies.** Across all datasets, using the Experiment 1 setup, we compare the performance of linear backbone with its memory-augmented counterpart, thereby validating the roles of both components. We report sensitivity analyses for key hyperparameters in Appendix I.2.

**Baselines.** We reimplement several representative models spanning neural operators, implicit neural representations (INR), CNN-based surrogates, and a finite-dimensional Koopman-learning baseline. **Neural operators**: FNO (Li et al., 2021), KNO (Koopman-based FNO) (Xiong et al., 2024), DeepONet (used autoregressively as in (Wang & Perdikaris, 2023)) (Lu et al., 2021), UNO (Rahman et al., 2023), and GKT (Cao, 2021); **CNN-based**: U-Net (Ronneberger et al., 2015); **INR**: SIREN (Sitzmann et al., 2020) (adapted to the current PDE forecasting setting) and DINo (Yin et al., 2023); **Finite-dimensional baseline**:

DeepKAE(Lusch et al., 2018)(deep Koopman autoencoder that learns low-dimensional latent linear dynamics).

### 5.2. Experiments Results

**Long-term prediction.** Table 2 summarizes MSE over the training and test horizons across different models. For MERLIN, we report the linear backbone and the two memory-augmented variants. On the synthetic PDE benchmarks (Wave and Navier–Stokes), FNO attains the lowest MSE in both regimes, while MERLIN with memory augmentation is consistently second-best. Crucially, MERLIN exhibits substantially lower error accumulation when extrapolating beyond the training horizon (Wave: MERLIN-FMM $\approx 2.7\times$ vs. FNO $5.4\times$; Navier–Stokes: MERLIN-FMM $\approx 4.4\times$ vs. FNO $\approx 14.2\times$), indicating improved long-horizon stability. On the real-world SST task, MERLIN outperforms all baselines by a clear margin. **Furthermore**, the MERLIN linear backbone already outperforms the finite-dimensional baseline, while memory brings additional gains. This suggests that the improvement comes from the combination of the Koopman backbone and memory closure, rather than from either component alone. Moreover, LMM and FMM yield comparable performance, suggesting robustness of MERLIN to memory parameterization. Visualizations of the long-term predictions provided by our model are shown in Appendix H. Moreover, since rollouts are performed in a low-dimensional latent space, MERLIN is computationally much more efficient at inference time; a detailed runtime comparison is provided in Appendix G.4 Table 5.

**Flexibility with random observations.** We compare against models designed for handling irregular grids and pointwise queries (SIREN, DINo, and DeepONet); quantitative results are reported in Appendix G.3 Table 3. MER-

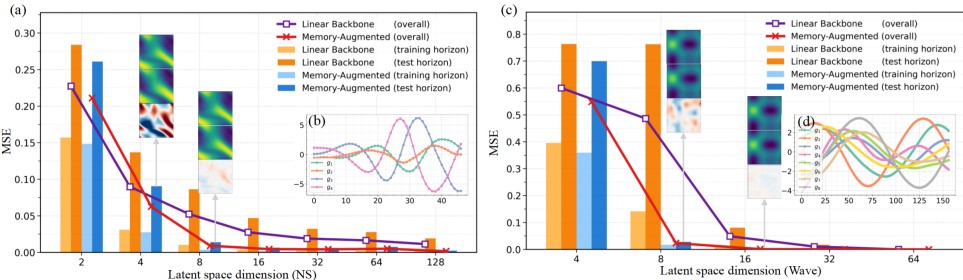

*Figure 4.* **Linear backbone vs. memory-augmented model (NS).** (a) Latent phase-space trajectories in a 2D PCA subspace. "——": encoded from ground-truth states; "——": generated by the linear model; "——": generated by the memory-augmented model. (b) Top: predictions on Navier–Stokes up to 20 s (every 2 s) using the linear model and the corresponding error maps; bottom: heatmap and eigenvalue spectrum of the linear propagator. (c) Top: predictions of the memory-augmented model and error maps; bottom: energy contributions of the linear vs. memory components.

*Figure 5.* **ROM results.** Left (NS): (a) performance of linear vs. memory-augmented ROM across $d \in \{2, 4, 8, 16, 32, 64, 128\}$; (b) time evolution of observables for $d = 4$. Right (Wave): (c) ROM; (d) time evolution of observables for $d = 8$.

LIN achieves the best MSE in most settings and degrades more gracefully as the mask ratio increases. We attribute this behavior to (i) treating measurements as discrete representations of continuous fields and learning observation functionals via an integral transform that can be stably approximated by numerical quadrature, and (ii) the Koopman linear backbone, which regularizes latent dynamics and promotes globally consistent behavior.

**Linear backbone vs. memory-augmented model.** Complementing the numerical comparison in Table 2, Figure 4 further compares the linear backbone with its memory-augmented counterpart. We observe that (i) the linear backbone *already* captures most evolutionary patterns; (ii) introducing the memory pathway pulls the linear trajectory toward the ground-truth latent ("——"), which translates into a notable reduction of prediction error in the decoded fields; and (iii) by calculating the energy contributions of the linear and memory terms, we find that a fairly *weak* memory injection *suffices* to correct the dynamics toward the ground truth, thereby *preserving interpretability and the dominance* of the Koopman backbone.

**Interpretability and ROM.** Our framework enables interpretability and ROM benefiting from the Koopmanism. Following Section 4.4, we further compress the model by appending a projection head constrained to the Stiefel manifold, yielding a $d$-dimensional latent subspace. As shown in Figure 5(a,c), moving from the linear to the memory-augmented model yields a pronounced MSE reduction; moreover, the improvement exhibits a *critical transition*: a sharp MSE drop occurs when increasing the latent dimension from $d = 4 \rightarrow 8$ for NS and from $d = 8 \rightarrow 16$ for Wave. These patterns indicate an ultra-low intrinsic dimensionality despite the systems' infinite-dimensional nature, enabling faithful surrogate evolution via a low-dimensional non-Markovian model. From panels (b,d), we recover oscillatory Koopman modes. For Navier–Stokes (b), amplitudes are not preserved and a clear 0-10-frame transient is observed ($\approx$0-10 s). For Wave, both frequency and amplitude are approximately preserved, consistent with a conservative (energy-preserving) system. A complementary comparison with Figures 4 and 16 demonstrates that, the linear and memory energy contributions for wave remain unchanged over time, and latent trajectories evolve near equipotential contours—corroborating the interpretability of our paradigm.

### 5.3. Further Discussion

• **Mitigation of spectral bias.** As shown in Table 2, MER-LIN slightly underperforms FNO on in-horizon MSE. We attribute this gap to FNO's built-in Fourier bias: it applies FFTs on regular grids and learns matrix multipliers in the frequency domain, which is highly favorable when the dynamics are nearly diagonalizable in a Fourier basis. MERLIN,

by contrast, does not perform per-layer FFTs and naturally accommodates irregular and partial observations. As discussed in Appendix D.3, FNO uses fixed sinusoidal bases to parametrize the solution operator, whereas our encoder *learns* kernel bases via attention. Since synthetic PDEs are indeed Fourier-friendly (e.g., the wave equation is exactly diagonalizable), we inject explicit spectral bias into MERLIN by augmenting the projections that form $Q, K, V$ with learnable Fourier embeddings. Experiments in Appendix G.6 show this improves in-horizon accuracy while preserving MERLIN's long-horizon stability. • **Potential on more challenging benchmarks.** FNO's strong performance is partly attributable to its Fourier inductive bias; however, chaos may undermine this Fourier bias. To explicitly target this setting, we stress-test MERLIN on the Kuramoto–Sivashinsky (KS) equation (dataset details in Appendix F; results in Appendix G.7). MERLIN attains substantially lower test-horizon error and smaller train-to-test gap than FNO, indicating more robust long-horizon generalization on a chaotic PDE. Separately, to assess *scalability* beyond controlled benchmarks, we evaluate MERLIN on the real-world ERA5 dataset, where it also outperforms FNO; quantitative results are summarized in Table 8 with qualitative visualizations in Figure 15. • **Quality of latent representation.** MERLIN's latent representation is not only predictive but also *informative*: (i) solving an inverse initial-condition recovery problem *directly in latent space* via a Tikhonov objective yields accurate reconstructions, indicating that the latent manifold retains sufficient information (Appendix G.5, Fig. 6); and (ii) simple linear probes on frozen latents achieve near-perfect decodability of global observables, suggesting that physically meaningful quantities are almost linearly encoded in the latent state (Fig. 7).

## 6. Related Works

**Neural operators.** Neural solution operators are a standard route to data-driven PDE surrogates, amortizing computation across equation families for fast inference (Chen & Chen, 1995; Bhattacharya et al., 2021; Lu et al., 2021; Kovachki et al., 2023). Representative designs include Fourier-based operators (FNO (Li et al., 2021); FFNO (Tran et al., 2023)), Transformer-style surrogates with spatial attention (Cao, 2021; Li et al., 2023b), and U-Net derived models (Gupta & Brandstetter, 2022; Rahman et al., 2023). Many neural-operator models assume structured discretizations. FNO is typically formulated on uniform grids, as its FFT-based spectral convolutions presuppose regular sampling, which limits its flexibility in encoding spatial observations; while extensions (e.g., Li et al. (2023a)) mitigate geometric constraints, they do not natively support querying unobserved spatial locations. DeepONet (Lu et al., 2021) commonly fixes sensor locations during training and testing.

**Encoder-latent-decoder paradigm and memory correction.** A large class of scientific machine learning methods represents dynamics in latent space via an encoder-latent-decoder pipeline, with latent evolution parameterized by Koopman linear propagators (Lee & Carlberg, 2021; Wiewel et al., 2019; Morton et al., 2018; Li et al., 2019; Pan & Duraisamy, 2020; Lusch et al., 2018; Wang et al., 2023; Liu et al., 2023; Naiman et al., 2024; Azencot et al., 2020; Gao et al., 2026), neural ODEs (Song et al., 2024; Buzhardt et al., 2025), spectral or latent-spectral models (Wu et al., 2023), or ROM/RNN hybrids (Vlachas et al., 2022; Tomasetto et al., 2025). For infinite-dimensional systems, Koopman frameworks and PDE-level formulations have been developed (Budišić et al., 2012; Mauroy, 2021; Nakao & Mezić, 2020), but most practical architectures remain grid-based on discretized fields (Kutz et al., 2018; Brunton & Kutz, 2023) and tacitly assume a finite-dimensional invariant subspace that is difficult to identify in practice (Colbrook et al., 2023). Despite these developments, a unified function-space framework that is robust to random, irregular observations is still lacking. From a complementary perspective, the Mori-Zwanzig formalism explains how unresolved degrees of freedom induce non-Markovian memory on resolved observables (Mori, 1965; Zwanzig, 1973; 2001), motivating recent data-driven memory-operator estimators (Ma et al., 2019; Lin et al., 2021; 2023; Gupta et al., 2025; Menier et al., 2025; Buitrago Ruiz et al., 2025). Within this broader landscape, MERLIN adopts a functional Koopman view of PDEs and couples Koopman backbone with explicit MZ-inspired memory module, yielding a resolution-free model that handles irregular observations while retaining an interpretable linear-plus-memory latent decomposition. This stands in contrast to more black-box surrogate modeling approaches.

## 7. Conclusion

Extensive experiments demonstrate that MERLIN delivers accurate predictions, handles random partial observations, and supports interpretable ROM. Guided by the functional Koopman-MZ perspective, MERLIN augments a linear backbone with efficient memory modules, leading to stable long-horizon rollouts while retaining an interpretable linear-plus-memory dynamics. Our analyses further reveal latent physical structure in several settings, suggesting the potential of MERLIN for data-driven discovery of governing laws. A current limitation is that MERLIN focuses on memory closure and neglects the fluctuation term in the full Mori-Zwanzig decomposition, which is most justified when the unresolved dynamics are dissipative or act as small effective noise. Future work will scale MERLIN to more complex systems and develop richer memory and fluctuation closures for different dynamical regimes.

## Acknowledgements

Q. Zhu is supported by the National Key R&D Program of China (No. 2025YFA1016503), by the National Natural Science Foundation of China (NSFC) (Nos. T2541024, 62406072, and 12171350), and by the STCSM (No. 23YF1402500). W. Lin is supported by the NSFC (Nos. 11925103, 12531018), the STCSM (Nos. 2021SHZDZX0103, 22JC1402500, 22JC1401402, 25JS2810400), and the SMEC (Nos. 2023ZKZD04, 2023KEJ105-72).

## Impact Statement

This paper presents a technical contribution to machine learning for spatiotemporal dynamical systems. By learning from random, partial, and irregular observations, MERLIN may support applications where full-field measurements are expensive or unavailable, such as reconstructing sparse geophysical fields, modeling dynamics from sensor networks, and building efficient reduced-order surrogates for weather, ocean, and physical systems. These capabilities may help improve scientific understanding and reduce the computational cost of large-scale simulation and forecasting. At the same time, predictions from learned surrogate models can be unreliable outside the data distribution or under regime shifts, and should not be used as standalone decision-making tools in safety-critical settings without careful validation and domain oversight. This work does not introduce new data collection involving human subjects or direct deployment in high-stakes systems.

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

# A. Theory

## A.1. Preliminaries and Notations on PDE

Below we provide supplementary material for parts of the main text concerning PDEs, chiefly to define notation that was left unexplained. For completeness and the reader's convenience, we also reproduce some content from the main text.

**Abstract Cauchy formulation.** We reformulate the spatiotemporal dynamics as the evolution of abstract states $u(t, \cdot) \triangleq u_t \in \mathcal{H}$ and write

$$\dot{u}(t) = \mathcal{F}(t, u(t)), \qquad u(s) = u_s \in \mathcal{H}, \tag{8}$$

where $\mathcal{F}(t, u) \in \mathcal{H}$ is given by $\mathcal{F}(t, u)(x) = \mathcal{L}[u](t, x)$ for $x \in \Omega$. We fix a realization of the spatial operator $\mathcal{L}$ on $\mathcal{H}$ whose domain $\mathcal{D}(\mathcal{L}(t)) \subset \mathcal{H}$ encodes the boundary conditions via traces on $\partial\Omega$, or periodicity via the choice of a periodic function space.

**Flow map.** Well-posedness yields a two-parameter family of solution operators, the *flow*, $\Phi_{t,s} : \mathcal{H} \to \mathcal{H}$, $t \geq s$, defined by $\Phi_{t,s}(u_s) = u(t)$ where $u$ solves (2). These operators satisfy the cocycle properties

$$\Phi_{s,s} = I, \qquad \Phi_{t,r} \circ \Phi_{r,s} = \Phi_{t,s} \quad (t \geq r \geq s), \tag{9}$$

and are jointly continuous in $(t, s, u)$ on the set where solutions exist; moreover, if $\mathcal{F}$ is locally Lipschitz in $u$ on bounded subsets (uniformly on compact time intervals), then $\Phi_{t,s}$ is locally Lipschitz in $u$ on bounded subsets of $\mathcal{H}$. In the *autonomous* case, $\mathcal{F}(t, u) = \mathcal{F}(u)$, the flow reduces to a one-parameter $C_0$ (semi-)group $S_t := \Phi_{t,0}$ with $S_{t+s} = S_t S_s$; in the linear autonomous case where $\mathcal{L}$ generates a $C_0$-semigroup, one writes $S_t = e^{t\mathcal{L}}$.

**Standing assumptions.** We assume throughout that the abstract Cauchy problem (2) on $\mathcal{H}$ is *autonomous* and *well posed*—that is, it admits a unique solution with continuous dependence on the initial state. This well-posedness induces the semiflow $S_t := \Phi_{t,0}$. We also assume that $(\mathcal{H}, \|\cdot\|)$ is a separable Hilbert space.

**Fréchet differentiable functionals and function spaces.** In the main text we call a functional an observable, henceforth, we do not distinguish between the two terms. Consider $g : \mathcal{H} \to \mathbb{R}^m$. We say that $g$ is Fréchet differentiable at $x$ if there exists a bounded linear map $Dg(x) : \mathcal{H} \to \mathbb{R}^m$ such that

$$\lim_{\|h\| \to 0} \frac{\|g(x+h) - g(x) - Dg(x)[h]\|}{\|h\|} = 0.$$

Higher derivatives $D^j g(x)$ are continuous $j$-linear maps on $\mathcal{H}^j$. Write $C^1(\mathcal{H}; \mathbb{R}^m)$ for the class of Fréchet $C^1$ maps (i.e., $x \mapsto Dg(x)$ is continuous in the operator norm). We use the observable space

$$\mathcal{O} := \left\{ f : \mathcal{H} \to \mathbb{R} \, \middle| \, f \in C^1(\mathcal{H}; \mathbb{R}), \ \|f\|_\infty + \sup_{u \in \mathcal{H}} \|Df(u)\| < \infty \right\}, \tag{10}$$

where $\|Df(u)\|$ denotes the operator norm. Vector-valued observables are handled componentwise.

## A.2. Koopman theory for PDEs and generalized Langevin equation

Although the theory of Koopman operators for PDEs has been discussed in the literature (Nakao & Mezić, 2020), we collect the relevant results here for completeness and the reader's convenience.

**Properties of koopman operator.** In the main text, we use the following properties of the Koopman operator; we collect their proofs below.

- **Identity:** $\mathcal{K}_0 g[u] = g[S_0 u] = g[\Phi_{0,0} u] = g[u]$.
- **Linearity:** $\mathcal{K}_t(c_1 g_1[u] + c_2 g_2[u]) = c_1 g_1[S_t u] + c_2 g_2[S_t u] = c_1 \mathcal{K}_t g_1[u] + c_2 \mathcal{K}_t g_2[u]$ for any $c_1$ and $c_2$.
- **Commutativity:** $g[S_s S_t u] = S_s S_t g[u]$ and $g[S_t S_s u] = S_t S_s g[u]$ imply $\mathcal{K}_t \mathcal{K}_s g[u] = \mathcal{K}_s \mathcal{K}_t g[u]$.
- **Composition equivariance:** To verify $\mathcal{K}_t f(g[\cdot]) = f(\mathcal{K}_t g[\cdot])$, we just need to verify for all state $u$, $\mathcal{K}_t f(g[u]) = f(\mathcal{K}_t g[u])$, and we have $\mathcal{K}_t f(g[u]) = f(g[S_t u]) = f(\mathcal{K}_t g[u])$ by definition.

**Infinitesimal generator.** The infinitesimal generator $\mathcal{A}$ of Koopman operator $\mathcal{K}_t$ is defined as

$$\mathcal{A}g[u] = \lim_{\tau \to 0} \frac{\mathcal{K}_\tau g[u] - g[u]}{\tau}.$$

We give a brief explanation. Specifically,

$$\mathcal{A}g[u] = \lim_{\tau \to 0} \frac{\mathcal{K}_\tau g[u] - g[u]}{\tau} = \int_\Omega \mathcal{F}(u(x)) \frac{\delta g[u]}{\delta u(x)} \mathrm{d}x,$$

where $\mathcal{F}(u(x))$ is defined in 2, and $\frac{\delta g[u]}{\delta u(x)}$ is a functional derivative of $g[u]$ respect to $u(x)$. This derivative was mentioned in (Nakao & Mezić, 2020), where it is interpreted as the Gâteaux derivative; in this paper, its validity is guaranteed by assuming the stronger Fréchet differentiability in 10.

**From finite-dimensional invariance to memory-corrected reduced dynamics.** Many Koopman learning frameworks implicitly assume the existence of a non-trivial finite-dimensional *invariant subspace* $\mathcal{M} = \mathrm{span}\{g_1, \ldots, g_D\}$ (Colbrook et al., 2023). If such an invariant subspace $\mathcal{M}$ exists and can be explicitly parametrized, for instance via a prescribed dictionary of observables or a learned neural encoder, then one could in principle obtain a *closed* finite-dimensional *linear* dynamics $\frac{\mathrm{d}}{\mathrm{d}t} \mathbf{g}(t) = A \mathbf{g}(t)$ on subspace $\mathcal{M}$, where $\mathbf{g} = (g_1, \ldots, g_D)^\top$. In practice, however, (i) even when invariant finite-dimensional subspaces exist, identifying a finite family of observables that exactly closes the dynamics is notoriously difficult, and (ii) in many systems the assumption fails outright, even for ODEs, let alone for PDE-governed dynamics where observables live in an infinite-dimensional functional space $\mathcal{O}$.

This motivates a different viewpoint: rather than insisting on exact finite-dimensional closure, we fix an ***arbitrary*** finite family of observables $\{g_i\}_{i=1}^D$ and treat their span $\mathcal{M} = \mathrm{span}\{g_i\}$ as the "resolved" subspace, with the complement playing the role of unresolved observables. The central question then becomes:

> Given a finite family of resolved observables, what is the exact closed evolution equation satisfied by the resolved observables after the unresolved (i.e., not explicitly modeled) observables have been eliminated?

The classical Mori–Zwanzig (MZ) formalism (Mori, 1965; Zwanzig, 1973), originally developed in non-equilibrium statistical mechanics (Zwanzig, 2001), answers precisely this question for finite-dimensional many-body systems. In that setting, one starts from a (typically finite-dimensional) system in full state variables, chooses a projection $\mathcal{P}$ that maps functions of the full state to functions of a low-dimensional set of "resolved" macroscopic variables, and then applies Dyson's formula to derive an *effective evolution* for these resolved variables. The resulting generalized Langevin equation consists of a Markovian term, a time-nonlocal memory integral, and a fluctuation term induced by the unresolved dynamics.

**Projection-operator formulation of the generalized Langevin equation in the functional Koopman setting.** We conceptually extend the Mori–Zwanzig formalism to the functional Koopman setting for PDEs, where the chosen observables $\{g_i\}_{i=1}^D$ play the role of resolved variables and all observables lying in the complement of $\mathcal{M}$ are treated as unresolved. Specifically, given a finite family of observables $\{g_i\}_{i=1}^D$, we let $\mathcal{M} = \mathrm{span}\{g_i\} \subset \mathcal{O}$ denote the resolved observable subspace, define a projection $\mathcal{P} : \mathcal{O} \to \mathcal{M}$ and $\mathcal{Q} = I - \mathcal{P}$ onto the unresolved complement, and consider the Koopman generator $\mathcal{A}$. At the operator level, Dyson's identity for $e^{t\mathcal{A}}$ yields an exact decomposition of the Koopman evolution into a Markovian part acting on $\mathcal{M}$, a memory integral that encodes the influence of $\mathcal{Q}\mathcal{O}$, and a fluctuation term driven by the orthogonal dynamics. The algebraic form of this decomposition mirrors the classical MZ theory; what is new here is its ***reinterpretation*** in terms of functional observables for infinite-dimensional PDE dynamics, and its explicit use to motivate our "linear Koopman backbone + memory correction" architecture in latent space.

For the purposes of this work, it suffices to state the resulting generalized Langevin equation for the resolved observables, given in the theorem below; a more detailed discussion and an analysis of its limitations are deferred to Appendix A.3.

**Theorem** (Generalized Langevin equation). *Let* $\mathbf{g}_{\mathcal{M}}(t) = [g_1(t), \ldots, g_D(t)]^\top$ *denote the time evolution of a finite family of observables under the Koopman dynamics, where the* $g_i$ *are arbitrary (linear or nonlinear, fixed or learned, not necessarily linearly independent). Then* $\mathbf{g}_{\mathcal{M}}$ *admits a generalized Langevin representation of the form*

$$\frac{\mathrm{d}}{\mathrm{d}t} \mathbf{g}_{\mathcal{M}}(t) = \mathbf{M} \mathbf{g}_{\mathcal{M}}(t) + \int_0^t \mathbf{K}(s, \mathbf{g}_{\mathcal{M}}(t-s)) \, \mathrm{d}s + \mathbf{F}(t), \tag{11}$$

*where* $\mathbf{M}$ *is a Markovian (instantaneous) transition matrix acting on the resolved observables,* $\mathbf{K}$ *is a memory kernel capturing the influence of unresolved observables through time-nonlocal interactions, and* $\mathbf{F}(t)$ *is a fluctuation term induced by the orthogonal dynamics on the unresolved subspace.*

**Proof.** Define the projection operator $\mathcal{P} : \mathcal{O} \to \mathcal{M}$ with $\mathcal{P}^2 = \mathcal{P}$ and $\mathcal{Q} \triangleq I - \mathcal{P}$. By construction, (i) $\mathcal{P}g_i = g_i$, (ii) $\mathcal{P}\mathcal{A}\mathcal{P}g_i \in \mathcal{M}$, and furthermore (iii) $\mathcal{P}\mathcal{A}\mathcal{P}g_i = \sum_{j=1}^{D} a_{i,j}g_j$. According to (3) along with the definition of projection operator, the evolution of any observable $g \in \mathcal{O}$ can be represented as

$$\frac{\mathrm{d}}{\mathrm{d}t}\mathcal{K}_t g = \frac{\mathrm{d}}{\mathrm{d}t}e^{t\mathcal{A}}g = \mathcal{A}g(t) = e^{t\mathcal{A}}(\mathcal{P} + \mathcal{Q})\mathcal{A}g = e^{t\mathcal{A}}\mathcal{P}\mathcal{A}g + e^{t\mathcal{A}}\mathcal{Q}\mathcal{A}g. \tag{12}$$

For the second term, by applying Dyson's formula (Zwanzig, 2001)

$$e^{t\mathcal{A}} = e^{t\mathcal{Q}\mathcal{A}} + \int_0^t e^{(t-s)\mathcal{A}}\mathcal{P}\mathcal{A}e^{s\mathcal{Q}\mathcal{A}}\mathrm{d}s, \tag{13}$$

we have

$$e^{t\mathcal{A}}\mathcal{Q}\mathcal{A}g = e^{t\mathcal{Q}\mathcal{A}}\mathcal{Q}\mathcal{A}g + \int_0^t e^{(t-s)\mathcal{A}}\mathcal{P}\mathcal{A}e^{s\mathcal{Q}\mathcal{A}}\mathcal{Q}\mathcal{A}g\mathrm{d}s. \tag{14}$$

Note that $\mathcal{P}\mathcal{A}e^{s\mathcal{Q}\mathcal{A}}\mathcal{Q}\mathcal{A}g \in \mathcal{M}$ by the projection property, we denote this quantity by a (generally nonlinear) function $K(s,g)$ of the time index $s$ and $g$. By the composition equivariance of Koopman opertor, we have

$$e^{(t-s)\mathcal{A}}K(s,g) = \mathcal{K}_{t-s}K(s,g) = K(s,g(t-s)). \tag{15}$$

Substituting (15) into (14), and (14) into (12), (12) becomes

$$\frac{\mathrm{d}}{\mathrm{d}t}g(t) = e^{t\mathcal{A}}\mathcal{P}\mathcal{A}g + \int_0^t K(s,g(t-s))\mathrm{d}s + e^{t\mathcal{Q}\mathcal{A}}\mathcal{Q}\mathcal{A}g.$$

When $g = g_i \in \mathcal{M}$, for the first term, we use properties of projection operator to interpret it as

$$e^{t\mathcal{A}}\mathcal{P}\mathcal{A}g_i = e^{t\mathcal{A}}\mathcal{P}\mathcal{A}\mathcal{P}g_i = e^{t\mathcal{A}}\sum_{j=1}^{D}a_{i,j}g_j = \sum_{j=1}^{D}a_{i,j}e^{t\mathcal{A}}g_j = \sum_{j=1}^{D}a_{i,j}g_j(t), \tag{16}$$

which depends linearly on observables in $\mathcal{M}$. Denoting evolution of the unresolved observable $e^{t\mathcal{Q}\mathcal{A}}\mathcal{Q}\mathcal{A}g$ as $F(t)$, we obtain the Generalized Langevin Equation (GLE)

$$\frac{\mathrm{d}}{\mathrm{d}t}g_i(t) = \sum_{j=1}^{D}a_{i,j}g_j(t) + \int_0^t K(s,g_i(t-s))\mathrm{d}s + F(t).$$

The derivation also hold for vector-valued functionals $\mathbf{g} : \mathcal{H} \to \mathbb{R}^D$. We use the compact vector notation $\mathbf{g}_{\mathcal{M}}(t) = [g_1, \cdots, g_D(t)]^{\top}$ and express the full dynamics as

$$\frac{\mathrm{d}}{\mathrm{d}t}\mathbf{g}_{\mathcal{M}}(t) = \mathbf{M} \cdot \mathbf{g}_{\mathcal{M}}(t) + \int_0^t \mathbf{K}(s,\mathbf{g}_{\mathcal{M}}(t-s))\mathrm{d}s + \mathbf{F}(t),$$

where $\mathbf{M}$ is the Markov transition matrix, $\mathbf{K}$ is the memory kernel, and $\mathbf{F}$ is the external forcing term (often referred to as noise). This completes the proof.

*Remark* A.1 (Applicability of Theorem A.2). The proof of Theorem A.2 only uses that the resolved space $\mathcal{M} = \mathrm{span}\{g_1, \ldots, g_D\} \subset \mathcal{O}$ is a finite-dimensional linear subspace generated by a finite family of observables $\{g_i\}_{i=1}^{D}$. In particular, the $g_i$ may be linear or nonlinear, fixed or learned by a neural network, and are not required to be linearly independent: for any finite set of observables, their linear span is always well-defined. When linear independence fails, this only affects the uniqueness of the decomposition in (16); the subspace $\mathcal{M}$ and the resulting generalized Langevin structure remain unchanged.

**Koopman Representation of the Generalized Langevin Equation.** A more intuitive route to the memory-corrected dynamics is to eliminate the unresolved variables directly, in a way that is closely parallel to the linear PDE case study in the main text.

Specifically, although finite-dimensional invariant subspace of $\mathcal{A}$ does not exist in general, but there exists a countable-dimensional functional subspace $\mathcal{R}$, $\mathcal{M} \subset \mathcal{R} \subset \mathcal{O}$, that remains invariant under the action of $\mathcal{A}$, i.e. $\mathcal{A}\mathcal{R} \subset \mathcal{R}$. A simple example is the Krylov subspace generated by cyclic action of $\mathcal{A}$, i.e. $\mathcal{R} = \text{span}\{\mathcal{M}, \mathcal{A}\mathcal{M}, \cdots, \mathcal{A}^s\mathcal{M}, \cdots\}$. With $\mathcal{R}$ in hand, we can restrict ourself to consider only the action of linear operator $\mathcal{A}_{\mathcal{R}} \triangleq \mathcal{P}_{\mathcal{R}}\mathcal{A}\mathcal{P}_{\mathcal{R}}$. Moreover, following the Gram-Schmidt process, we are able to construct a set of basis observation functionals $\{\overline{g}_i\}_{i=1}^{\infty} \subset \mathcal{R}$, that are linearly independent with functionals in $\mathcal{M}$, we denote $\text{span}\{\overline{g}_i\}_{i=1}^{\infty}$ by $\overline{\mathcal{M}}$. It is easy to verify that together with $\{g_i\}_{i=1}^{D}$, $\{g_i\}_{i=1}^{D} \cup \{\overline{g}_i\}_{i=1}^{\infty}$ forms a complete set of basis functionals of $\mathcal{R}$.

We now use the terse vector notation $\mathbf{g}_{\mathcal{M}}(t) = [g_1(t), g_2(t), \cdots, g_D(t)]^\top$, $\mathbf{g}_{\overline{\mathcal{M}}}(t) = [\overline{g}_1(t), \overline{g}_2(t), \cdots]^\top$. Under basis $\{g_i\}_{i=1}^{D} \cup \{\overline{g}_i\}_{i=1}^{\infty}$, the restricted linear operator $\mathcal{A}_{\mathcal{R}}$ admits a ***countable-dimensional matrix representation***, denoted by $L$ and the full-order dynamics of observables can then be described in the following compact form:

$$\frac{\mathrm{d}}{\mathrm{d}t}\begin{pmatrix} \mathbf{g}_{\mathcal{M}}(t) \\ \mathbf{g}_{\overline{\mathcal{M}}}(t) \end{pmatrix} = L \cdot \begin{pmatrix} \mathbf{g}_{\mathcal{M}}(t) \\ \mathbf{g}_{\overline{\mathcal{M}}}(t) \end{pmatrix} \triangleq \begin{pmatrix} L_{\mathcal{M}\mathcal{M}} & L_{\mathcal{M}\overline{\mathcal{M}}} \\ L_{\overline{\mathcal{M}}\mathcal{M}} & L_{\overline{\mathcal{M}}\overline{\mathcal{M}}} \end{pmatrix} \cdot \begin{pmatrix} \mathbf{g}_{\mathcal{M}}(t) \\ \mathbf{g}_{\overline{\mathcal{M}}}(t) \end{pmatrix}.$$

The matrix $L$, divided into four parts, quantify the interaction between $\mathcal{M}$ (resolved functional space) and $\overline{\mathcal{M}}$ (unresolved functional space) during forward propagation. Subsequently, to obtain closed-form dynamics for the resolved observables of interest in set $\{g_1, \cdots, g_D\}$, we eliminate $\mathbf{g}_{\overline{\mathcal{M}}}(t)$ by first solving the ODEs in $\mathbf{g}_{\overline{\mathcal{M}}}(t)$ driven by $\mathbf{g}_{\mathcal{M}}(t)$ and then substitute the solution into the ODEs in $\mathbf{g}_{\mathcal{M}}(t)$, which in turn leads to closed evolutionary equations for observables $\mathbf{g}_{\mathcal{M}}(t)$:

$$\frac{\mathrm{d}}{\mathrm{d}t}\mathbf{g}_{\mathcal{M}}(t) = L_{\mathcal{M}\mathcal{M}}\mathbf{g}_{\mathcal{M}} + L_{\mathcal{M}\overline{\mathcal{M}}}\int_0^t e^{(t-s)L_{\overline{\mathcal{M}}\overline{\mathcal{M}}}} \cdot L_{\overline{\mathcal{M}}\mathcal{M}} \cdot \mathbf{g}_{\mathcal{M}}(s)\mathrm{d}s + L_{\mathcal{M}\overline{\mathcal{M}}}e^{tL_{\overline{\mathcal{M}}\overline{\mathcal{M}}}} \cdot \mathbf{g}_{\overline{\mathcal{M}}}(0).$$

We now drop the subscript $\mathcal{M}$ in $\mathbf{g}_{\mathcal{M}}$, as we only care about the dynamics of the resolved observables. By defining an $D \times D$ matrix $\mathbf{M} \triangleq L_{\mathcal{M}\mathcal{M}}$, an $D \times D$ matrix $\mathbf{K}(s) \triangleq -L_{\mathcal{M}\overline{\mathcal{M}}}e^{sL_{\overline{\mathcal{M}}\overline{\mathcal{M}}}} \cdot L_{\overline{\mathcal{M}}\mathcal{M}}$ parametrized by $s \in \mathbb{R}^+$, and an $D \times 1$ matrix $\mathbf{F}(t) \triangleq L_{\mathcal{M}\overline{\mathcal{M}}}e^{sL_{\overline{\mathcal{M}}\overline{\mathcal{M}}}} \cdot \mathbf{g}_{\overline{\mathcal{M}}}(0)$, again, we arrive at the celebrated Generalized Langevin Equation (GLE) for observation *functionals*:

$$\frac{\mathrm{d}}{\mathrm{d}t}\mathbf{g}(t) = \mathbf{M} \cdot \mathbf{g}(t) - \int_0^t \mathbf{K}(t-s) \cdot \mathbf{g}(s)\mathrm{d}s + \mathbf{F}(t).$$

We refer to $\mathbf{M}$ as the Markov transition matrix, $\mathbf{K}(s)$ as the memory kernel, and $\mathbf{F}(t)$ as the orthogonal dynamics which is often referred to as noise and dropped because of its resemblance to a Langevin noise in a Langevin equation.

## A.3. Scope and Limitations of the functional Koopman–Mori–Zwanzig theory

In this section, we further clarify several limitations of the theory and of the resulting modeling choices.

**Impact of the fluctuation term.** Although the decomposition obtained via Dyson's formula $e^{t\mathcal{A}} = e^{t\mathcal{Q}\mathcal{A}} + \int_0^t e^{(t-s)\mathcal{A}}\mathcal{P}\mathcal{A}e^{s\mathcal{Q}\mathcal{A}}\,\mathrm{d}s$ is formally exact for any finite set of observables, the fluctuation term associated with the orthogonal dynamics, typically of the form $e^{t\mathcal{Q}\mathcal{A}}\mathcal{Q}\mathcal{A}\mathbf{g}$ for a vector of observables $\mathbf{g}$, need not be small. To obtain a applicable surrogate dynamics model, one usually assumes that the orthogonal dynamics generated by $\mathcal{Q}\mathcal{A}$ are sufficiently *dissipative* or *mixing* so that this term can be neglected or modeled as effective noise. Making this precise is subtle even in finite dimensions, and establishing rigorous conditions for infinite-dimensional PDEs and for general spaces of observables is, to the best of our knowledge, largely open. In this work we follow the standard modeling practice of *neglecting* this term and focus on the memory correction.

**Need for "good" observables.** Because the fluctuation term is neglected in our framework, it is crucial to choose (or learn) observables $\mathbf{g}$ for which $e^{t\mathcal{Q}\mathcal{A}}\mathcal{Q}\mathcal{A}\mathbf{g}$ is small. The ideal case is $\mathcal{A}\mathbf{g} \in \mathcal{M}$, i.e., the span of $\{g_i\}$ is invariant under the Koopman operator $\mathcal{A}$, so the fluctuation vanishes and the reduced dynamics are exactly Markovian. For nonlinear PDEs this is generically impossible. In MERLIN we therefore aim for *approximate* invariance: we seek observables that make the Koopman backbone as close to linear as possible and induce a memory kernel that decays on a moderate time scale. This motivates our combination of a linear backbone with leaky-/finite-memory modules, implicitly assuming that, in the learned observable space, the memory kernel is sufficiently decaying and any remaining fluctuation acts as a small residual. When this assumption fails (e.g., genuinely long-range memory or strongly non-dissipative unresolved dynamics), MERLIN would require a larger latent dimension or a richer memory model, and performance may degrade.

**Case study.**

> **Case study: linear PDE and emergence of memory**
>
> Let $H$ be a Hilbert space such as $L^2(\Omega)$ and consider a linear evolution equation $\partial_t u(t) = Lu(t)$, where $L$ generates a strongly continuous semigroup $S_t = e^{tL}$. The Koopman operator acts on observables $g : H \to \mathbb{R}$ via $(K_t g)(u) = g(S_t u)$. If $L$ is self-adjoint with an orthonormal eigenbasis $\{e_k\}_{k \geq 1} \subset H$ satisfying $Le_k = \lambda_k e_k$, then the linear functionals $g_k(u) = \langle e_k, u \rangle_H$ are Koopman eigen-observables, since $(K_t g_k)(u) = g_k(e^{tL}u) = e^{t\lambda_k} g_k(u)$.
>
> Writing $u(t) = \sum_{k \geq 1} a_k(t) e_k$, the PDE is equivalent to the infinite-dimensional linear ODE $\dot{a}(t) = Aa(t)$ for the coefficients $a(t) = (a_1(t), a_2(t), \dots)^\top$, with entries $A_{ij} = \langle e_i, Le_j \rangle_H$. We split $a(t) = (x(t), y(t))^\top$, where $x(t) \in \mathbb{R}^D$ collects the first $D$ modes and $y(t)$ the remaining ones, and write $A$ in block form $A = \begin{pmatrix} A_{11} & A_{12} \\ A_{21} & A_{22} \end{pmatrix}$.
>
> The dynamics read $\dot{x}(t) = A_{11}x(t) + A_{12}y(t)$ and $\dot{y}(t) = A_{21}x(t) + A_{22}y(t)$, and the resolved variables $x(t)$ are precisely the values of the observables on the PDE state, i.e., $x_k(t) = g_k(u(t)) = \langle e_k, u(t) \rangle_H$ for $k = 1, \dots, D$. Formally solving the unresolved dynamics gives $y(t) = e^{tA_{22}}y(0) + \int_0^t e^{(t-s)A_{22}} A_{21}x(s)\, ds$. Substituting into the $\dot{x}$ equation yields the classical linear MZ generalized Langevin equation $\dot{x}(t) = A_{11}x(t) + \int_0^t K(t-s)\, x(s)\, ds + F(t)$, with memory kernel $K(\tau) = A_{12}e^{\tau A_{22}} A_{21}$ and "noise" term $F(t) = A_{12}e^{tA_{22}}y(0)$ due to unresolved initial conditions. The reduced dynamics become exactly Markovian and closed (no memory, no fluctuation) if and only if the resolved subspace is invariant under $A$, i.e., $A_{12} = A_{21} = 0$, in which case $K(\tau)$ and $F(t)$ vanish. More generally, if $A_{22}$ generates a strongly dissipative (or strongly stable) semigroup, then $e^{tA_{22}}$ decays and both $K(\tau)$ and $F(t)$ become small or short-ranged in time, justifying short-memory approximations. By contrast, when the orthogonal dynamics are not dissipative, neglecting the fluctuation term may lead to a large deviation from the true closed dynamics, and analyzing this regime is beyond the scope of the present work.

## B. Problem Setup

In this paper, we focus on modeling deterministic spatiotemporal dynamics governed by (1) via a data-driven approach. Our training dataset consists of several observation trajectories. The $i$-th observation trajectory is obtained by simulating (1) from an initial condition $u_0^{(i)} \in \mathcal{H}$ and evaluating the solution on a time-varying irregular spatial grid $\mathcal{S}_t^{(i)} \subset \Omega$ that is randomly generated ($|\mathcal{S}_t^{(i)}| < \infty$). The data is recorded as $\{u^{(i)}(0)|_{\mathcal{S}_0^{(i)}}, u^{(i)}(\Delta t)|_{\mathcal{S}_{\Delta t}^{(i)}}, \cdots, u^{(i)}(K\Delta t)|_{\mathcal{S}_{K\Delta t}^{(i)}}\}$, where $\Delta t$ is the sampling time interval, $K\Delta t$ is the time horizon used for train, and $u^{(i)}(k\Delta t)|_{\mathcal{S}_{k\Delta t}^{(i)}}$ denotes evaluation of function $u^{(i)}(k\Delta t, \cdot)$ at spatial locations $x \in \mathcal{S}_{k\Delta t}^{(i)}$. Our goal is to capture the long-term evolution of the underlying dynamics over a time horizon $t = K'\Delta t > K\Delta t$. At test time, we are given $N_{\text{test}}$ short-term trajectories, each consisting of $l$ conditioning frames $v(0), \dots, v((l-1)\Delta t)$ observed on random grids $\mathcal{S}'_t$, and we roll out the trained model to produce predictions $v_{\text{pred}}(\ell\Delta t)|_{\Omega_d}, \dots, v_{\text{pred}}(K'\Delta t)|_{\Omega_d}$, where $\Omega_d \subset \Omega$ is a discretization on which we query the predicted states. We report performance on the test dataset over the portions of the time horizon corresponding to the training and test ranges, using

$$\text{loss on training horizon:} \quad \ell_{\text{train-t}} = \frac{1}{N_{\text{test}}} \sum_{j=1}^{N_{\text{test}}} \frac{1}{K - l + 1} \sum_{k=\ell}^{K} \ell\Big(v_{\text{pred}}^{(j)}(k\Delta t)|_{\Omega_d}, v_{\text{true}}^{(j)}(k\Delta t)|_{\Omega_d}\Big),$$

$$\text{loss on test horizon:} \quad \ell_{\text{test-t}} = \frac{1}{N_{\text{test}}} \sum_{j=1}^{N_{\text{test}}} \frac{1}{K' - K} \sum_{k=K+1}^{K'} \ell\Big(v_{\text{pred}}^{(j)}(k\Delta t)|_{\Omega_d}, v_{\text{true}}^{(j)}(k\Delta t)|_{\Omega_d}\Big),$$

where $\ell(\cdot, \cdot)$ denotes the chosen pointwise discrepancy on the query grid $\Omega_d$, and $l < K$ is the number of conditioning steps. Unless stated otherwise, $\ell$ is mean square error (MSE).

## C. Two-phase training methodology and ROM

### C.1. Phase I Training Details

**Training objective.** As mentioned in the main text (Section 4.1), we learn universal observation functionals of the underlying spatiotemporal dynamics, whose evolution is approximately linear in a lifted functional space and that can be

decoded back to the state space $\mathcal{H}$. Accordingly, we minimize the following objective comprising a reconstruction loss, a linear dynamics loss, and a one-step prediction loss (evolve one step in the latent space via the linear propagator, and then decode to the state variable $u(\cdot) \in \mathcal{H}$):

$$\mathcal{L}_{\text{phase1}}(\boldsymbol{\phi}, \boldsymbol{\theta}) = \mathbb{E}\left[\lambda_{\text{recon}}^{(1)} \cdot \underbrace{\|\boldsymbol{\mathcal{D}_\theta}(\boldsymbol{\mathcal{E}_\phi}[\boldsymbol{u}_{\text{delay}}(k\Delta t)]) - u(k\Delta t)\|_2^2}_{\text{reconstruction loss}} + \lambda_{\text{linear}}^{(1)} \cdot \underbrace{\|z_{k+1} - A^* z_k - b^*\|_2^2}_{\text{linear dynamics loss}}\right.$$

$$\left. + \lambda_{\text{pred}}^{(1)} \cdot \underbrace{\|\boldsymbol{\mathcal{D}_\theta}(A^* z_k + b^*) - u((k+1)\Delta t)\|_2^2}_{\text{(linear) prediction loss}}\right], \tag{17}$$

where $\boldsymbol{\phi}, \boldsymbol{\theta}$ are the parameters for the encoder and decoder, $z_k = \boldsymbol{\mathcal{E}_\phi}[\boldsymbol{u}_{\text{delay}}(k\Delta t)] \in \mathbb{R}^D$, $\lambda_{\text{recon}}^{(1)}$, $\lambda_{\text{linear}}^{(1)}$, and $\lambda_{\text{pred}}^{(1)}$ are scalar hyperparameters weighting the reconstruction, linearization, and prediction losses, respectively. Note that $A^*$ and the optional bias $b^*$ is computed offline (i.e., without training) via ridge regression, discussed in the following sections.

**Closed-form ridge estimate of the latent propagator.** For a mini-batch, let

$$Z_0 = \begin{bmatrix} z_k^{(1)} \\ \vdots \\ z_k^{(B)} \end{bmatrix} \in \mathbb{R}^{B \times D}, \quad Z_+ = \begin{bmatrix} z_{k+1}^{(1)} \\ \vdots \\ z_{k+1}^{(B)} \end{bmatrix} \in \mathbb{R}^{B \times D}.$$

We fit a linear one-step model with optional bias,

$$Z_+ \approx Z_0 A^\top + \mathbf{1}\, b^\top \qquad (A \in \mathbb{R}^{D \times D},\ b \in \mathbb{R}^D),$$

by ridge regression:

$$(A^\star, b^\star) = \arg\min_{A,b}\ \|Z_+ - Z_0 A^\top - \mathbf{1}\, b^\top\|_F^2 + \lambda_{\text{ridge}}\big(\|A\|_F^2 + \|b\|_2^2\big). \tag{18}$$

Writing $X = [\, Z_0\ \mathbf{1}\,] \in \mathbb{R}^{B \times (D+1)}$, $Y = Z_+ \in \mathbb{R}^{B \times D}$ and $\Theta = [\, A\ b\,] \in \mathbb{R}^{D \times (D+1)}$, the closed-form solution is given by

$$\Theta^\star = Y^\top X\, (X^\top X + \lambda_{\text{ridge}} I)^{-1}. \tag{19}$$

Unless otherwise stated we set $\lambda_{\text{ridge}} = 5 \times 10^{-3}$ throughout our experiments.

**Streaming EMA of global sufficient statistics.** The per-batch $A_B^\star, b_B^*$ (with subscript $B$ standing for mini-batch) may vary across batches, since batches arrive in a streaming fashion. To maintain a *global* linear propagator representative of the entire training distribution, we keep exponentially moving average (EMA) sufficient statistics

$$S_{xx} \approx \mathbb{E}[X^\top X], \qquad S_{yx} \approx \mathbb{E}[Y^\top X],$$

updated per batch by

$$S_{xx} \leftarrow \beta S_{xx} + (1 - \beta)\, X^\top X, \qquad S_{yx} \leftarrow \beta S_{yx} + (1 - \beta)\, Y^\top X, \tag{20}$$

with $\beta \in (0, 1)$ (we use $\beta = 0.97$). Given $(S_{xx}, S_{yx})$, the current global ridge solution is

$$\Theta_{\text{ema}} = S_{yx}\, (S_{xx} + \lambda_{\text{ridge}} I)^{-1}, \qquad A_{\text{ema}} = \Theta_{\text{ema}}[:, 1{:}D], \quad b_{\text{ema}} = \Theta_{\text{ema}}[:, D{+}1]. \tag{21}$$

We compute $A_{\text{ema}}$ once per epoch either via EMA using (20) or via a full-pass that encodes the entire training set to assemble $(Z_0, Z_+)$ and solves (18) globally. The global estimate $(A_{\text{ema}}, b_{\text{ema}})$ is saved for Phase-II initialization, as an interpretable linear backbone.

## C.2. Phase II Training Details

Although the linear backbone obtained from Phase I already captures the dominant variance of the underlying spatiotemporal evolution, the lack of a closed Koopman-invariant subspace together with the Mori-Zwanzig principle motivate an additional non-Markovian correction to close the near-linear evolution of the learned observables. Here we adopt a discrete formulation of the Mori-Zwanzig formalism, since common spatiotemporal datasets are sampled at fixed regular intervals, and also since we have computed the discrete generator of the Koopman operator during phase-I training; continuous formulation shows similar performances, whereas neural ODE training tends to be more complex and computationally inefficient.

We have already proposed the following two memory correction models in the main text.

$$(\textit{leaky memory}) \; e_t = r \odot \Phi_{\text{dec}}(m_t), \quad \text{where } m_t = \gamma \odot m_{t-1} + \Phi_{\text{enc}}(z_t);$$
$$(\textit{finite memory}) \; e_t = \text{LSTM}(z_t, \cdots, z_{t-1-l_{\text{mem}}}).$$

The first one, leaky memory model, is widely utilized for modeling memory term in Mori-Zwanzig decomposition (Menier et al., 2025), it is based on the exponentially-decaying assumption of the memory kernel and therefore the GLE can be Markovianized by augmenting $z_t$ with an extra memory state $m_t$, forming an augmented ODE. The another model is based on the finite-memory assumption, and previous works adopt diverse sequence models to represent memory corrections, for examples (Wang et al., 2020; Gupta et al., 2025; Fu et al., 2020).

Next, we discuss the training details step by step.

**Setup from Phase I and preprocessing.** Let $(\mathcal{E}_{\phi^*}, \mathcal{D}_{\theta^*})$ denote the learned encoder-decoder in Phase I, and let $A \in \mathbb{R}^{D \times D}, b \in \mathbb{R}^D$ be the corresponding latent linear propagator and bias. We absorb the effect of $b$ by shifting the equilibrium to the origin. Specifically, we compute the fixed point $z^* \in \mathbb{R}^D$ by solving the (regularized) linear system $(I - A)z^* = b$ and cache $z^*$ as the centering vector. By shifting $z \mapsto \tilde{z} \triangleq z - z^*$, the equilibrium of the linear dynamics is moved to the origin. Furthermore, to improve numerical conditioning and balance per-coordinate variance, we diagonally whiten the latent features. Specifically, with centered latents $\tilde{z}$ encoded by $\mathcal{E}_{\phi^*}$ over a held-out stream, we set

$$S \triangleq \text{diag}(s_1, \ldots, s_D), \qquad \text{where } s_j = \sqrt{\mathbb{E}[(\tilde{z}_j)^2]},$$

and define whitened coordinates $w = S^{-1}\tilde{z}$ (with inverse map $\tilde{z} = S\,w$). The corresponding linear propagator in the centered-whitened latent space is given by

$$A_w = S^{-1}A\,S,$$

which is the fixed linear skeleton for the latent process in the $w$-space. This leads to an overall preprocessing chain $z \mapsto \tilde{z} \mapsto w, A \mapsto A_w$. We model the subsequent non-Markovian latent process in $w$-space.

Given a sequence of discretized fields $\{u^{(i)}(k\Delta t, \cdot)|_{\mathcal{S}^{(i)}}\}_{k=0}^K$, we form delay-embedded inputs and obtain the *teacher* latent sequence

$$\alpha_k^* = \mathcal{E}_{\phi^*}[\boldsymbol{u}_{\text{delay}}(k\Delta t)] \in \mathbb{R}^D, \qquad k = l-1, \ldots, K,$$

where $l$ is the number of conditional frames. At inference and for decoding, we always invert the preprocessing in order: $w \mapsto \tilde{z} = Sw \mapsto z = \tilde{z} + z^*$, then pass $z$ to the decoder. For notation convenience, we reserve $z$ for the latent dynamical state, and $A$ for the generator in the normalized (centered–whitened) latent space.

**Non-Markovian latent evolution.** Suggested by the theory, we augment the linear backbone with a memory correction (for concreteness, a leaky-memory instance, see (5)),

$$z_{t+1} = Az_t + \text{res}(z_t, m_t), \qquad m_t = \gamma \odot m_{t-1} + \Phi_{\text{enc}}(z_t),$$

where $m_t$ are the memory states, $\gamma$ is a (learnable) element-wise memory decay, $\Phi_{\text{enc}}$ encodes latent history into memory, and $\text{res}(z_t, m_t)$ is a learned correction decoded from memory, e.g. $\text{res}(z_t, m_t) = r \odot \Phi_{\text{dec}}(m_t)$. During training we freeze $\mathcal{E}_{\phi^*}$ (encoder), train the latent dynamics module (with the linear backbone frozen), and optionally fine-tune $\mathcal{D}_{\theta^*}$ with a small learning rate.

**Training objective.** With teacher latents $\{\alpha_t^*\}$ obtained as above, we minimize

$$\mathcal{L}_{\text{phase2}} = \mathbb{E}\left[\lambda_{\text{dyn}}^{(2)} \|z_t - \alpha_t^*\|_2^2 + \lambda_{\text{pred}}^{(2)} \left\|\mathcal{D}_{\boldsymbol{\theta}}(z_t) - u(t,\cdot)\right\|_2^2 + \lambda_{\text{corr}} \|e_t\|_2^2\right],$$

where $\lambda_{\text{dyn}}^{(2)}, \lambda_{\text{pred}}^{(2)}, \lambda_{\text{corr}} \geq 0$ are loss weights, and $e_t$ regularizes the memory pathway, discouraging overly aggressive non-Markovian updates. Here $z_t$ is produced by the trainable memory-augmented dynamics, while $\alpha_t^*$ is the teacher latent from $\mathcal{E}_{\phi^*}$; the prediction term decodes $z_t$ with the (optionally fine-tuned) decoder $\mathcal{D}_{\boldsymbol{\theta}}$.

**Data pipeline and supervision.** Given streaming field data

$$\left\{ u^{(i)}(0)\big|_{\mathcal{S}_0^{(i)}},\ u^{(i)}(\Delta t)\big|_{\mathcal{S}_{\Delta t}^{(i)}},\ \ldots,\ u^{(i)}(K\Delta t)\big|_{\mathcal{S}_{K\Delta t}^{(i)}} \right\},$$

we first obtain teacher latents $\{\alpha_{l-1}^*, \ldots, \alpha_K^*\}$ via frozen decoder $\mathcal{E}_{\phi^*}$ (with centering/whitening applied only for the internal dynamics). We then train the memory-augmented model with annealed teacher forcing to track these teacher latents (the $\lambda_{\text{dyn}}^{(2)}$ term), while maintaining field-space fidelity through the decoder (the $\lambda_{\text{pred}}^{(2)}$ term). The decoder $\mathcal{D}_{\boldsymbol{\theta}}$ is fine-tuned with a small learning rate; while the encoder remains frozen. This schedule stabilizes optimization and improves long-horizon rollouts.

**Evaluation protocol.** At test time, given new (possibly delay embedded) initial observations $u(0,\cdot)|_{\Omega_d}$, we proceed as follows:

$$\text{encode: } z_0 = \mathcal{E}_{\phi^*}(u(0,\cdot)),$$

$$\text{latent marching: } \begin{cases} z_{t+1} = A z_t + r \odot \Phi_{\text{dec}}(m_t) \\ m_t = \gamma \odot m_{t-1} + \Phi_{\text{enc}}(z_t) \end{cases}, \tag{22}$$

$$\text{decode: } u(t,x) = \mathcal{D}_{\boldsymbol{\theta}^{**}}(z_t)(x),$$

where $\boldsymbol{\theta}^{**}$ denotes the fine-tuned decoder parameters. We report training-horizon (within training window $K$) and test-horizon (rollout) errors by comparing decoded sequences $\{\mathcal{D}_{\boldsymbol{\theta}^{**}}(z_t)\}_t$ to ground truth; in all cases, unwhitening and decentering are applied prior to decoding. Rollouts beyond the training window $t = K$ produce $z_{K+1}, \ldots, z_{K'}$ and their decoded fields for long-term evaluation.

## C.3. Details for Reduced Order Modeling

On top of the pre-trained encoder-decoder, we can train a projection head $U \in \text{St}(d, D)$ (stiefel manifold (Edelman et al., 1998) consisting of matrices satisfying $U^*U = I_d$) to further reduce the dimensionality of the latent space. We choose our projection head lying on the stiefel manifold for the following reasons: (i) $z = [z^{(1)}, z^{(2)}, \cdots, z^{(D)}]$ is pre-trained to span approximately linear invariant subspace of $\mathcal{K}$, thus restricting linearity over the projection head preserves approximate linear invariance property; (ii) among all linear matrices, $U \in \text{St}(d, D)$ admits orthonormality and is norm-preserving, thereby avoiding information loss during projection, while simultaneously reducing redundancy among features and regularizing the parameter space to stabilize training. Once $U$ is learned, we train an another memory-augmented model to capture the evolution of the projected latents $\beta_k = U^\top z_k$ in the reduced space $\mathbb{R}^d$.

The training logic mimics Phase-I by minimizing the following loss:

$$\mathcal{L}_{\text{ROM}}(\boldsymbol{\theta}, U) = \mathbb{E}\left[ \lambda_{\text{recon}}^{(\text{ROM})} \cdot \underbrace{\left\|\mathcal{D}_{\boldsymbol{\theta}}(UU^\top \mathcal{E}_{\phi^*}[\boldsymbol{u}_{\text{delay}}(k\Delta t)]) - u(k\Delta t)\right\|_2^2}_{\text{reconstruction loss}} + \right.$$

$$\left. + \lambda_{\text{linear}}^{(\text{ROM})} \cdot \underbrace{\left\|\beta_{k+1} - \overbrace{\boxed{U^\top A^* U}}^{\text{Markovian generator}} \beta_k\right\|_2^2}_{\text{dynamics loss}} + \lambda_{\text{pred}}^{(\text{ROM})} \cdot \underbrace{\left\|\mathcal{D}_{\boldsymbol{\theta}}(U \cdot \boxed{U^\top A^* U} \beta_k) - u((k+1)\Delta t)\right\|_2^2}_{\text{(linear) prediction loss}} \right], \tag{23}$$

where $\beta_k = U^\top z_k$ are projected observables, $U^\top \circ \mathcal{E}_{\phi^*}, \mathcal{D}_{\boldsymbol{\theta}} \circ U$ are the current encoder and decoder, respectively. Note that suggested by our theory, $U^\top A^* U$ is the generator of Markovian component of the reduced latent dynamics; whereas

the evolution of $\beta_k$ is driven by purely linear dynamics *only if* observables $z$ obtained at phase-I span an exact Koopman invariant subspace and that $U$ spans the eigenspace of $A^*$, otherwise non-Markovian correction term must be incorporated to close the dynamics of $\beta_k = U^\top \circ \mathcal{E}_{\phi^*}[u_{\text{delay}}(k\Delta t)]$. Thus we retrain a dynamics model based on (5) and attain the surrogate dynamics model on $\beta$:

$$\text{encode: } \beta_0 = U^\top \mathcal{E}_{\phi^*}(u(0, \cdot)),$$

$$\text{latent marching: } \begin{cases} \beta_{t+1} = U^\top A^* U \beta_t + r \odot \widetilde{\Phi_{\text{dec}}}(m_t) \\ m_t = \gamma \odot m_{t-1} + \widetilde{\Phi_{\text{enc}}}(\beta_t) \end{cases}, \tag{24}$$

$$\text{decode: } u(t, x) = (\mathcal{D}_{\theta^*} \circ U \beta_t)(x).$$

### C.4. Two-phase training vs. end-to-end joint training

For completeness, we report our experience with fully end-to-end training, where the encoder, decoder, linear backbone, and memory term are all optimized jointly under the same reconstruction and dynamical forecasting losses. In our experiments, this setup consistently underperforms the proposed two-phase strategy, both in terms of interpretability and robustness.

**Loss of approximate linearity in the latent space.** When the linear backbone and memory term are trained jointly from scratch, the model quickly learns to rely on the expressive non-Markovian component (e.g., an LSTM-based memory) to fit the dynamics. In this regime, the autoencoder is no longer encouraged to produce approximately Koopman-invariant observables, and the latent dynamics cease to be "almost linear". As a consequence, the learned spectrum and ROM behavior become much harder to interpret, and the model effectively collapses back to a generic black-box sequence model rather than a Koopman–Mori–Zwanzig-based decomposition. By contrast, Phase I does not merely initialize parameters but actively shapes the representation: the encoder–decoder is trained to behave like a functional Koopman autoencoder before any non-Markovian correction is introduced, and the linear propagator is estimated in closed form (EDMD-style least squares) with an explicit linearity objective in latent space. Without this phase, end-to-end training tends to let the memory component "shortcut" the need for a nearly linear latent representation, leading to degenerate allocations where the linear backbone carries negligible weight.

**Optimization robustness and computational considerations.** In principle, one could try to recover a meaningful linear-plus-memory decomposition in an end-to-end scheme by carefully designing a warm-up schedule for the linear backbone, using different learning rates for linear and memory parameters, and adding explicit regularizers to keep the memory contribution small. However, this introduces several additional hyperparameters and requires substantial tuning. In contrast, the two-phase procedure is simple and robust: Phase II (memory correction) is comparatively cheap to train (on the order of one hour in our experiments), so the overall computational cost is dominated by training the Koopman autoencoder, for which many existing works have already demonstrated good efficiency. In practice, we therefore do not observe a significant computational disadvantage from using two phases, while we gain substantially in interpretability and stability.

## D. Function Encoder and Function Decoder

We discuss in detail of our adopted function encoder and decoder.

### D.1. Function Encoder as Learnable Functionals

**Encoder $\mathcal{D}_\phi$.** We employ Galerkin Transformer (Cao, 2021) as the backbone of our encoder. The encoder first takes discretized (delay-embedded) states $\{u_{\text{delay}}(t, x_i)\}_{i=1}^N$ as the inputs (presumably concatenated with coordinates $\{x_i\}_{i=1}^N$) and maps it into high dimensional latent representation $Y_0 \in \mathbb{R}^{N \times d_1}$ with a simple linear embedding layer, where $\{x_i\}_{i=1}^N$ denote the discretization of the underlying spatial domain $\Omega$. Then $Y_0 \in \mathbb{R}^{N \times d_1}$ is passed to a stack of $L$ identical *self-attention* based encoder layers to produce another element $Y_L \in \mathbb{R}^{N \times d_1}$. Note that the size of $Y_L$ scales with the number of input locations $N$. Different from the common practice as is used in previous work (Cao, 2021; Li et al., 2023b), which models time-forward evolution directly within the space $\mathbb{R}^{N \times d_1}$, we further encode $Y_L$ to get global embedding vector $z_t \in \mathbb{R}^D$, where $D \ll N$, thus achieving tremendous dimension reduction regardless of the resolution of the function discretization.

**Self-Attention Layers.** The update protocol inside each self-attention block is similar to the standard Transformer (Vaswani et al., 2017):

$$Y_{l'} = \text{LayerNorm}(Y_l + \text{Attn}(Y_l)), \qquad Y_{l+1} = \text{LayerNorm}(Y_{l'} + \text{FFN}(Y_{l'})),$$

where $\mathrm{Attn}(\cdot), \mathrm{LayerNorm}(\cdot), \mathrm{FFN}(\cdot)$ stands for attention mechanism, layer normalization (Ba et al., 2016) and feed-forword network, respectively. Unlike traditional NLP tasks, $\mathrm{Attn}(\cdot)$ is instantiated with linear attention $\mathrm{Atten}(\boldsymbol{Y}) = \frac{1}{N}\boldsymbol{Q}\boldsymbol{K}^{\top}\boldsymbol{V}$ first proposed in (Cao, 2021), where each column in the query/key/value matrices $\boldsymbol{Q} = \boldsymbol{Y}\boldsymbol{W}^{\boldsymbol{Q}}, \boldsymbol{K} = \boldsymbol{Y}\boldsymbol{W}^{\boldsymbol{K}}, \boldsymbol{V} = \boldsymbol{Y}\boldsymbol{W}^{\boldsymbol{V}}$ is interpreted as some learnable basis functions $q_j(\cdot), k_j(\cdot), v_j(\cdot)$ evaluated at grid points $\{x_i\}_{i=1}^{N}$, and the calculation of the linear attention can be viewed as the numerical quadrature of different forms of integral transformation of functions:

$$\boldsymbol{Y}_{ij} = \frac{1}{N}\sum_{m=1}^{N}\langle\boldsymbol{Q}[i,:],\boldsymbol{K}[m,:]\rangle_{\mathbb{R}^{d_1}}\cdot\boldsymbol{V}_{mj} \approx \int_{\Omega}\kappa(x_i,\xi)v_j(\xi)\,\mathrm{d}\xi; \qquad \text{(Fourier Type)}$$

$$\boldsymbol{Y}_{ij} = \frac{1}{N}\sum_{s=1}^{d_1}\boldsymbol{Q}[i,s]\cdot\langle\boldsymbol{K}[:,s],\boldsymbol{V}[:,j]\rangle_{\mathbb{R}^N} \approx \sum_{s=1}^{d_1}\left(\int_{\Omega}k_s(\xi)v_j(\xi)\,\mathrm{d}\xi\right)\cdot q_s(x_i). \quad \text{(Galerkin Type)}$$

Thus, by feeding function representation $\boldsymbol{Y}_0 \in \mathbb{R}^{N\times d_1}$ discretized from $\boldsymbol{f}^{(0)} \in \mathcal{H}^{\otimes d_1}$ into $L$ consecutive self-attention layers, we actually transform $\boldsymbol{f}^{(0)} \in \mathcal{H}^{\otimes d_1}$ to $\boldsymbol{f}^{(L)} \in \mathcal{H}^{\otimes d_1}$ with some learnable non-linear integral operators.

**Cross-Attention Layers.** To obtain global embedding vector by aggregating information encoded within $\boldsymbol{Y}_L$, we initialize $K$ [cls] tokens $z^{(0)} \in \mathbb{R}^{K\times d_1}$ with learnable latent coordinates as queries, and perform **_cross-attention_** with $\boldsymbol{Y}_L$. Similar with previous self-attention layers, we compute attention via $\frac{1}{N}\boldsymbol{Q}_z\boldsymbol{K}_{\boldsymbol{Y}}^{\top}\boldsymbol{V}_{\boldsymbol{Y}}$, with subscripts standing for the source for the $\boldsymbol{Q}, \boldsymbol{K}, \boldsymbol{V}$ matrices. Indeed, we can also interpret cross-linear attention as learnable integral transformation:

$$z = \frac{1}{N}\sum_{m=1}^{N}\langle\boldsymbol{Q}_z,\boldsymbol{K}_{\boldsymbol{Y}}[m,:]\rangle_{\mathbb{R}^{d_1}}\cdot\boldsymbol{V}[m,:] = \frac{1}{N}\sum_{m=1}^{N}\langle\boldsymbol{Q}_z,\boldsymbol{k}(x_m)\rangle_{\mathbb{R}^{d_1}}\boldsymbol{v}(x_m) \approx \int_{\Omega}\widetilde{\boldsymbol{\kappa}}(\xi)\boldsymbol{v}(\xi)\,\mathrm{d}\xi,$$

where $\boldsymbol{v}(\cdot) \in \mathcal{H}^{\otimes d_1}$ and $\widetilde{\boldsymbol{\kappa}}(\cdot) \in \mathcal{C}(\Omega, \mathbb{R}^{K\times d_1})$ is a learnable integral kernel mapping $\mathcal{H}^{\otimes d_1}$ to $\mathbb{R}^{K\times d_1}$. After performing $L'$ layers of cross-attention querying from $\boldsymbol{Y}_L \in \mathbb{R}^{N\times d_1}$, we feed the obtained tokens $z^{(L')} \in \mathbb{R}^{K\times d_1}$ to a linear projection layer leading to the final embedded latent representation $z \in \mathbb{R}^D$.

**Learning Observation Functionals.** The overall architecture of our encoder is presented in figure 2. We stress that our encoder realizes transformation between the following _function spaces_ via non-linear integral transformations:

$$\boldsymbol{u}_{\mathrm{delay}}(t,\cdot) \in \mathcal{H}^{\otimes l} \xrightarrow{\text{ linear embedding }} \boldsymbol{f}^{(0)} \in \mathcal{H}^{\otimes d_1} \cdots \xdashrightarrow{\text{ self-attention }} \boldsymbol{f}^{(L)} \in \mathcal{H}^{\otimes d_1}$$

$$z^{(0)} \in \mathbb{R}^{K\times d_1} \cdots \xdashrightarrow[\text{cross-attention}]{} z^{(L')} \in \mathbb{R}^{K\times d_1} \xrightarrow[\text{linear projection}]{} z \in \mathbb{R}^D$$

self-attention layers mimics the roll of feedforward neural networks, while cross-attention layers are infinte-dimensional counterpart for CNN pooling layers, acting on function spaces instead of regular grids, thus inherently being resolution invariant. Combining these building blocks, we actually parametrize a nonlinear functional $\mathcal{H}^{\otimes d_1} \to \mathbb{R}^D$, suggested by the our _functional_ Koopman theory.

### D.2. Function Decoder as Parametrization of Function Spaces

Our function decoder is built upon modulated Fourier network (Yin et al., 2023), however, the adopted modulation in our work differs from (Yin et al., 2023) by imposing non-linearity on the latent modulation vector $z$, benefiting the autoencoder to learn from a more expressive representation space.

Moreover, following similar arguments as in (Fathony et al., 2021), we can show that the output of our decoder can be written as

$$h_j^{(L)}(x) = \sum_r \xi_j^{(r)}(z) \cdot \sin(\beta^{(r)}x + \gamma^{(r)}), \tag{25}$$

a linear combination of $\sin - \cos$ basis functions with learnable frequencies. Attributed to the decomposition (25), we (i) have guarantees w.r.t. the approximation capability of the decoder; (ii) can access the function value at flexible query points (possibly irregular masked inputs, whereas queries on full regular grids); (iii) reach a time-space separable representation of the solution, corresponding to the modeling assumption of separation of variables for spatial-temporal dynamics (Le Dret et al., 2016).

**D.3. Spectral bias and beyond**

**FNO as a Fourier multiplier.**    On a periodic spatial domain $\Omega = \mathbb{T}^d$, FNO parameterizes the time-$\Delta t$ evolution operator on $\mathcal{H}^{\otimes d}$ via $\boldsymbol{v}(x) \longmapsto \sigma\big(W\boldsymbol{v}(x) + (\mathcal{K}_\phi \boldsymbol{v})(x)\big)$, where $(\mathcal{K}_\phi \boldsymbol{v})(x) = \int_{\mathbb{T}^d} \kappa_\phi(x - y)\,\boldsymbol{v}(y)\,\mathrm{d}y$ is a learned integral operator and

$$(\mathcal{K}_\phi \boldsymbol{v})(x) = \mathcal{F}^{-1}\big(R_\phi \cdot (\mathcal{F}\boldsymbol{v})\big)(x) = \sum_{k \in \mathbb{Z}^d} \big(\hat{R}_k \hat{\boldsymbol{v}}_k\big)\, \mathrm{e}^{\mathrm{i}k \cdot x}$$

for the Fourier transform $\mathcal{F}$. Thus FNO reduces to learning a multiplier $R_\phi$ in the frequency domain on a fixed sinusoidal basis $\{\mathrm{e}^{\mathrm{i}k \cdot x}\}$; its spectral block is explicitly diagonal (or block-diagonal) in this basis and hard-wires a Fourier bias into the forward operator.

**MERLIN's Galerkin attention as a Fredholm operator.**    In contrast, MERLIN's function encoder uses linear attention $\mathrm{Attn}(\boldsymbol{Y}) = \frac{1}{N}\,\boldsymbol{Q}\boldsymbol{K}^\top \boldsymbol{V}$, where rows of $\boldsymbol{Q}, \boldsymbol{K}, \boldsymbol{V}$ are associated with spatial locations $\{x_i\}_{i=1}^N$ and channels of $\boldsymbol{v}$. At the level of individual entries,

$$Y_{ij} = \frac{1}{N} \sum_{m=1}^N \langle \boldsymbol{Q}[i,:], \boldsymbol{K}[m,:] \rangle\, \boldsymbol{V}_{mj} \; \approx \; \int_\Omega \kappa(x_i, \xi)\, v_j(\xi)\, \mathrm{d}\xi \; \triangleq \; \widetilde{v}_j(x_i),$$

so self-attention realizes a *Fredholm integral operator* on $\mathcal{H}^{\otimes d}$. Viewed as a Galerkin-type attention, the output can be written as

$$(\mathcal{K}_\theta \boldsymbol{v})(x) \approx \sum_s \Big( \int_\Omega k_s(\xi)\, \boldsymbol{v}(\xi)\, \mathrm{d}\xi \Big)\, q_s(x),$$

where $\{k_s\}$ and $\{q_s\}$ are learned basis functions implicitly encoded by $\boldsymbol{K}$ and $\boldsymbol{Q}$. If we choose $k_s(\xi) = \mathrm{e}^{-\mathrm{i}s \cdot \xi}\hat{R}_s$ and $q_s(x) = \mathrm{e}^{\mathrm{i}s \cdot x}$, this recovers the FNO parameterization. In general, however, MERLIN uses *learned* kernel bases rather than fixed sinusoidal ones. This makes the encoder naturally compatible with irregular, partial observations, but also means it does not automatically inherit the same explicit low-frequency Fourier bias as FNO.

**A simple Fourier-diagonalizable case.**    For illustration, consider the 1D wave equation $u_{tt} = c^2 u_{xx}$ on $[0, 2\pi]$ with periodic boundary conditions. The Fourier modes $\mathrm{e}^{\mathrm{i}kx}$ are eigenfunctions of $\partial_{xx}$ with eigenvalues $-k^2$, so in the Fourier basis the PDE decouples into independent harmonic oscillators, and the time-$\Delta t$ evolution operator is block-diagonal as a Fourier multiplier. FNO directly exploits this by learning multipliers on a few low-frequency modes in a fixed sinusoidal basis. MERLIN's attention-based encoder instead learns a more general integral operator in physical space, without hard-wired sinusoidal eigenfunctions. This yields greater flexibility across irregular grids and sampling patterns, but on smooth periodic benchmarks where the dynamics are *exactly* diagonalizable in the Fourier basis, FNO's explicit spectral bias is particularly advantageous.

**Beyond Fourier-diagonalizable linear PDEs.**    For linear PDEs where the Fourier basis is not well aligned with the dynamics, the infinite matrix representation $A$ of the generator in the Fourier basis is no longer diagonal; Fourier modes (i.e., observables of the form $\langle \mathrm{e}^{\mathrm{i}kx}, u \rangle$) are coupled, and memory emerges once we restrict to a finite set of modes. In such cases, simply parameterizing a matrix multiplier in frequency space may fail to approximate the true propagator. MERLIN's linear-attention encoder, through its learned basis $\{k_s, q_s\}$, can approximate a broader class of non-Fourier observables, while the subsequent memory module compensates for entangled interactions between resolved and unresolved modes. Moving to nonlinear PDEs, one must first invoke functional Koopman theory to obtain a global linearization in an observable space, and then use the operator-theoretic Mori–Zwanzig formalism to describe the induced memory on the resolved observables; MERLIN's linear-backbone-plus-memory design is precisely motivated by this viewpoint.

**Spectral bias from the decoder and multi-scale latent modulation.**    The current decoder employs a single global latent vector to modulate Fourier features at all spatial locations, in a way that is analogous to applying an inverse Fourier transform in a FNO spectral block. While such a Fourier-modulated implicit representation is universal in principle, in practice a single global latent modulation tends to prioritize capturing large-scale dynamical structure and may under-represent highly localized, multi-scale phenomena. A natural extension is to introduce richer latent structure (e.g., hierarchical latents or band-wise latents) that separately modulate different frequency bands or spatial regions; we leave this multi-scale latent modulation as promising future work.

**Injecting explicit Fourier bias into MERLIN.** Motivated by the above theoretical discussion on spectral bias, we hypothesize that injecting an explicit Fourier bias into MERLIN can boost performance on synthetic benchmarks whose spatiotemporal dynamics are (approximately) diagonalizable in a Fourier basis. Concretely, we augment the linear projections that produce the attention tensors $Q, K, V$ with learnable Fourier embeddings of the spatial coordinates (e.g., sinusoidal features). This explicitly exposes Fourier basis functions to the Galerkin Transformer encoder and biases the learned integral kernels toward Fourier-like structure in the underlying Fredholm operator. Experiments with this "+ Fourier embeddings" variant (see Appendix G.6) indeed show improved in-horizon accuracy on synthetic PDEs, while preserving MERLIN's long-horizon stability.

## E. Parameterization of the memory kernel

Our Theorem A.2 motivates correcting the linear backbone with a non-Markovian memory integral. In the linear PDE case study in the main text, we split the dynamics into resolved and unresolved modes and eliminate the unresolved part, obtaining a memory kernel of the form $K(\tau) = A_{12}e^{\tau A_{22}}A_{21}$ and a fluctuation term $F(t) = A_{12}e^{tA_{22}}y(0)$. If $A_{22}$ generates a strongly dissipative or strongly stable semigroup, then $e^{tA_{22}}$ decays and both the memory kernel $K(\tau)$ and the fluctuation term $F(t)$ are short-ranged in time. In this regime, it is natural to approximate the memory by a decaying kernel.

More generally, viewing the Koopman generator $\mathcal{A}$ in a countable matrix representation and partitioning observables into resolved and unresolved components, one arrives at the generalized Langevin equation

$$\frac{\mathrm{d}}{\mathrm{d}t}\mathbf{g}_{\mathcal{M}}(t) = L_{\mathcal{MM}}\mathbf{g}_{\mathcal{M}}(t) + L_{\mathcal{M}\overline{\mathcal{M}}}\int_0^t \mathrm{e}^{(t-s)L_{\overline{\mathcal{MM}}}}L_{\overline{\mathcal{M}}\mathcal{M}}\mathbf{g}_{\mathcal{M}}(s)\,\mathrm{d}s + L_{\mathcal{M}\overline{\mathcal{M}}}\mathrm{e}^{tL_{\overline{\mathcal{MM}}}}\mathbf{g}_{\overline{\mathcal{M}}}(0),$$

where $L_{\mathcal{MM}}$ is the restriction of $\mathcal{A}$ to the resolved subspace, $L_{\overline{\mathcal{MM}}}$ is the generator on the unresolved subspace, and the cross blocks $L_{\mathcal{M}\overline{\mathcal{M}}}$ and $L_{\overline{\mathcal{M}}\mathcal{M}}$ encode the coupling between them. Assuming that the generator for the unresolved part, $L_{\overline{\mathcal{MM}}}$, is (strongly) *dissipative*, we may neglect the fluctuation term and obtain

$$\frac{\mathrm{d}}{\mathrm{d}t}\mathbf{g}_{\mathcal{M}}(t) = L_{\mathcal{MM}}\mathbf{g}_{\mathcal{M}}(t) + L_{\mathcal{M}\overline{\mathcal{M}}}\int_0^t \mathrm{e}^{(t-s)L_{\overline{\mathcal{MM}}}}L_{\overline{\mathcal{M}}\mathcal{M}}\mathbf{g}_{\mathcal{M}}(s)\,\mathrm{d}s. \tag{26}$$

As is common in previous MZ-based works, we approximate the memory operator

$$L_{\mathcal{M}\overline{\mathcal{M}}}\int_0^t \mathrm{e}^{(t-s)L_{\overline{\mathcal{MM}}}}L_{\overline{\mathcal{M}}\mathcal{M}}\mathbf{g}_{\mathcal{M}}(s)\,\mathrm{d}s$$

by a neural parametrization. Specifically, we introduce two MLPs $\Phi_{\mathrm{enc}} : \mathbb{R}^D \to \mathbb{R}^{d_{\mathrm{mem}}}$ and $\Phi_{\mathrm{dec}} : \mathbb{R}^{d_{\mathrm{mem}}} \to \mathbb{R}^D$, and approximate (26) by

$$\frac{\mathrm{d}}{\mathrm{d}t}\mathbf{g}_{\mathcal{M}}(t) = A\,\mathbf{g}_{\mathcal{M}}(t) + \Phi_{\mathrm{dec}}\left(\int_0^t \mathrm{e}^{(t-s)\Lambda_\theta}\Phi_{\mathrm{enc}}(\mathbf{g}_{\mathcal{M}}(s))\,\mathrm{d}s\right),$$

where $\Lambda_\theta \in \mathbb{R}^{d_{\mathrm{mem}}\times d_{\mathrm{mem}}}$ is a diagonal matrix with negative entries, reflecting the strong dissipativity assumption on $L_{\overline{\mathcal{MM}}}$. Denoting

$$\mathbf{m}(t) \triangleq \int_0^t \mathrm{e}^{(t-s)\Lambda_\theta}\Phi_{\mathrm{enc}}(\mathbf{g}_{\mathcal{M}}(s))\,\mathrm{d}s,$$

we obtain an augmented Markovian system

$$\begin{cases} \dfrac{\mathrm{d}}{\mathrm{d}t}\mathbf{g}_{\mathcal{M}}(t) = A\,\mathbf{g}_{\mathcal{M}}(t) + \Phi_{\mathrm{dec}}(\mathbf{m}(t)), \\ \dfrac{\mathrm{d}}{\mathrm{d}t}\mathbf{m}(t) = \Lambda_\theta\,\mathbf{m}(t) + \Phi_{\mathrm{enc}}(\mathbf{g}_{\mathcal{M}}(t)). \end{cases}$$

In this paper, we adopt a discrete-time representation of this memory model, which leads to the **leaky memory model (LMM)** in the main text: a lightweight parametric approximation of an approximately exponentially decaying convolution kernel.

However, when the unresolved dynamics are not effectively dissipative (for instance, when they contain slow or strongly chaotic components), the true memory kernel can be long-ranged or highly structured, and a simple exponentially decaying

ansatz may be too restrictive. This motivates the **finite-memory model (FMM)**: an LSTM-based sequential module that directly processes a finite history window in latent space. FMM assumes only a finite memory horizon, but within that window it is strictly more expressive than LMM; one can always increase the history length to capture longer effective memory, at the cost of additional parameters and a more sensitive hyperparameter (the window length). In discrete time, the model writes

$$\mathbf{g}_{\mathcal{M}}(t+1) = A\,\mathbf{g}_{\mathcal{M}}(t) + \mathrm{LSTM}\big(\mathbf{g}_{\mathcal{M}}(t), \ldots, \mathbf{g}_{\mathcal{M}}(t-1-l_{\mathrm{mem}})\big),$$

where $l_{\mathrm{mem}}$ is the hyperparameter controling the memory length.

## F. Datasets

**Wave equation.** We consider the two-dimensional linear wave equation

$$\partial_t^2 u = c\,\Delta u,$$

where $u$ is the displacement field and $c > 0$ is the wave speed. We consider the first-order formulation in the variables $(u, v)$ with $v = \partial_t u$: $\partial_t u = v$, $\partial_t v = c\,\Delta u$. We set $c = 1/16$ and impose periodic boundary conditions. The domain is $\Omega = [-1, 1] \times [-1, 1]$, discretized on a uniform $64 \times 64$ grid. The initial displacement $u(\cdot, 0)$ is an isotropic Gaussian bump with random radius $r \sim \mathrm{Unif}[0.25, 0.30]$ and amplitude $A \sim \mathrm{Unif}[2.0, 4.0]$ (here $\mathrm{Unif}[a, b]$ denotes the continuous uniform distribution on $[a, b]$). Concretely,

$$u(x_1, x_2, 0) \;=\; A \exp\left(-\frac{x_1^2 + x_2^2}{2r^2}\right),$$

followed by an independent random circular (periodic) shift along each axis to avoid grid alignment. The initial velocity is set to zero, $v(\cdot, 0) \equiv 0$. Time integration uses a fixed internal solver step of $10^{-3}$ for stability, and we record fields every $\Delta t = 0.25$ over a horizon of $40.0$, yielding $T = 160$ equally spaced frames per trajectory (adjacent frames are separated by $\Delta t = 0.25$).

With these settings we generate $N = 500$ independent trajectories; $400$ are randomly assigned to the training set and $100$ to the test set. For the training data, from each length-160 trajectory we uniformly sample 10 starting indices and extract the subsequent 20 frames; within each 20-frame segment, the first 10 frames are used as the training horizon and the remaining 10 frames are reserved as the test horizon, with the latter used only for extrapolation evaluation. Our data generation protocol follows Yin et al. (2023).

**Navier-Stokes equations.** We consider the two-dimensional incompressible Navier–Stokes equation in vorticity–velocity form,

$$\partial_t w(x, t) + u(x, t) \cdot \nabla w(x, t) = \nu\,\Delta w(x, t) + f(x), \qquad \nabla \cdot u(x, t) = 0, \qquad w(x, 0) = w_0(x),$$

where $w(x, t)$ denotes the scalar vorticity (the state variable used for learning) and $u(x, t)$ is the divergence-free velocity field. We consider the periodic square domain $\Omega = [-1, 1] \times [-1, 1]$ with viscosity $\nu = 1 \times 10^{-3}$. The deterministic body force is

$$\forall\, x = (x_1, x_2) \in \Omega, \qquad f(x) = 0.1\Big( \sin\big(2\pi(x_1 + x_2)\big) + \cos\big(2\pi(x_1 + x_2)\big)\Big).$$

The flow is discretized on a uniform $64 \times 64$ grid. The initial vorticity is sampled from a zero-mean Gaussian random field on the periodic square domain $\Omega$. Specifically,

$$\omega_0(\cdot) \sim \mathrm{GRF}\Big(0,\; 7^{3/2}\,(-\Delta + 49\,I)^{-2.5}\Big),$$

where $\Delta$ denotes the Laplacian on $\Omega$ with periodic boundary conditions and $I$ is the identity. Time integration uses a fixed recording cadence: fields are saved every $\Delta t = 1$ units of simulated time, and each trajectory contains $T = 50$ frames.

Using this configuration, we generate $N = 1000$ independent trajectories. We randomly assign 800 trajectories to the training set and 200 to the test set. For the training data, from each length-50 trajectory we uniformly sample 2 starting indices and extract the subsequent 20 frames; within each 20-frame segment, the first 10 frames are used as the training horizon and the remaining 10 frames are reserved as the test horizon, with the latter used only during extrapolation evaluation. Our data generation protocol follows (Li et al., 2021).

**Kuramoto–Sivashinsky.** We consider the one-dimensional Kuramoto–Sivashinsky equation with periodic boundary conditions,

$$\partial_t u(x,t) = -u(x,t)\,\partial_x u(x,t) - \partial_{xx} u(x,t) - \partial_{xxxx} u(x,t), \qquad x \in [0,L], \qquad u(x,0) = u_0(x),$$

where $u(x,t)$ is the scalar state variable used for learning. We take the periodic domain length $L = 64$ and discretize space on a uniform grid with $N_x = 128$ points. Spatial derivatives are evaluated by a Fourier pseudospectral method (implemented via `psdiff`). Time integration is performed with the stiff solver Radau (`solve_ivp`) using tolerances `atol = rtol = 10^{-6}`. Initial conditions are sampled from randomized low-frequency Fourier series. Specifically, we draw

$$N = 10, \qquad A_k \sim \mathrm{Unif}([-0.5, 0.5]), \qquad \phi_k \sim \mathrm{Unif}([0, 2\pi)), \qquad l_k \sim \mathrm{Unif}\{1, 2\},$$

and set

$$u_0(x) = \sum_{k=1}^{N} A_k \sin\left( \frac{2\pi l_k}{L} x + \phi_k \right).$$

Each trajectory is simulated over $t \in [0, T_{\mathrm{final}}]$ with $T_{\mathrm{final}} = 100$, and we record $T = 200$ frames (i.e., saving every $\Delta t = T_{\mathrm{final}}/(T-1) \approx 0.5$).

Using this configuration, we generate $N = 500$ independent trajectories. We randomly assign 800 trajectories to the training set and 200 to the test set. For the training data, from each length-200 trajectory we uniformly sample 20 starting indices and extract the subsequent 20 frames; within each 20-frame segment, the first 10 frames are used as the training horizon and the remaining 10 frames are reserved as the test horizon, with the latter used only during extrapolation evaluation. Our data generation protocol follows (Brandstetter et al., 2022).

**Sea surface temperature.** We use daily fields from the CMEMS Global Ocean Physics Reanalysis at native horizontal resolution $1/12°$ (https://data.marine.copernicus.eu/product/GLOBAL_ANALYSISFORECAST_PHY_001_024/services). Following (Yin et al., 2023; De Bézenac et al., 2019), we employ a fixed rectangular oceanic bounding box and a tiling protocol similar to that reference: within this region we extract four non-overlapping $64 \times 64$ tiles aligned to the native grid. The exact coordinates and grid indices—together do not affect our results.

With these settings, we construct $N = 4 \times 250 = 1000$ sequences by sampling, for each tile, 250 start days $t_0$ uniformly at random in 1 Jan 2022–30 Sep 2025 such that a contiguous 20-day segment starting at $t_0$ lies entirely within the record; within each 20-day segment, the first 10 days are used as the training horizon and the next 10 days are reserved as the test horizon, used only for extrapolation evaluation. We randomly split the 1000 sequences into 800 for training and 200 for testing. The exact preprocessing code, tile definitions, and the specific data used will be provided with our code.

**ERA5.** We consider the ERA5 global atmospheric reanalysis produced by the European Centre for Medium-Range Weather Forecasts (ECMWF (Hersbach et al., 2020)). ERA5 provides hourly estimates of the state of the atmosphere on a $0.25°$ longitude–latitude grid (about $720 \times 1440$ points globally) from 1979 to the present, which are obtained via state-of-the-art data assimilation that combines heterogeneous observations with a numerical weather prediction model.

We retain only a single temperature variable and downsample each global field to a $180 \times 360$ grid to reduce memory and computational costs. With these settings, we construct $N = 4 \times 125 = 500$ sequences by sampling 125 start days $t_0$ in each year from 2011 to 2014, chosen uniformly at random, such that a contiguous 20-day segment starting at $t_0$ lies entirely within the record. Within each 20-day segment, the first 15 days are used as the training horizon and the next 5 days are reserved as the test horizon, used only for extrapolation evaluation. We also construct 50 sequences from 2015 for testing. The exact preprocessing code and specific data used will be provided with our code. Our data generation protocol follows (Ren et al., 2025; Song et al., 2024).

## G. Experimental details and additional results

### G.1. Implementation of Baselines

We rely on public implementations for all baselines and, whenever possible, follow the recommended hyperparameters from the corresponding repositories: • **FNO, GKT, U-Net, UNO.** implemented via the unified Neural-Solver-Library (https://github.com/thuml/Neural-Solver-Library); we use the default or recommended settings (network depth/width, learning rate schedules, etc.), and only adapt dataset-specific options such as input/output

*Table 3.* **Flexibility with masked observations.** Mean-squared error (MSE) on Wave and Navier–Stokes under different spatial mask ratios.

| Model | Wave | | Navier–Stokes | |
|---|---|---|---|---|
| | Training horizon | Test horizon | Training horizon | Test horizon |
| Mask ratio = 25% | | | | |
| SIREN | 1.691e-2 | 2.824e-2 | 2.465e-1 | 3.671e-1 |
| DINo | **4.519e-4** | **1.487e-3** | **1.205e-3** | **5.359e-3** |
| DeepONet | 1.723e-2 | 3.305e-2 | 9.615e-3 | 1.259e-2 |
| **MERLIN** | **3.755e-4** | **6.547e-4** | **5.283e-4** | **4.249e-3** |
| Mask ratio = 50% | | | | |
| SIREN | 2.963e-2 | 5.412e-2 | 2.571e-1 | 4.819e-1 |
| DINo | **4.679e-4** | **1.467e-3** | **1.124e-3** | **5.187e-3** |
| DeepONet | 1.734e-2 | 3.461e-2 | 9.241e-3 | 1.296e-2 |
| **MERLIN** | **5.734e-4** | **9.407e-4** | **7.172e-4** | **5.786e-3** |
| Mask ratio = 75% | | | | |
| SIREN | 3.132e-2 | 6.974e-2 | 3.223e-1 | 6.435e-1 |
| DINo | **5.237e-4** | **1.372e-3** | **1.187e-3** | **5.448e-3** |
| DeepONet | 1.748e-2 | 3.124e-2 | 9.376e-3 | 1.384e-2 |
| **MERLIN** | **8.489e-4** | **1.225e-3** | **7.988e-4** | **5.339e-3** |
| Mask ratio = 95% | | | | |
| SIREN | 3.339e-2 | 7.894e-2 | 3.656e-1 | 9.027e-1 |
| DINo | 2.056e-2 | **3.576e-2** | **3.469e-3** | **1.235e-2** |
| DeepONet | **1.692e-2** | 3.625e-2 | 1.306e-2 | 1.425e-2 |
| **MERLIN** | **2.908e-3** | **3.928e-3** | **2.208e-3** | **1.021e-2** |

horizon and batch size to match our forecasting protocol. • **KNO.** implemented following the KoopmanLab codebase (https://github.com/Koopman-Laboratory/KoopmanLab), using the authors' default architecture and training configuration, with minimal changes to accommodate our grids and rollout horizons. • **DeepONet.** implemented in an autoregressive forecasting setting; both the branch and trunk networks are realized as fully connected MLPs with the same depth and a relatively large hidden width, using ReLU activations throughout. The overall configuration follows common practice in recent DeepONet applications, with minor tuning on the validation set for each dataset. • **DINo.** implemented with the public code from https://github.com/mkirchmeyer/DINo; we preserve the latent Neural ODE architecture but train for 5000 epochs, as the default number of epochs in the repository is overly aggressive and computationally demanding in our setting. • **SIREN.** based on the official implementation (https://github.com/vsitzmann/siren), adapted to parameterize the spatiotemporal solution field $u(t, x)$ in our PDE forecasting tasks; the core sinusoidal MLP architecture and initialization follow the original work, and we modify only the input/output heads to match our train/test splits and rollout horizons.

## G.2. MELIN hyperparameters

We summarize key hyperparameters of MERLIN for each dataset in Table 4. Further sensitivity analyses are provided in Appendix I.2.

## G.3. Results on masked-observation experiments

The results of the masked-observation experiments are summarized in Table 3.

## G.4. Analysis of computational efficiency

To quantify the computational efficiency of our model, we report the wall-clock inference time per 100 samples and the number of trainable parameters on the wave dataset, where long-horizon rollouts are feasible. Results are summarized in Table 5.

Despite using a Transformer-based encoder, MERLIN is significantly faster at inference than FNO (about $2.6\times$) and much faster than U-Net, while also using fewer parameters. The key reason is that MERLIN performs time marching entirely in

*Table 4.* Hyperparameters for MERLIN on each dataset.

| Hyperparameter | Navier–Stokes | Wave | SST |
|---|---|---|---|
| Function Encoder $\mathcal{E}_\phi$ | | | |
| Number of self-attention layers | 3 | 4 | 3 |
| Number of cross-attention layers | 2 | 2 | 2 |
| Hidden dimensions | 128 | 128 | 128 |
| Dimension of each token | 32 | 16 | 64 |
| Number of tokens $K$ | 4 | 4 | 4 |
| Latent dimension | 128 | 64 | 256 |
| Function Decoder $\mathcal{D}_\theta$ | | | |
| Fourier hidden dimensions | 128 | 128 | 192 |
| Number of Fourier Layers | 4 | 3 | 4 |
| Frequency scale factor | 64 | 64 | 128 |
| Memory-Augmented Dynamics model (Finite Memory Model) | | | |
| Memory length $l_{\mathrm{mem}}$ | 4 | 4 | 4 |
| LSTM layers | 2 | 3 | 3 |
| LSTM hidden dimension | 256 | 256 | 1024 |
| Optimization | | | |
| Phase I learning rate | $10^{-3}$ | $10^{-3}$ | $10^{-3}$ |
| Phase II learning rate | $5 \times 10^{-4}$ | $5 \times 10^{-4}$ | $5 \times 10^{-4}$ |
| Number of epochs (Phase I) | 800 | 800 | 800 |
| Number of epochs (Phase II) | 500 | 500 | 500 |

*Table 5.* Wave dataset: wall-clock runtime per 100 samples (seconds) and number of trainable parameters.

| | FNO | MERLIN | U-Net |
|---|---|---|---|
| Runtime per 100 samples (s) | 4.282 | **1.640** | 21.403 |
| # model parameters | 8,418,946 | **3,372,277** | 50,801,430 |

the low-dimensional latent space: once the initial observations are encoded, the linear-plus-memory latent dynamics evolve a small latent state, and the decoder is queried only at desired times/locations. In contrast, FNO rolls out the full spatial field at every time step on the physical grid. We expect this latent-space marching to become even more advantageous for longer rollout horizons and higher spatial resolutions.

Regarding encoder complexity, our Galerkin Transformer uses a linear-attention variant $\mathrm{Atten}(\boldsymbol{Q}, \boldsymbol{K}, \boldsymbol{V}) = \frac{1}{N} \boldsymbol{Q} \boldsymbol{K}^\top \boldsymbol{V}$ instead of standard softmax attention $\mathrm{Atten}(\boldsymbol{Q}, \boldsymbol{K}, \boldsymbol{V}) = \mathrm{softmax}(\boldsymbol{Q} \boldsymbol{K}^\top) \boldsymbol{V}$, where $\boldsymbol{Q}, \boldsymbol{K}, \boldsymbol{V} \in \mathbb{R}^{N \times d}$ and $N$ is the number of sensor points. For standard attention, the dominant cost is forming the $N \times N$ matrix $\boldsymbol{Q} \boldsymbol{K}^\top$, which scales as $O(N^2 d)$, whereas our implementation scales essentially as $O(Nd)$ (up to factors in the embedding dimension), i.e., linear in $N$ rather than quadratic. In addition, the encoder can ingest a (potentially very sparse) set of observed sensor locations directly, without requiring access to the full spatial grid. Further reducing encoder cost via more specialized efficient Transformer architectures for high-dimensional PDEs (e.g., FactFormer (Li et al., 2023c)) is an interesting direction for future work, but orthogonal to the main contribution of this work.

Finally, the two-phase training scheme is computationally efficient in practice: Phase II (memory correction) is relatively cheap, so the overall training time is dominated by learning the Koopman autoencoder (Phase I). Empirically, we do not observe a significant overhead compared to a single-stage training scheme with an equally expressive recurrent core.

### G.5. Quality of latent representation

Beyond low rollout error, an important question is what kind of representation MERLIN learns in its latent space and to what extent this representation retains physically meaningful information. We therefore perform a suite of qualitative and quantitative diagnostics that explicitly probe the quality of the learned latent variables.

First, we visualize latent trajectories and their spectra (Fig. 4 and Appendix Fig. 16), revealing a clear "linear backbone + memory correction" structure: a small number of dominant modes evolve approximately linearly, while the learned memory term provides low-rank corrections that account for non-Markovian effects. Second, we inspect the reduced-order modes and their temporal evolution obtained from MERLIN's ROM head (Fig. 5). These modes exhibit interpretable temporal evolution (e.g., oscillations in the wave benchmark), indicating that the latent coordinates align with coherent structures of the underlying PDE. Third, we evaluate MERLIN on masking and spatial extrapolation tasks, where the model must reconstruct the full field from sparse and irregular sensor observations (Appendix Table 3). Accurate reconstruction under strong subsampling is only possible if the latent representation captures the global dynamics rather than overfitting to a particular sensor layout. These results are already discussed in the main text.

Finally, we design two *additional* explicit **downstream tasks** that directly test how much information about the physical state is linearly or nonlinearly decodable from the latent variables. The first is an inverse problem of recovering the initial condition from its subsequent evolution, formulated and solved in latent space using a Tikhonov-type optimization over the latent initial condition. The second is a linear probing experiment, where we freeze the trained MERLIN model and fit a small linear head on top of the latent features to regress global physical observables such as enstrophy and spatially averaged displacement. In both experiments MERLIN achieves low reconstruction errors and near-perfect linear decodability of these observables, providing strong evidence that the latent space learned by MERLIN forms a well-structured, dynamically meaningful representation of the underlying PDE.

**Inverse problem on inferring the initial condition.**    We further consider the following inverse task to assess the quality of the learned representation beyond forward temporal prediction: recovering an initial condition from its subsequent evolution. Let $(\Phi_t)_{t \geq 0}$ denote the PDE *flow* on the state space $\mathcal{H}$ and $\mathcal{R} : \mathcal{H}^{\otimes T} \to \mathcal{Y}$ an observation operator, where $\mathcal{H}^{\otimes T}$ denotes the solution space over the time indices $0, \ldots, T-1$. Given a single observed trajectory $\{y_k\}_{k=0}^{K}$ with $y_k \approx \mathcal{R}(\Phi_{t_k}(u_0))$, the classical inverse problem "recover the initial state $u_0$" is in general ill-posed and is typically posed as a Tikhonov variational problem

$$\hat{u}_0 \in \arg\min_{u_0 \in \mathcal{H}} \sum_{k=0}^{K} \left\| \mathcal{R}(\Phi_{t_k}(u_0)) - y_k \right\|_{\mathcal{Y}}^2 + \lambda\,\Theta(u_0),$$

where $\Theta$ encodes a prior or regularization on the initial field.

In MERLIN, we surrogate the dynamical flow $\Phi_t$ by the learned encoder–latent–dynamics–decoder model and, crucially, solve this inverse problem in latent space rather than in the high–dimensional physical space. Let $\mathcal{E} : \mathcal{H}^{\otimes l} \to \mathcal{Z}$ be the encoder (where $\mathcal{Z} \subset \mathbb{R}^D$ denotes the latent manifold), $\Psi_{t_k} : \mathcal{Z} \to \mathcal{Z}$ the memory–corrected latent dynamics, and $\mathcal{D} : \mathcal{Z} \to \mathcal{H}$ the decoder. Given a candidate latent initial condition $z_0 \in \mathcal{Z}$, we generate predicted observations

$$\hat{y}_k(z_0) := \mathcal{R}\big(\mathcal{D}(\Psi_{t_k}(z_0))\big), \qquad k = 0, \ldots, K,$$

and solve

$$\hat{z}_0 \in \arg\min_{z_0 \in \mathcal{Z}} \sum_{k=0}^{K} \left\| \mathcal{R}\big(\mathcal{D}(\Psi_{t_k}(z_0))\big) - y_k \right\|_{Y}^2 + \lambda\,\Theta_{\mathcal{Z}}(z_0),$$

with all network parameters frozen and only $z_0$ optimized by gradient descent. The reconstructed initial field is then $\hat{u}_0 := \mathcal{D}(\hat{z}_0)$. This latent–space Tikhonov formulation uses the fixed decoder $\mathcal{D}$ as an implicit structural prior, constraining $\hat{u}_0$ to lie on the learned low–dimensional manifold $\{\mathcal{D}(z) : z \in \mathcal{Z}\}$ rather than in the full state space $\mathcal{H}$. Illustrative results for Navier–Stokes and 2D wave are shown in Fig. 6, where MERLIN achieves small error between $\hat{u}_0$ and the ground–truth initial field, indicating that the latent representation and its linear–plus–memory latent dynamics retain sufficient information to support nontrivial inverse problems beyond forward temporal prediction.

**Linear probing of latent representations.**    To further assess the quality of the latent representations learned by MERLIN, we perform a simple linear probing experiment. We freeze the encoder, latent dynamics, and decoder of a trained model, and

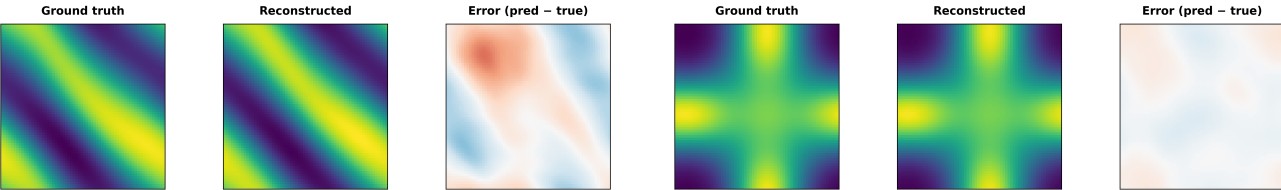

*Figure 6.* Inverse initial–condition experiment used to probe representation quality beyond forward prediction. Left: Navier–Stokes; right: 2D wave. For each case we show the ground–truth initial field, the reconstruction obtained by optimizing the latent initial condition $z_0$, and the corresponding error field.

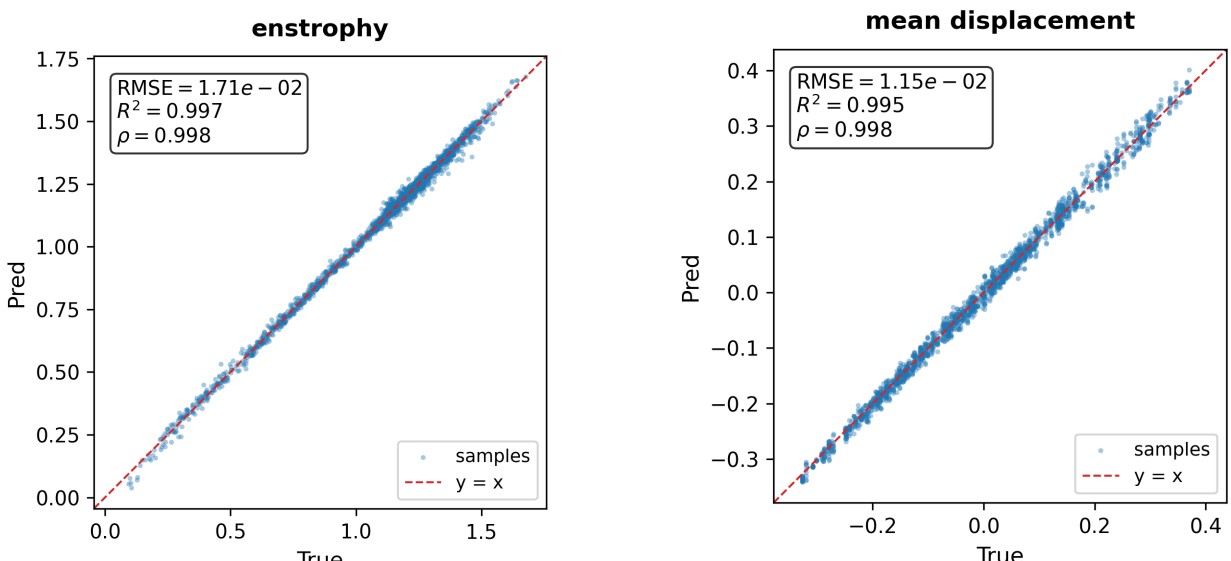

*Figure 7.* Linear probing of MERLIN latent representations on Navier–Stokes (left) and the 2D wave equation (right). The dashed line indicates the identity $y = x$. Insets report RMSE, $R^2$, and Pearson correlation $\rho$. The tight concentration around the diagonal shows that global physical observables are almost linearly decodable from the latent state.

extract the latent sequence $\{z_t\}$ along trajectories. On top of these frozen latents we train a small linear head $y_t = W z_t + b$ with an $\ell_2$ loss to regress physically meaningful observables.

For the Navier–Stokes dataset we consider the enstrophy $E(t) = \int_\Omega \omega(t, x)^2 \, \mathrm{d}x$ computed from the vorticity field. For the 2D wave equation we focus on the spatially averaged displacement $\bar{u}(t) = |\Omega|^{-1} \int_\Omega u(t, x) \, \mathrm{d}x$. Importantly, only the linear readout is trained in this experiment, while the underlying latent dynamics are exactly those used for forward prediction.

Figure 7 shows scatter plots of the true versus predicted observables, together with RMSE, $R^2$, and Pearson correlation $\rho$. In both datasets the points concentrate tightly around the identity line and the coefficients of determination are close to 1, indicating that physically interpretable quantities are almost linearly decodable from the latent state. This provides additional evidence that MERLIN learns a well-structured latent space that aligns with the underlying PDE dynamics.

### G.6. Mitigating spectral bias via Fourier embeddings

We evaluate the Fourier-augmented design of the function encoder ("MERLIN + Fourier embeddings"), proposed in Appendix D.3, on the wave and Navier–Stokes benchmarks and compare against the original MERLIN and an FNO baseline. As summarized in Table 6, adding Fourier embeddings consistently reduces the training-horizon error on synthetic PDEs while preserving MERLIN's long-horizon stability, and yields performance that is competitive with FNO.

*Table 6.* Effect of adding Fourier embeddings to MERLIN on synthetic PDE benchmarks.

| Dataset | Model design | Loss (training horizon) | Loss (test horizon) |
|---|---|---|---|
| Wave | MERLIN (original) | 6.194e-5 | 1.659e-4 |
| | MERLIN + Fourier embeddings | 2.342e-5 | 4.802e-5 |
| | FNO | 1.012e-5 | 5.426e-5 |
| Navier–Stokes | MERLIN (original) | 4.590e-4 | 2.035e-3 |
| | MERLIN + Fourier embeddings | 4.867e-5 | 2.635e-4 |
| | FNO | 2.797e-5 | 3.967e-4 |

### G.7. Additional chaotic PDE benchmark: Kuramoto–Sivashinsky

We additionally evaluate MERLIN on the Kuramoto–Sivashinsky (KS) equation, a canonical chaotic PDE, to further demonstrate its superiority over FNO in terms of temporal extrapolation. Table 7 reports mean-squared error (MSE) on both the training and test horizons, comparing MERLIN against FNO.

*Table 7.* Performance on the Kuramoto–Sivashinsky (KS) benchmark.

| Metric | FNO | MERLIN |
|---|---|---|
| Training horizon MSE | 2.609e-3 | 3.906e-3 |
| Test horizon MSE | 1.405e-1 | 4.393e-2 |

On this strongly chaotic system, FNO attains a slightly lower MSE on the training horizon, but its test error grows by roughly $54\times$ from train to test, whereas MERLIN's test error is about $3.2\times$ lower than FNO's and its train-to-test ratio is only about $11\times$. This pattern mirrors our observations on the wave and Navier–Stokes systems and further indicates that MERLIN provides substantially more robust long-horizon generalization on a chaotic PDE.

This behaviour is consistent with prior work showing that chaotic dynamics can often be decomposed into a dominant linear backbone plus a low-energy forcing term. In particular, the HAVOK analysis of Brunton et al. (2017) demonstrates that a broad class of chaotic finite-dimensional systems (including the Lorenz attractor and several real-world datasets) can be represented as a linear dynamical system in delay coordinates driven by an intermittent forcing signal. Their framework combines delay embedding with Koopman theory and decomposes chaos into (i) a nearly autonomous linear evolution on leading delay coordinates and (ii) a forcing signal carried by remaining coordinates, whose heavy-tailed statistics correspond to intermittent switching and bursting events.

Our framework is conceptually aligned with this viewpoint. In Phase I, we learn functional Koopman observables and a linearly evolving latent backbone (analogous to the leading delay coordinates in HAVOK). In Phase II, we add a data-driven memory/forcing term in latent space, which plays a similar role to the HAVOK forcing, capturing the influence of unresolved modes and strongly nonlinear episodes. The key difference is that we work directly with infinite-dimensional, spatiotemporal PDE states, and we learn the memory term as a parametric approximation of a Mori–Zwanzig-type memory kernel, which is not present in Brunton et al. (2017).

### G.8. Additional real-world benchmark: ERA5 temperature forecasting

We further stress-test MERLIN on a real-world climate benchmark: forecasting ERA5 near-surface temperature fields. We compare against FNO under the same training and evaluation protocol, and report mean-squared error (MSE) on held-out test segments from the year 2015 in Table 8.

*Table 8.* Performance on ERA5 temperature prediction task.

| Metric | FNO | MERLIN |
|---|---|---|
| Training horizon MSE | 5.907e-2 | **1.223e-2** |
| Test horizon MSE | 1.563e-1 | **1.565e-2** |

MERLIN attains lower error than FNO on both the training and test horizons, and the degradation from train to test is much smaller, indicating markedly improved temporal extrapolation on this real-world dataset. To complement the quantitative results, Figure 15 shows a qualitative comparison along a representative test trajectory.

## H. Visualizations and Qualitative Experiment results

In this section, we illustrate several prediction results among datasets adopted in our work. Figure 8 visualizes long term prediction for Navier-Stokes; Figure 9 and 10 visualize long term prediction for Wave; Figure 11 visualizes long term prediction for SST; Figure 12 and 13 visualize long term prediction for randomly subsampled observation datas of Navier-Stokes and Wave, respectively; Figure 14 shows reduced-order-modeling results for Navier-Stokes; Figure 15 visualizes ERA5 temporal prediction results over six consecutive days in 2015 and finally, Figure 16 displays detailed information of the memory-augmented model, trained on wave dataset.

# I. Ablation Study and Hyperparameter Sensitivity

## I.1. Ablation Study

**Memory module: leaky memory model (LMM) vs. finite memory model (FMM).** Beyond the theoretical insights discussed in Section E, we empirically compare the two memory parameterizations on synthetic PDE benchmarks (Navier–Stokes, wave equation, and Kuramoto–Sivashinsky).

*Table 9.* Additional comparison between leaky memory model (LMM) and finite memory model (FMM) on synthetic benchmarks.

| Dataset | Model | Loss (training horizon) | Loss (test horizon) |
|---------|-------|-------------------------|---------------------|
| NS | Linear + FMM | 4.590e-4 | 2.035e-3 |
|    | Linear + LMM | 4.947e-4 | 2.518e-3 |
| Wave | Linear + FMM | 6.194e-5 | 1.659e-4 |
|      | Linear + LMM | 6.403e-5 | 2.074e-4 |
| KS | Linear + FMM | 3.906e-3 | 4.393e-2 |
|    | Linear + LMM | 2.579e-2 | 4.529e-1 |

Empirically, these results align with our theoretical interpretation: on the simpler synthetic benchmarks (see also Section I.2), LMM already achieves competitive fit and generalization, typically with fewer parameters, and FMM provides only modest gains. In contrast, on the strongly chaotic Kuramoto–Sivashinsky experiment, FMM is clearly preferable: LMM struggles to maintain long-horizon generalization, whereas FMM can exploit a richer, longer-range history to capture strong nonlinear interactions between resolved and unresolved modes. This suggests the following practical guideline: for moderately mixing or effectively dissipative PDEs where one expects short-range memory, LMM is a good default due to its simplicity and stability; for strongly chaotic or weakly damped regimes, FMM is better suited.

**Linear projection head: with/without Stiefel manifold constraint.** In Phase-II, given a fixed Phase-I latent space of dimension $D$, the reduced-order model (ROM) is obtained by projecting the latent dynamics onto a $d$-dimensional subspace via a projection head $U \in \mathbb{R}^{D \times d}$. Our default choice is to constrain $U$ to lie on the Stiefel manifold $\mathrm{St}(D, d)$ (columns orthonormal). This design is motivated by both theoretical and empirical considerations.

From a theoretical viewpoint, the ROM only depends on the subspace $U^\top \mathcal{M} = \mathrm{span}(U^\top \mathbf{g}) \cong \mathbb{R}^d$, not on the particular basis used. Any full-rank linear map $W \in \mathbb{R}^{D \times d}$ can be factorized as $W = UR$ with $U \in \mathrm{St}(D, d)$ and an invertible matrix $R \in \mathbb{R}^{d \times d}$. Replacing $W$ by $U$ therefore amounts to *a change of coordinates* in the reduced space (from $y = W^\top z$ to $y' = R^\top y$), without altering the underlying $d$-dimensional subspace on which the projected dynamics live. In this sense, constraining $U$ to the Stiefel manifold does not reduce the expressive power of the ROM; it simply selects an orthonormal basis for the same subspace, and makes the projected operator $A_{\mathrm{eff}} = U^\top A U$ a standard Galerkin projection.

Empirically, we compared a Stiefel-constrained projector to an unconstrained linear projector on the wave-ROM experiment with reduced dimension $d = 32$, training both variants for 100 epochs under identical hyperparameters (we choose $\lambda_{\mathrm{recon}}^{(\mathrm{ROM})} = 1$ and $\lambda_{\mathrm{linear}}^{(\mathrm{ROM})} = 5 \times 10^{-2}$ in this ROM experiment). As summarized in Table 10 and Fig. 17, the Stiefel-constrained variant exhibits faster and more stable convergence of the reconstruction/dynamics loss and reaches a substantially lower final loss (on the order of 5e-4), whereas the unconstrained projector shows noisier training and a higher asymptotic error (on the order of 1e-2). In practice, the Stiefel constraint thus acts as a useful regularizer: it keeps the low-dimensional propagator $A_{\mathrm{eff}}$ well-conditioned, makes the ROM modes easier to interpret, and does not introduce any observable degradation in long-horizon performance compared to the unconstrained case.

*Table 10.* Wave ROM ($d = 32$): final epoch-averaged training losses for linear projection heads with vs. without Stiefel constraint.

| Projector | Recon. loss | (Linear) Dyn. loss | Total loss |
|-----------|-------------|--------------------|------------|
| Linear (unconstrained) | 1.038e-2 | 1.757e-2 | 1.125e-2 |
| Stiefel-constrained | **5.119e-4** | **5.996e-4** | **5.419e-4** |

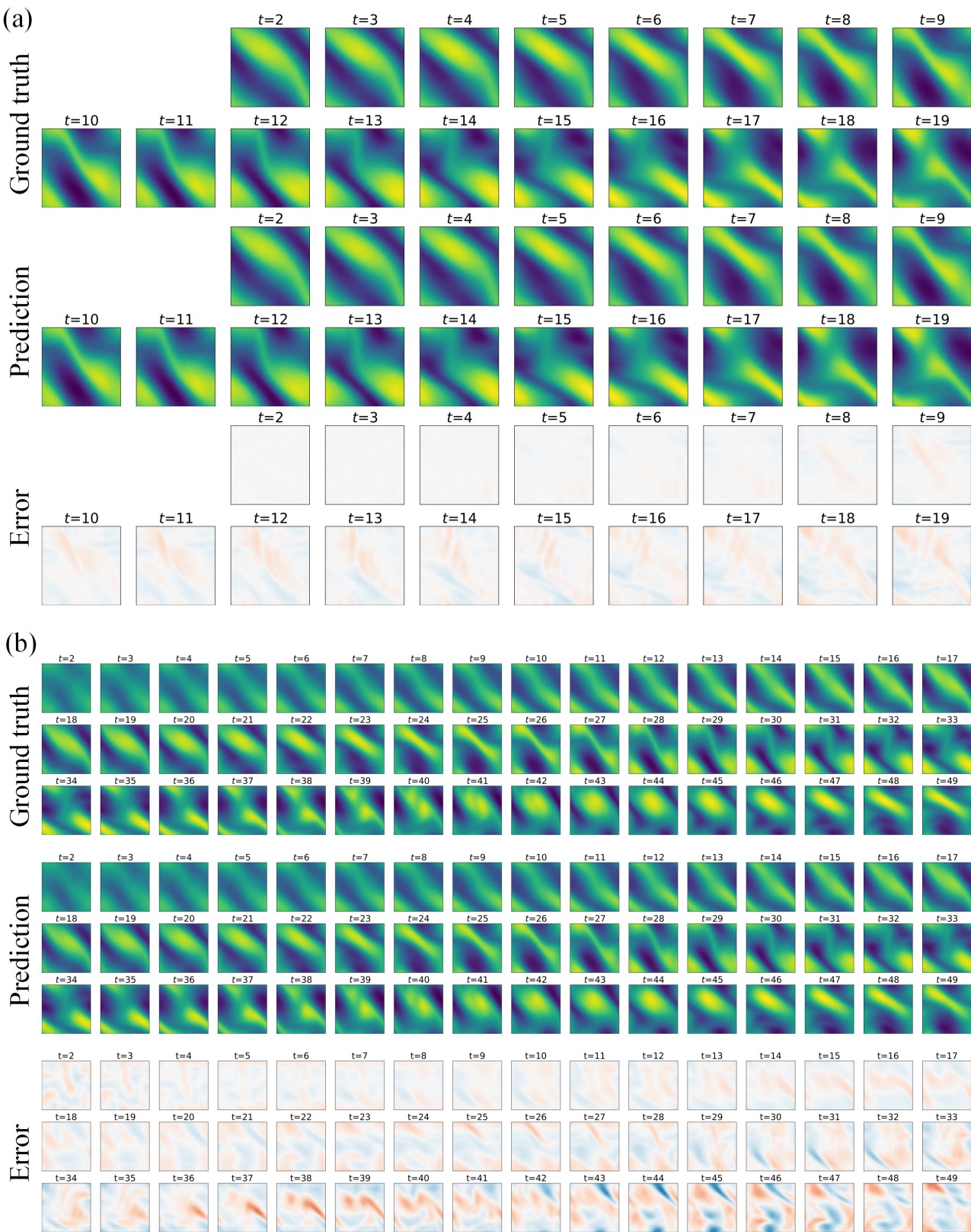

*Figure 8.* Temporal rollout on the Navier–Stokes long-term prediction task using a memory-augmented model. (a) Ground truth, model prediction, and error map along one trajectory. Frames $t = 0, 1, 2$ are used as conditioning inputs to the encoder; frames $t = 3, ..., 19$ are extrapolated predictions. (b) More challenging long-term prediction results: a 50-step rollout conditioned on $t = 0, 1, 2$.

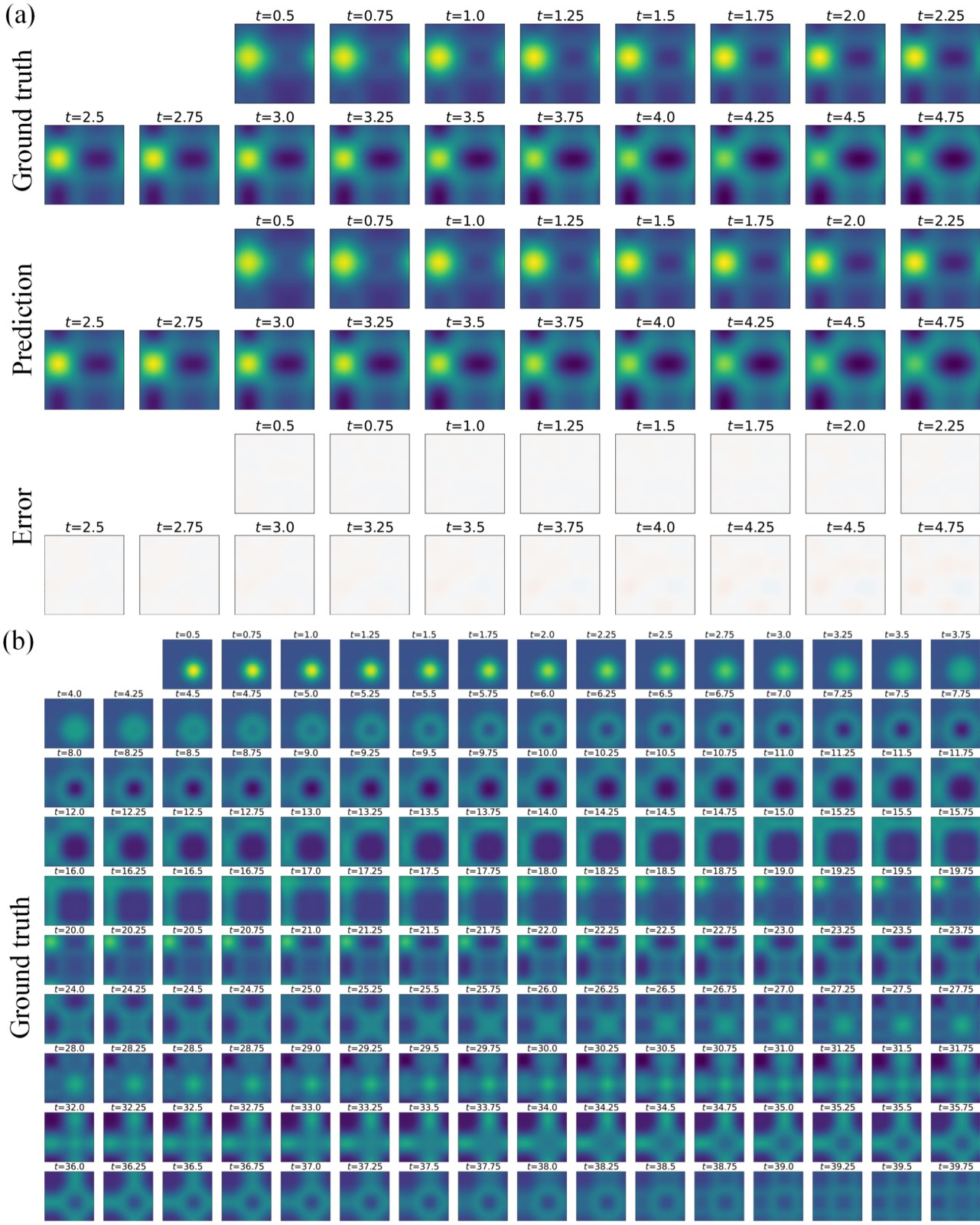

*Figure 9.* Temporal rollout on the Wave long-term prediction task using a memory-augmented model. (a) Ground truth, model prediction, and error map along one trajectory. Frames $t = 0, 0.25, 0.5$ are used as conditioning inputs to the encoder; frames $t = 0.75, ..., 4.75$ are extrapolated predictions. (b) More challenging long-term prediction results: a 160-step rollout conditioned on $t = 0, 1, 2$.

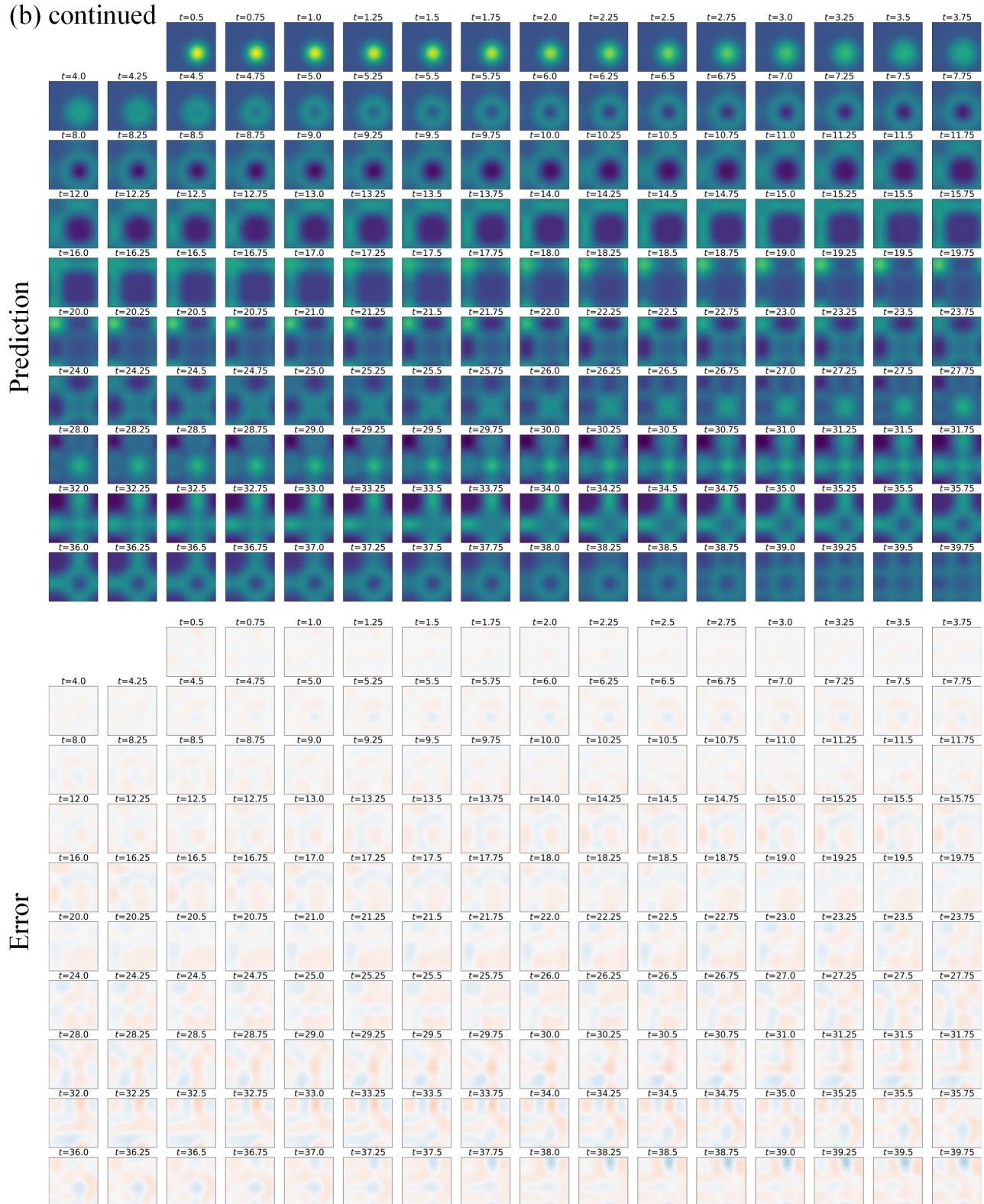

*Figure 10.* Figure 9(b) continued.

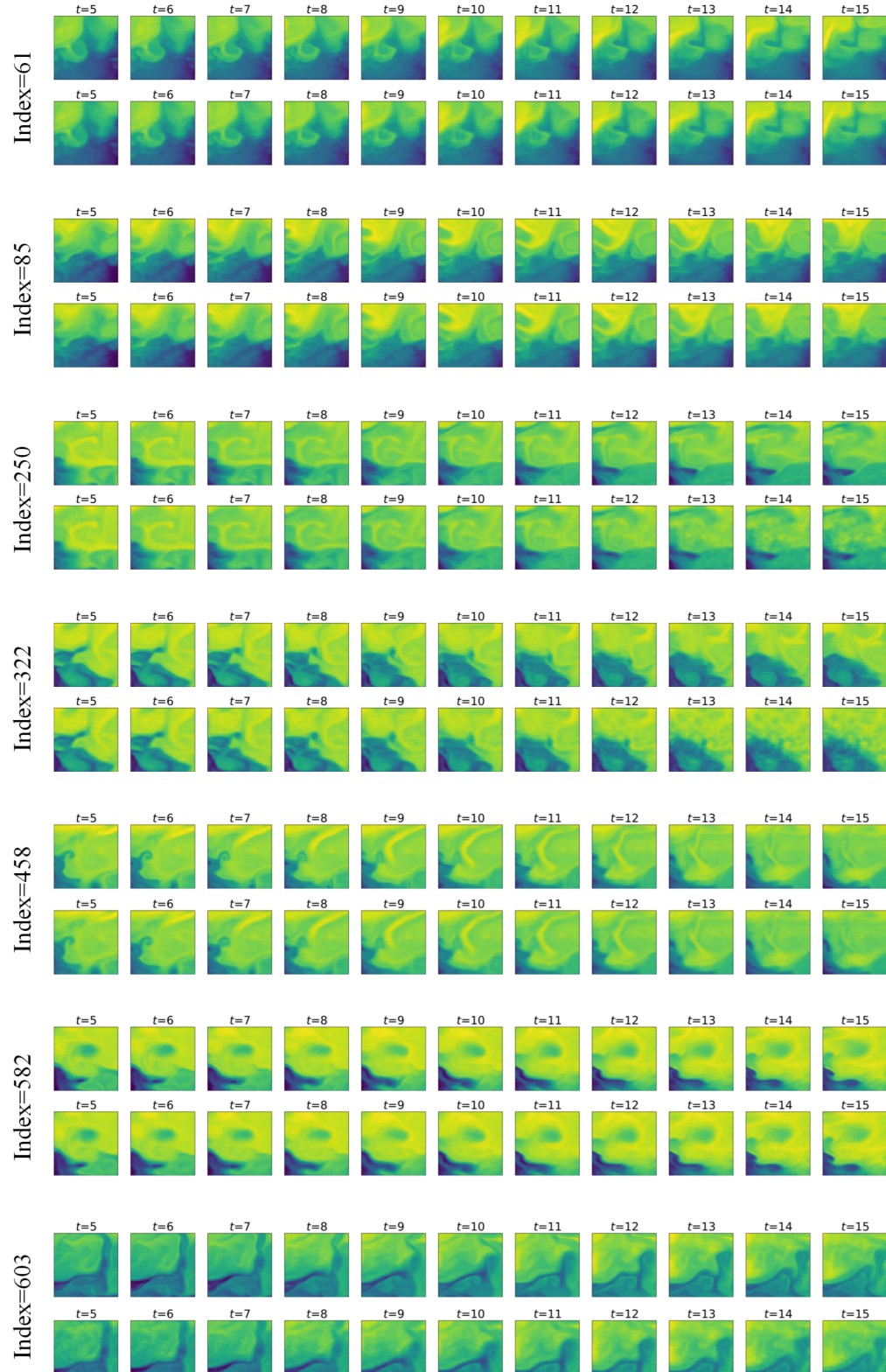

*Figure 11.* Temporal rollout on the SST prediction task using a memory-augmented model. Seven trajectories from the dataset are used for example. For each trajectory, the top row shows the ground truth and the bottom row shows the model rollout. The interval between adjacent frames is one day. Frames $t = 0, \ldots, 5$ are used as conditioning inputs to the encoder; frames $t = 6, \ldots, 15$ are extrapolated predictions.

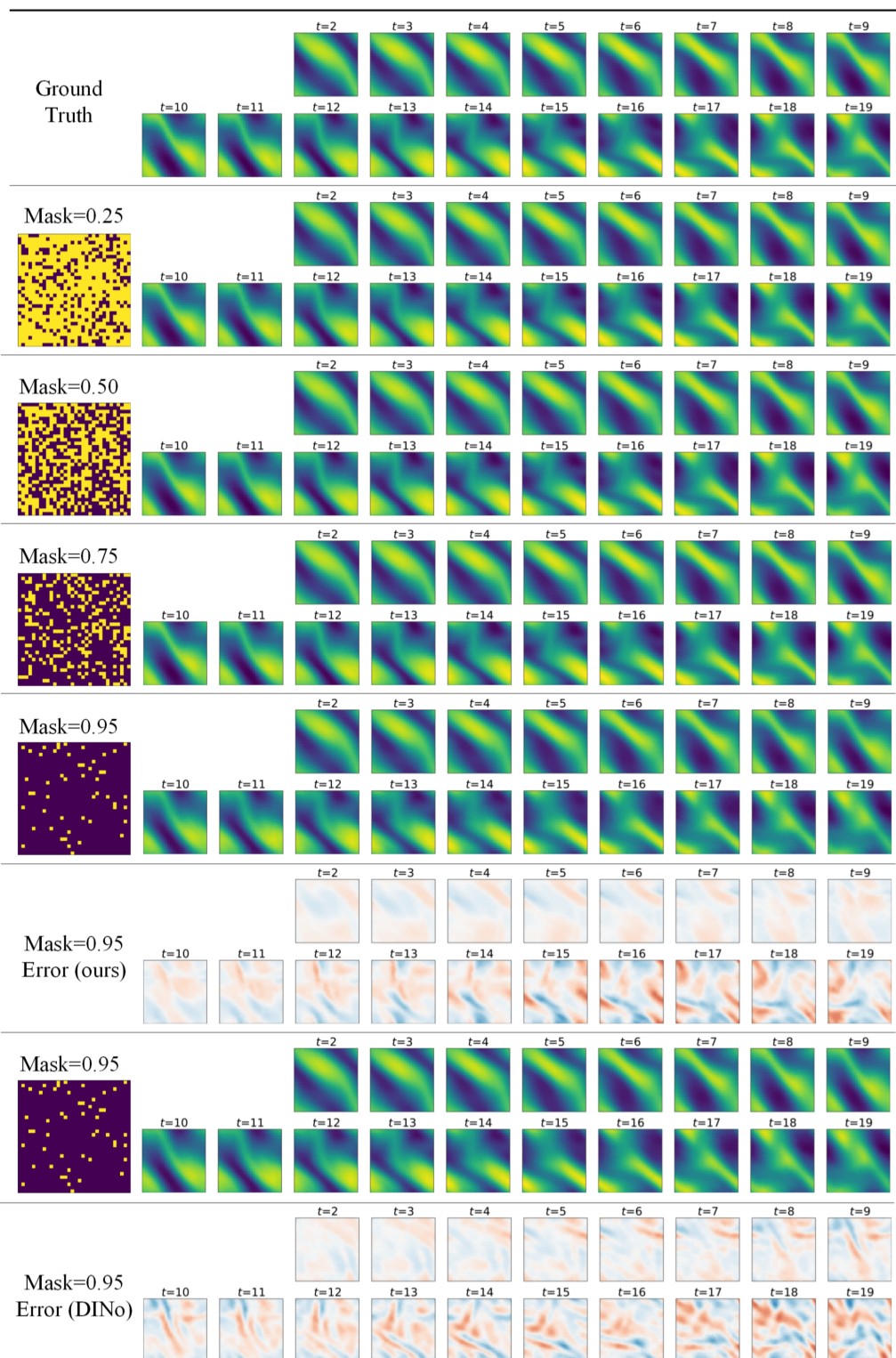

*Figure 12.* Temporal rollout on the masked Navier–Stokes long-term prediction task using a memory-augmented model. From top to bottom: ground truth; our model's predictions under different mask ratios; our model's error map at $95\%$ mask ratio; DINo's predictions at $95\%$ mask ratio; DINo's error map at $95\%$ mask ratio. For our model, frames $t = 0, \ldots, 2$ are used as conditioning inputs to the encoder; frames $t = 3, \ldots, 19$ are extrapolated predictions.

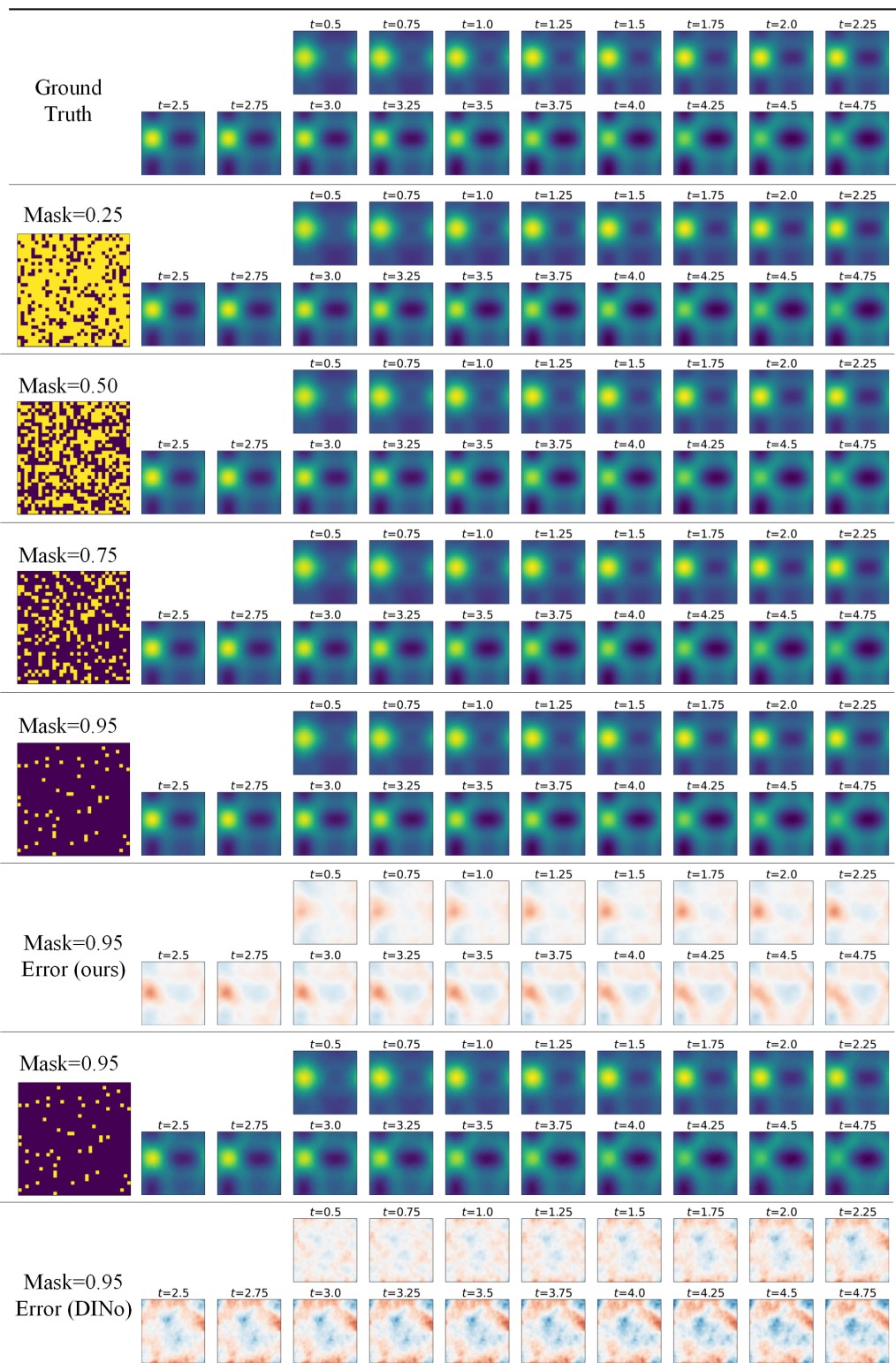

*Figure 13.* Temporal rollout on the masked Wave long-term prediction task using a memory-augmented model. From top to bottom: ground truth; our model's predictions under different mask ratios; our model's error map at 95% mask ratio; DINo's predictions at 95% mask ratio; DINo's error map at 95% mask ratio. For our model, frames $t = 0, \ldots, 0.5$ are used as conditioning inputs to the encoder; frames $t = 0.75, \ldots, 4.75$ are extrapolated predictions.

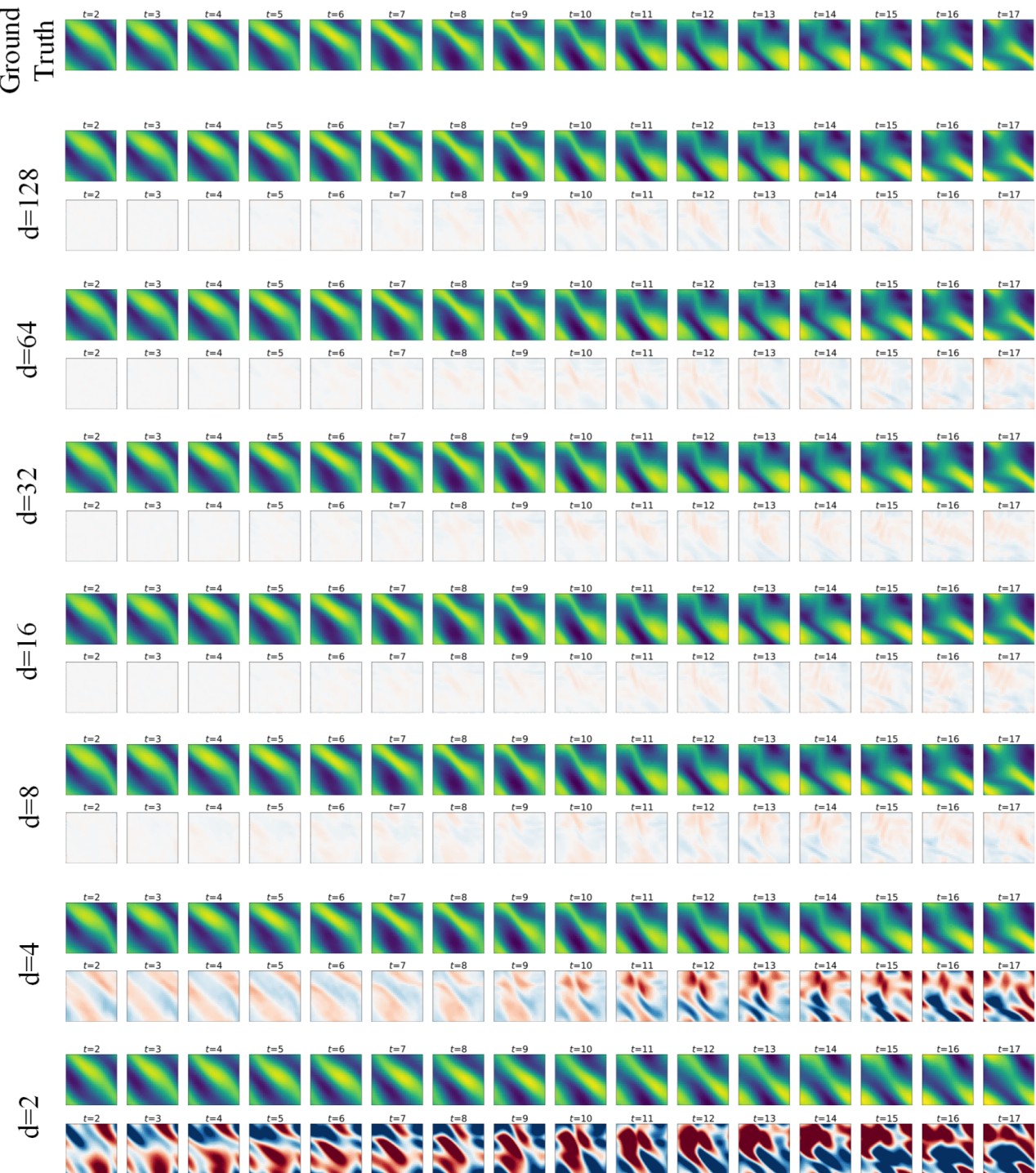

*Figure 14.* Reduced-order modeling (ROM) results. We present the predictions of a memory-augmented model for the Navier–Stokes equations along a sample trajectory. From top to bottom: ground truth; results for different latent dimensions (field and error map).

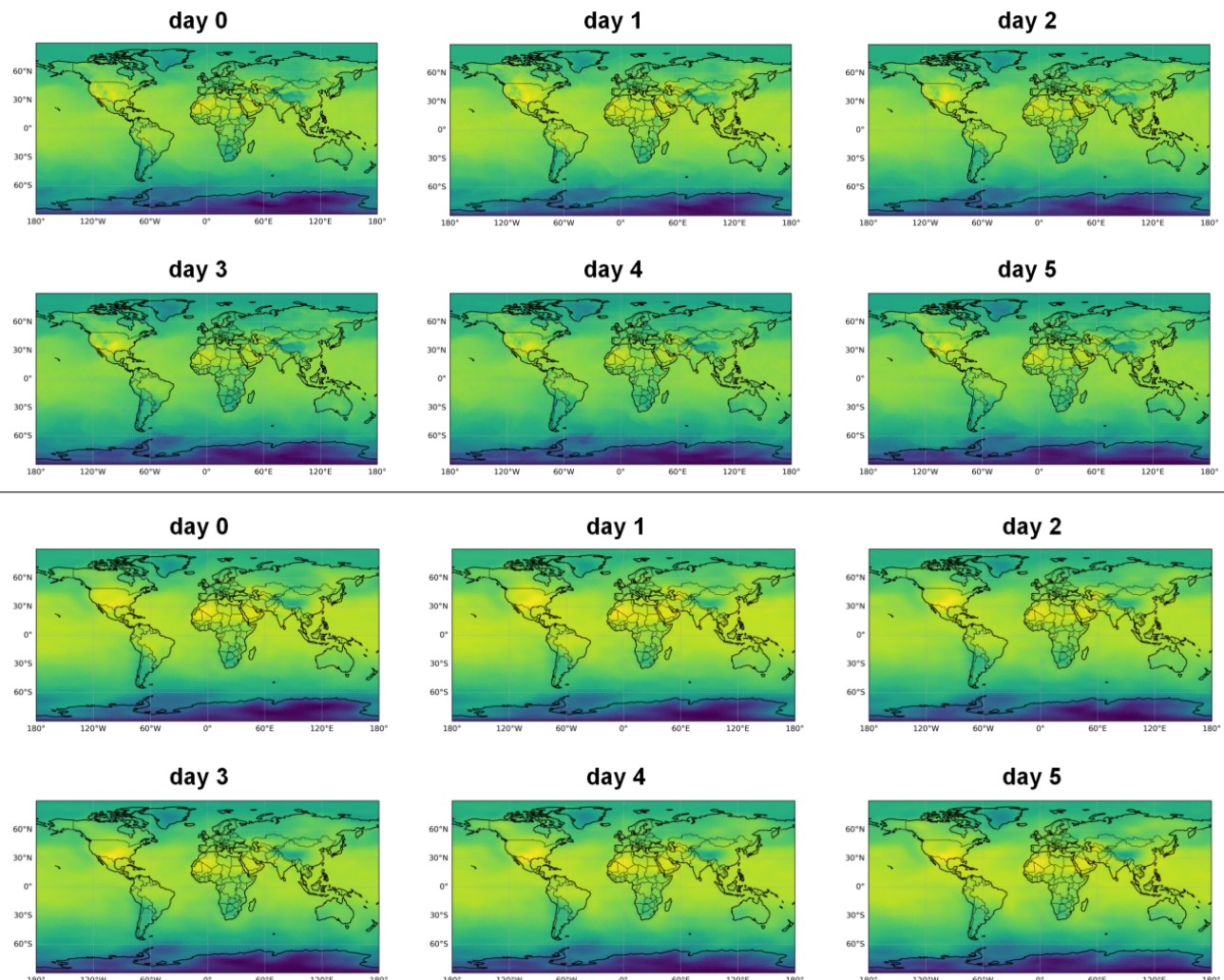

*Figure 15.* ERA5 temperature forecasting. Top: ground-truth temperature fields along a test trajectory. Bottom: MERLIN predictions on the same trajectory.

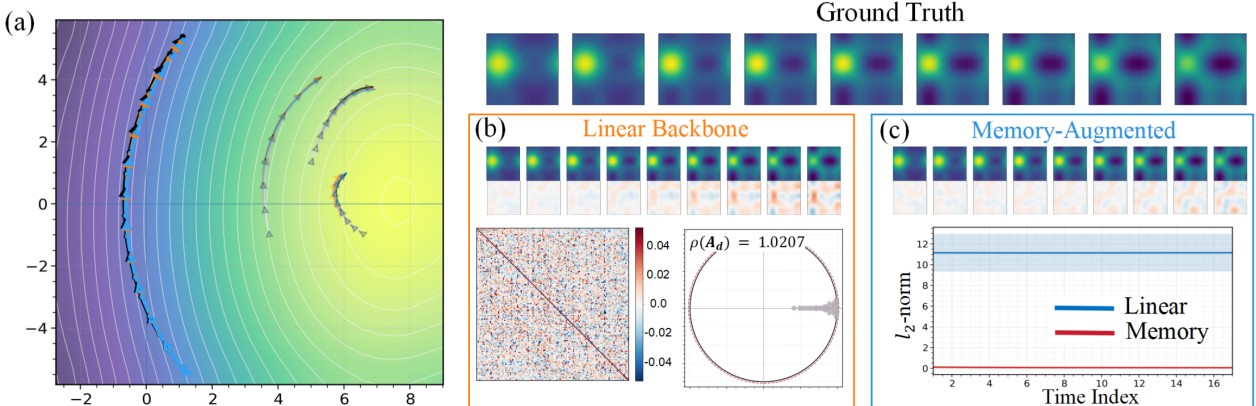

*Figure 16.* **Linear backbone vs. memory-augmented model (Wave).** (a) Latent phase-space trajectories in a 2D PCA subspace. "———": encoded from ground-truth states; "———": generated by the linear model; "———": generated by the memory-augmented model. (b) Top: predictions on Wave up to 5 s (every 0.5 s) using the linear model and the corresponding error maps; bottom: heatmap and eigenvalue spectrum of the linear propagator. (c) Top: predictions of the memory-augmented model and error maps; bottom: energy contributions of the linear vs. memory components.

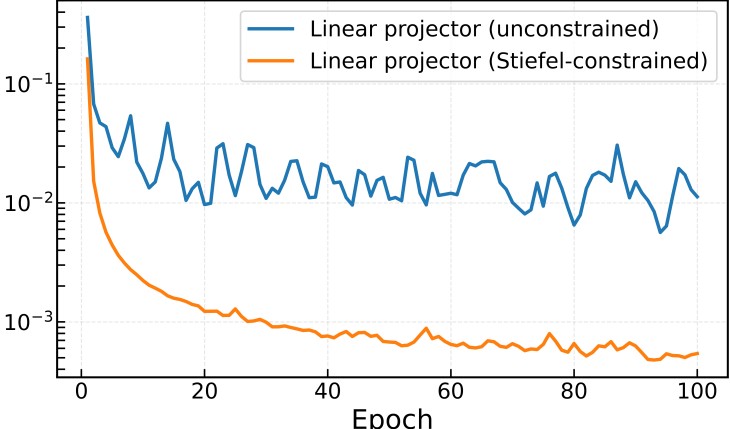

*Figure 17.* Training-loss comparison between unconstrained and Stiefel-constrained projection heads on the wave ROM experiment ($d = 32$).

### I.2. Hyperparameter Sensitivity of MERLIN

**Sensitivity to the number of `[CLS]` tokens $K$ and latent dimension $D$.** To assess model sensitivity with respect to this key hyperparameter $K$, we run an ablation on the Navier–Stokes dataset varying the number of `[CLS]` tokens as $K \in \{1, 2, 4, 6, 8, 10, 12\}$ with the latent dimension scaled as $D = 16K$ (which means each token is 16-dimensional). The detailed results are reported below in Table 11.

*Table 11.* Sensitivity to the number of `[CLS]` tokens $K$ and latent dimension $D$ on the Navier–Stokes dataset ($D = 16K$).

| Setting | Model | MSE | |
| --- | --- | --- | --- |
| | | Training horizon | Test horizon |
| Navier–Stokes, $K = 1, D = 16$ | Linear + FMM | 5.760e-4 | 1.218e-3 |
| | Linear + LMM | 8.702e-4 | 3.576e-3 |
| Navier–Stokes, $K = 2, D = 32$ | Linear + FMM | 3.312e-4 | 9.769e-4 |
| | Linear + LMM | 4.121e-4 | 1.678e-3 |
| Navier–Stokes, $K = 4, D = 64$ | Linear + FMM | 3.348e-4 | 1.436e-3 |
| | Linear + LMM | 4.080e-4 | 2.408e-3 |
| Navier–Stokes, $K = 6, D = 96$ | Linear + FMM | 3.841e-4 | 1.459e-3 |
| | Linear + LMM | 4.161e-4 | 2.097e-3 |
| Navier–Stokes, $K = 8, D = 128$ | Linear + FMM | 3.276e-4 | 1.638e-3 |
| | Linear + LMM | 2.924e-4 | 1.971e-3 |
| Navier–Stokes, $K = 10, D = 160$ | Linear + FMM | 3.686e-4 | 1.655e-3 |
| | Linear + LMM | 3.534e-4 | 1.954e-3 |
| Navier–Stokes, $K = 12, D = 192$ | Linear + FMM | 5.062e-4 | 2.323e-3 |
| | Linear + LMM | 4.796e-4 | 2.851e-3 |

Overall, we observe that with memory correction (both FMM and LMM), the test-horizon MSE remains in a narrow band across all $K$; e.g., for FMM the test MSE varies on the order of 1e-3, and even the smallest configuration $K = 1, D = 16$ already attains a very low test error (approximately 1.218e-3). This supports our claim that the dynamics are effectively low-dimensional.

Note that in this ablation we changed only $K$ (and $D$) while keeping the memory modules fixed. Thus, for larger $K$ the memory model is relatively underparameterized compared to the higher-dimensional latent space, which explains why performance does not monotonically improve and can even slightly degrade for very large $K$—this is consistent with expectations rather than instability.

**Sensitivity to the delay-embedding dimension.** Regarding the delay embedding, in our synthetic experiments we use a training horizon length $K = 10$ and a delay length $l = 3$. Since the effective number of usable time steps for training the latent dynamics is $K - l + 1$, we choose $l = 3$ as a conservative setting that preserves enough in-horizon steps while still enriching the observables with short-delay information, which is sufficient for these benchmarks.

To more directly assess sensitivity to the delay length, we additionally run a controlled study on the Navier–Stokes dataset, varying $l$ from 2 to 8. To keep the effective training horizon $K - l + 1$ reasonably large for all $l$, here we set $K = 15$. The results are shown in Table 12.

We observe that MERLIN is quite robust to the choice of $l$: both variants maintain stable long-horizon performance across a wide range of delay lengths, and performance tends to improve when $l$ is increased moderately (up to around 6–7 in this experiment). This supports the view that MERLIN does not require finely tuned delay-embedding hyperparameters to work well; the delay embedding mainly enriches the functional observables, while the memory module can flexibly adjust to correct the dynamics.

**Sensitivity to memory parameterization.** We further examine how the two memory parameterizations depend on their main memory-capacity hyperparameters on the Navier–Stokes dataset. For FMM, we vary the history length $l_{mem}$; for LMM, we vary the hidden memory dimension $d_{mem}$. The results are reported in Table 13.

The results show two complementary trends. First, FMM is relatively insensitive to the precise history length on this benchmark: varying $l_{mem}$ from 1 to 5 leads to only small changes in both training- and test-horizon errors. Second, LMM is

*Table 12.* Sensitivity to the delay length $l$ on the Navier–Stokes dataset ($K = 15$).

| Setting | Model | MSE | |
| --- | --- | --- | --- |
| | | Training horizon | Test horizon |
| Navier–Stokes, $l = 2$ | Linear + FMM | 1.069e-3 | 4.802e-3 |
| | Linear + LMM | 1.156e-3 | 6.691e-3 |
| Navier–Stokes, $l = 3$ | Linear + FMM | 8.820e-4 | 4.582e-3 |
| | Linear + LMM | 9.522e-4 | 4.690e-3 |
| Navier–Stokes, $l = 4$ | Linear + FMM | 7.182e-4 | 4.435e-3 |
| | Linear + LMM | 4.411e-4 | 3.003e-3 |
| Navier–Stokes, $l = 5$ | Linear + FMM | 3.062e-4 | 1.768e-3 |
| | Linear + LMM | 3.062e-4 | 2.590e-3 |
| Navier–Stokes, $l = 6$ | Linear + FMM | 4.368e-4 | 1.346e-3 |
| | Linear + LMM | 3.610e-4 | 2.025e-3 |
| Navier–Stokes, $l = 7$ | Linear + FMM | 1.441e-4 | 8.256e-4 |
| | Linear + LMM | 1.664e-4 | 1.840e-3 |
| Navier–Stokes, $l = 8$ | Linear + FMM | 2.220e-4 | 1.436e-3 |
| | Linear + LMM | 2.220e-4 | 2.611e-3 |

*Table 13.* Sensitivity to memory parameterization on the Navier–Stokes dataset. We vary the memory length $l_{\mathrm{mem}}$ for the finite memory model (FMM) and the hidden memory dimension $d_{\mathrm{mem}}$ for the leaky memory model (LMM).

| Model | Training horizon | Test horizon |
| --- | --- | --- |
| Linear backbone | 2.430e-3 | 1.372e-2 |
| FMM ($l_{\mathrm{mem}} = 1$) | 3.027e-4 | 1.627e-3 |
| FMM ($l_{\mathrm{mem}} = 2$) | 3.204e-4 | 1.592e-3 |
| FMM ($l_{\mathrm{mem}} = 3$) | 3.133e-4 | 1.617e-3 |
| FMM ($l_{\mathrm{mem}} = 4$) | 3.098e-4 | 1.616e-3 |
| FMM ($l_{\mathrm{mem}} = 5$) | 3.385e-4 | 1.713e-3 |
| LMM ($d_{\mathrm{mem}} = 8$) | 7.344e-4 | 7.708e-3 |
| LMM ($d_{\mathrm{mem}} = 16$) | 4.862e-4 | 3.794e-3 |
| LMM ($d_{\mathrm{mem}} = 32$) | 3.506e-4 | 2.460e-3 |
| LMM ($d_{\mathrm{mem}} = 64$) | 3.127e-4 | 2.007e-3 |
| LMM ($d_{\mathrm{mem}} = 128$) | **2.895**e-4 | 1.706e-3 |

more sensitive to the hidden memory dimension. When $d_{\mathrm{mem}}$ is too small, the memory state is under-parameterized and the test-horizon error increases; as $d_{\mathrm{mem}}$ grows, LMM approaches the performance of FMM. This is consistent with the interpretation of LMM as a low-dimensional exponential-memory realization: it can approximate the effective memory kernel well when sufficiently many memory modes are available, whereas FMM provides a more direct finite-history representation.

