# OpenReview forum: "Interpretable Functional Koopman Learning with Non-Markovian Closure for Spatiotemporal Systems"
_ICML.cc/2026/Conference — ICML 2026 spotlight_

### Official Review · Reviewer_Q3B7 · 2026-03-10

**Soundness:** 3
**Presentation:** 2
**Significance:** 3
**Originality:** 2
**Overall Recommendation:** 5
**Confidence:** 4

**Summary:**

This work introduces a methodology for inferring finite-dimensional representations of the Koopman operator that capture the dynamics of partial differential equations (PDEs). The method relies on an autoencoder framework that uses Galerkin Transformers as the encoder and FourierNets as the decoder, and leverages the Mori–Zwanzig projection formalism from statistical physics to motivate a non-Markovian correction. Both encoder and decoder are discretization-invariant, allowing the model to map partially observed functions to a finite-dimensional representation and back to function space, while the non-Markovian correction accounts for the truncation of the Koopman operator to finite dimensions.

The model is trained in two stages: first by fitting the autoencoder and the matrix representation of the Koopman operator, and subsequently by learning the memory correction. The authors demonstrate the capabilities of their approach on both synthetic and real-world datasets and compare it against state-of-the-art neural operator methods.

**Compliance With Llm Reviewing Policy:**

Affirmed.

**Final Justification:**

I thank the authors for their thorough response. I am largely satisfied with the rebuttal and have therefore increased my score to 5 (Accept).

**Key Questions For Authors:**

1. After fitting a given Koopman operator and the corresponding memory correction, why do you perform an additional compression step? Put differently, why are two projection-and-correction steps needed to determine the optimal latent dimension? What would happen if you directly set $d=8$ in the Navier–Stokes example, instead of using the current mapping $d=128 \rightarrow d = 8$

2. Following up on Question 1, how should one interpret the two sets of variables (e.g., the 128-dimensional versus the 8-dimensional representations)?

3. How sensitive is the method to the length of the delay window, and how does this choice affect the memory correction? Can you provide an example?

4. Why is the linear one-step model optimized via ridge regression? What happens if the linear operator is trained jointly with the autoencoder via gradient descent? Appendix D.4 discusses the behavior when the autoencoder, linear propagator, and memory correction are trained jointly, but what about training the autoencoder and the linear propagator together in Phase I?

5. In Appendix D.1 you define $Z_0$ and $Z_+$ as functions of the index $k$.  Do different batches correspond to different transition times? For example, does batch $i$  match $z_i$ at time $i\Delta t$ with $z_{i+1}$ at time $(i+1)\Delta t$?

**Overall, I believe the paper is strong. I would be happy to increase my score if the authors address the questions and weaknesses outlined above.**

**Limitations:**

There is no dedicated section summarising the limitations of the proposed approach (or at least I could not find one). Discussions of limitations appear scattered throughout the lengthy appendix. I suggest that the authors include a short limitations section at the end of the paper that consolidates the most relevant points, such as the neglect of the noise term that arises in truncated systems according to the Mori–Zwanzig formalism, as well as the complexity of the training pipeline and its many components.

**Strengths And Weaknesses:**

**Strengths**

- *Soundness*. The paper is technically sound. The experiments and ablation studies are well designed and clearly demonstrate the role and relevance of each component of the model. The explanations of most experimental findings are also convincing. Overall, the authors provide sufficient empirical evidence to support their claims.

- *Presentation*. The paper is easy to follow and, in my view, the extensive appendix provides sufficient detail to reproduce all results.

- *Significance*. The paper addresses an important problem. Inferring Koopman operators is well known to be challenging, and robust approaches are of broad interest to the machine learning community. In particular, using Koopman operators to model, forecast, and analyze spatio-temporal systems is of considerable theoretical and practical relevance.

- *Originality*. The authors build on a well-established theoretical framework and demonstrate how it can be used to learn finite-dimensional Koopman representations that enable stable long-term forecasting of spatio-temporal data while opening the door to theoretical analysis.

**Weaknesses**

- *Presentation*. Although the paper is generally easy to follow and the appendix is very extensive, many details that appear relevant are only discussed deep in the appendix and are not referenced in the main text. Two examples illustrate this issue:

1. The assumption that no noise is induced by the orthogonal dynamics on the unresolved subspace only appears in Appendix B.3. The authors also state in lines 914–915 that “when [the assumption that the memory kernel is sufficiently decaying and any remaining fluctuation acts as a small residual] fails, the method would require a larger latent dimension or a richer memory model, and performance may degrade.” However, they do not provide an explicit example of such a scenario, nor do they offer a clear cautionary remark in the main text.

2. Several technical components, including ridge regression with a frozen pretrained autoencoder, streaming exponential moving averages (EMA) of global sufficient statistics, latent-variable whitening, and memory regularization, are introduced only in the appendix without prior reference. Even if these details are primarily intended to improve training stability, they should at least be briefly mentioned in the main text. A concise list of these steps and their motivations would suffice, in my opinion.

- In addition, some claims appear somewhat overstated. In lines 70–72 the authors write that “when this linear invariance fails, we theoretically prove that a memory correction term should be incorporated, yielding an exact description of those observables via non-Markovian dynamics.” However, the results they present are standard in both the physics and machine learning literature. Indeed, the authors themselves later note in lines 792–793 that “what is new here is its reinterpretation in terms of functional observables for infinite-dimensional PDE dynamics.”

Finally, some information is missing. I refer to the questions below.

---

> ### Author Rebuttal · Authors · 2026-03-31
>
> We thank the reviewer for the thoughtful feedback. We are glad that the reviewer finds the paper technically sound and empirically convincing. Below we address the questions and refer to [an anonymous link](https://anonymous.4open.science/r/icml-C1EB) for additional results.
>
> **Clarifications and planned revisions.** We agree that several points should be stated more clearly.
> * Appendix D.1. We clarify that we sample $B$ trajectory windows, encode the $b$-th window into a latent sequence $\\{z_b^k\\}\_{k=1}^K$, and form $Z_0,Z_+$ by collecting all adjacent pairs $(z_b^k, z_b^{k+1})$ across both window and time indices $b,k$.
> * Limitations and technical components. We will add a brief Limitations section in the main text to summarize the main assumptions and trade-offs, including neglecting the noise term, assuming unresolved effects can be captured by a moderate latent space plus memory correction, and the added complexity of two-phase training. We will also briefly summarize the main technical components introduced in the appendix as design choices primarily aimed at improving stability and efficiency.
> * Theoretical wording. We will soften the claim around lines 70-72 and clarify that non-Markovian closure itself is classical MZ; our contribution is its reinterpretation for learned observables in PDE and its use in motivating the practical linear-plus-memory architecture.
>
> **On delay embedding.** We already report an explicit sensitivity study in Appendix Table 13, where we vary the delay $l$ and compare both memory models. The results show robustness across a range of delay lengths, while a moderate increase of $l$ generally improves long-horizon accuracy, with the best performance in this experiment around $l\approx 6-7$. Conceptually, a larger delay enriches parameterization of the observable subspace; more importantly, delay embedding enlarges the resolved observable space so part of the recent history is absorbed into the latent construction. From the MZ viewpoint, this will in turn reduce the residual memory that must be modeled explicitly. A simple example is $u_{tt}=c^2\Delta u$. Let $\phi_j$ be an eigenfunction of $-\Delta$ with eigenvalue $\lambda_j$, and define the observable $q_k:=\langle u(k\Delta t,\cdot),\phi_j \rangle$ whose evolution satisfies $q_{k+1}=2\cos(\omega_j\Delta t)q_k-q_{k-1}$ with $\omega_j=c\sqrt{\lambda_j}$. Thus, without delay embedding, the projected dynamics are not closed in the present observable $q_k$ alone. By contrast, once the encoder receives $(u_k,u_{k-1})$, it becomes much easier for the learned latent $z_k$ to encode this $q_{k-1}$-type history information, making the learned latent dynamics closer to Markovian and reducing the residual burden on the explicit memory correction.
>
> **On ROM.** Our primary model first learns a sufficiently expressive $D$-dimensional latent observable space with interpretable linear-plus-memory dynamics, serving as the **representation-learning** stage. The subsequent ROM stage performs **model reduction** within this learned space. Because the projection induces a new coarse-graining, an additional memory model is generally required for closure. This design allows us to identify the intrinsic ROM dimension efficiently, without retraining the full model for each candidate $d$. To verify that the observed transition is not an artifact of the projection head, we also trained direct low-dimensional models by fixing each token to 4d and varying the number of tokens, giving $D\in\\{4,8,12,16,32\\}$. The same trend appears: performance improves sharply from $D=4\to 8$, with only marginal gains beyond (Link Tab.6, Fig.2). Thus, the critical dimension is intrinsic to the predictive dynamics, while our projection-based route mainly offers a more efficient and cleaner way to identify it.
>
> Additionally, since ROM projection is trained jointly with reconstruction, linear-dynamics, and one-step prediction objectives, together with decoder fine-tuning, the reduced observables are chosen to remain both representationally useful and linearly predictable; they therefore need not coincide with a particular top eigenspace of the high-dimensional propagator. Empirically, the spectrum of the reduced propagator $U^\top A^\ast U$ remains near the unit circle (Link Fig.4), suggesting that the 8d ROM preserves the dominant slow dynamical backbone, while the remaining unresolved effects are handled by the memory term.
>
> **Ridge vs. joint training.** We use ridge regression since it gives a stable closed-form estimate of the latent linear propagator. We also tested joint Phase-I training of the autoencoder and linear operator (Link Fig.2, Tab.7). Although joint training attains similar training-horizon errors, it yields a less stable linear backbone; after memory correction, the ridge-based variant remains slightly better in extrapolation. This supports ridge regression as the more reliable choice for preserving an interpretable and robust latent linearization.

---

> > ### Author Rebuttal · Reviewer_Q3B7 · 2026-04-02
> >
> > I thank the authors for their thorough response. I am largely satisfied with the rebuttal and will accordingly increase my score.

---

> > > ### Author Response · Authors · 2026-04-03
> > >
> > > We sincerely thank the reviewer for the reconsideration of our rebuttal, and we are truly grateful that our clarifications were able to address the concerns and improve the reviewer’s assessment.

---

### Official Review · Reviewer_Meit · 2026-03-11

**Soundness:** 4
**Presentation:** 3
**Significance:** 3
**Originality:** 4
**Overall Recommendation:** 6
**Confidence:** 3

**Summary:**

This paper proposes a Koopman-inspired framework for spatiotemporal prediction of PDE-governed systems called MERLIN from partial, irregular, and resolution-varying observations. The main idea is to move from standard finite-dimensional Koopman learning to a functional Koopman setting for PDEs motivated by the Mori–Zwanzig formalism. Empirically, the paper evaluates MERLIN on 2D wave, Navier–Stokes, and SST forecasting tasks. The main results are that MERLIN appears to be especially strong in long-horizon stability and robusts to random/partial observations.

**Compliance With Llm Reviewing Policy:**

Affirmed.

**Final Justification:**

The authors have addressed my concern and I am satisfied with the results. The current score is my final evaluation.

**Key Questions For Authors:**

See the weakness.

**Limitations:**

See the weakness.

**Strengths And Weaknesses:**

Strength:
1. The paper is conceptually strong and fairly novel. The using of functional Koopman theory for PDE forecasting is interesting and original.
2. The experimental section is fairly broad since it has synthetic and real-world data, long-horizon rollout and ablations study.
3. The results on SST and the claims about controlled error growth in extrapolation look very compelling.

Weakness:
1. Although the theory is elegant, the bridge from the functional Koopman and generalized Langevin perspective to the actual learned neural model still feels somewhat not very closely connected.
2. Not many results on computational cost, training stability, and sensitivity to latent dimension settings or other parameters or factors.

---

> ### Author Rebuttal · Authors · 2026-03-31
>
> We thank the reviewer for the constructive feedback. Below we clarify the main concerns and refer to [an anonymous link](https://anonymous.4open.science/r/icml-C1EB) for additional quantitative results.
>
> **Connection between the functional Koopman/Mori-Zwanzig theory and the neural parameterization.** We explain this connection along three complementary aspects.
> * **Koopman-MZ-inspired decomposition and two-phase training.** Our latent dynamics are not modeled by a generic black-box non-Markovian sequence model; instead, the decomposition $z_{t+1}=Az_t+e_t$ is the discrete form suggested by the Mori-Zwanzig picture after projection onto a finite-dimensional observable subspace. Here, $Az_t$ is the Markovian linear backbone, while $e_t$ is the closure induced by unresolved observables when finite-dimensional invariance is lost. This also *directly* motivates our two-phase training: Phase I learns observables that approximately linearize the dynamics, and Phase II learns the residual non-Markovian closure rather than the full dynamics end-to-end.
> * **Neural memory parameterization.** Moreover, we discussed in Appendix F that our leaky memory model (LMM) was intended as a concrete GLE parameterization. At a high level (Appendix F), the memory in GLE can be parameterized from an operator-approximation viewpoint, by replacing the memory operator with a finite-dimensional realizable surrogate. An illustrative special case is to express the unresolved contribution as memory convolution $\int_0^t K(t-s)\psi(z_s)\mathrm{d}s$ and approximate the kernel by Prony-series, e.g., $K(\tau)\approx \sum_{j=1}^p C_j e^{-\beta_j \tau} B_j$. This yields a Markovian embedding by introducing auxiliary memory states $m_j(t):=\int_0^t e^{-\beta_j(t-s)} B_j\psi(z_s)\mathrm{d}s$, which satisfy $\dot m_j=-\beta_j m_j+B_j\psi(z)$, while the memory closure becomes $\sum_j C_j m_j$.  A discretization of such a finite-dimensional ansatz for the memory dynamics leads directly to the leaky-recursive form of LMM, $m_t=\gamma\odot m_{t-1}+\Phi_{\mathrm{enc}}(z_t)$, with $e_t=r\odot\Phi_{\mathrm{dec}}(m_t)$ as the corresponding closure term. By contrast, the finite-memory model (FMM) serves as a complementary, more agnostic parameterization when the effective kernel is not well captured by a small exponentially decaying surrogate. We apologize that this GLE interpretation of LMM, was not stated clearly enough in the main text, and will revise Section 4.2 and Appendix F accordingly.
> * **Empirical validation.** Finally, as suggested by Reviewer XM6E, we further strengthen this theory-model bridge with explicit quantitative diagnostics of non-linear-invariance and memory compensation. In particular, we report the invariance-violation measures before and after memory correction, the memory-to-backbone energy ratio (Link Tab.1), and the autocorrelation of the linear closure residual $r_t^{\mathrm{lin}}=z_{t+1}-Az_t$. These diagnostics make the intended picture directly testable: the linear backbone explains most of the latent evolution, while a relatively low-energy memory term compensates the remaining non-invariant component, consistent with our two-phase training design. On the synthetic benchmarks, the autocorrelation of $r_t^{\mathrm{lin}}$ exhibits an oscillatory-decaying pattern (Link Fig.1), indicating that the unresolved component is not well explained by unstructured noise, but instead has coherent temporal structure consistent with non-Markovian memory induced by unresolved dynamics. We refer to our response to Reviewer XM6E for the detailed definitions and quantitative evidence.
>
> **Computational cost, training stability, and sensitivity.** Our current draft already covers part of these aspects in the appendix, and, following suggestions from other reviewers, we also ran additional experiments to further strengthen the empirical study.
> * Computational cost: Appendix H.4 (Tab.5) provides a computational comparison showing that MERLIN uses fewer parameters and achieves faster inference, which we attribute to modeling dynamics in latent space.
> * Training stability: We compare two ROM training strategies in Appendix Fig.17, and, following Reviewer Q3B7’s suggestion, we additionally compare joint training of the linear propagator against our current closed-form ridge update (Link Fig. 2), report results across random seeds (Link Tab.4) and dataset sizes (Link Tab.3). These results consistently support the stability of the proposed training pipeline.
> * Hyperparameter sensitivity: We report sensitivity to latent dimension and delay embedding in Appendix J.2, compare the two memory modules in Appendix Tab.9, and further provide additional sensitivity results for the memory dimension $d_{mem}$ and memory length $l_{mem}$ (Link Tab.2). Overall, we observe consistent behavior across these variations.
>
> We agree that some of these results were not sufficiently emphasized in the main text, and we will highlight them more clearly in the revision.

---

> > ### Author Rebuttal · Reviewer_Meit · 2026-04-01
> >
> > Thanks for the detailed reply. Therefore, I will increase the score.

---

> > > ### Author Response · Authors · 2026-04-03
> > >
> > > We greatly appreciate the reviewer’s encouraging update and are thankful that our response helped clarify the main concerns and led to a more positive assessment of the paper.

---

### Official Review · Reviewer_Gr85 · 2026-03-12

**Soundness:** 3
**Presentation:** 2
**Significance:** 3
**Originality:** 3
**Overall Recommendation:** 4
**Confidence:** 3

**Summary:**

This paper studies spatiotemporal forecasting of PDEs and proposes an interpretable, equation free, and discretization invariant forecasting approach. The main contribution is a Koopman theory of infinite dimensional functionals with a Mori–Zwanzig compensation. This theory is empirically verified using an autoencoding architecture that learns a reduced order model with learned latent dynamics. The resulting model has the advantage of interpretability, and discretization invariance. Experiments on 2D Wave equation, Navier-Stokes, and Sea-surface temperature datasets show performance competitive with neural operators.

**Compliance With Llm Reviewing Policy:**

Affirmed.

**Final Justification:**

The authors addressed the concerns in my review with references to the appendix. One of my concerns (reorganization of the manuscript) was promised. I have changed my score accordingly.

**Key Questions For Authors:**

**Comments/Questions**.
- It would be helpful to expand the captions for Figures 2, and 3
- It might be clearer to write the loss function instead of describing it in the problem formulation
- Line 226: “As mentioned…” this hasn’t been mentioned before.
- Consider removing the bullets in Further discussion, as they don’t seem to improve readability
- I think it’s important to have related work in the main body, consider moving some of the Koopman theory to the appendix
- Many of the experiments are designed to show that MERLIN works for irregular discretizations; however, the memory correction is designed with an assumption of regular discretization. How do the authors reconcile this decision?
- Should the equation for m_{t-1} in (5) be the convex combination of the previous memory and current latent representation? I would expect this quantity to grow very quickly otherwise

**Limitations:**

The authors suggest there are societal impacts, but do not list any.

**Strengths And Weaknesses:**

**Soundness**: *The claims* of (1) being able to learn observation functionals from random partial observations; (2) achieving resolution free decoding; and (3) the method being interpretable due to Koopman structure with non-Markovian correction are supported. My concern here is that for the most part, the experiments are designed to show that MERLIN works for irregular discretizations; however, the memory correction is designed with an assumption of regular discretization. How do the authors reconcile this decision?
*The methods* appear appropriate.
*The proofs* were not reviewed carefully
*The empirical analysis* would benefit from ablation study. For example, It would be nice to see the performance of MERLIN without memory augmentation. Some conclusions are also non-intuitive. For example, the authors conclude that the sharp MSE drop that occurs when increasing the latent dimension indicates a low intrinsic dimension. I would expect the MSE to increase beyond the critical dimension. Some details of the experiments appear to be missing. For example, do the experiments in the main body use the leaky or finite memory augmentation strategy?

**Presentation**: Regarding *the writing*, the paper should be proof written. Possible typos include but are not limited to:
- “We model spatiotemporal dynamics governed by (1) from data.” → We model spatiotemporal dynamics from data.
- “The training set comprises trajectories {u(i)} obtained by simulating (1) from initial..” → “The training set is comprised of trajectories {u(i)} obtained by simulating spatial temporal dynamics from initial..”
- Line 141: “figure 1” → Figure 1
- Line 160: “enabling” → enable
- Line 226: “As mentioned…” this hasn’t been mentioned before.
*The organization* is fine, but there are occasionally references to material that hasn't yet been introduced. For example:
- Line 226: “As mentioned…” this hasn’t been mentioned before.
In terms of *the positioning*, the authors present related work, but push it to the appendices. I highly encourage including it in the main body. Space is tight, but I think some of the background on Koopman theory, and the reported training accuracy can be moved to the appendix to make more space.

**Significance**. The paper addresses an important problem (spatiotemporal forecasting of PDEs), and appears to do so in a way that would be impactful for not only accelerating PDE forecasting, but also discovering the intrinsic dimension of a given system.

**Originality**. As far as I can tell the work is original, and well justified.

---

> ### Author Rebuttal · Authors · 2026-03-31
>
> We thank the reviewer for the constructive feedback. We are encouraged that the reviewer recognized the novelty and practical value of learning from partial observations with a resolution-free decoder and an interpretable Koopman-plus-memory structure. We clarify the main concerns below and refer to [an anonymous link](https://anonymous.4open.science/r/icml-C1EB) for additional results.
>
> **Clarification on discretization.** Our paper involves two types of discretization: spatial and temporal. By irregular sampling, we mean **spatial** irregularity, namely learning Koopman observables from sparse, spatially irregular observations, as illustrated in the “Flexibility with random field observations” experiment and Appendix pp. 39–40. On the **temporal** side, we assume a fixed interval between frames, as is standard in spatiotemporal datasets. Thus, there is no inconsistency: observations are irregular in space, while the memory correction is implemented in with regular time interval. The leaky-memory update can also be viewed as discretization of the continuous-time model in Appendix F; extending to irregular time stamps via a neural-ODE is a natural future direction.
>
> **On the leaky-memory update in Eq. (5).** As explained in Appendix F, the leaky-memory model is motivated as a discretization of the continuous-time decaying memory model $\dot {m}(t)=\Lambda_\theta m(t)+\Phi_{enc}(g_{\mathcal{M}}(t))$, where $\Lambda_\theta$ is diagonal with negative entries. An Euler discretization gives $m_{k+1}\approx (I+\Delta t \Lambda_\theta)m_k+\Delta t\Phi_{enc}(z_{k+1})$, or equivalently, $m_{k+1}=\Gamma m_k+\widetilde{\Phi}\_{enc}(z_{k+1})$, where $\Gamma=\mathrm{diag}(\gamma_1,\dots,\gamma_{d_{mem}})$ with $0<\gamma_i<1$. This is the form used in Eq. (5), so the role of $\gamma$ is exponential decay of past memory rather than convex averaging with the latent state. As long as $\widetilde{\Phi}_{enc}(z_k)$ is bounded, $m_k$ is a geometrically weighted sum of past inputs and therefore does not grow unbounded.
>
> **On ablations and the memory realization.** We agree this should be stated more clearly in the main text. Appendix J of the current draft already includes ablations on ROM training, two memory parameterizations, and sensitivity to latent dimension and delay embedding; following other reviewers’ suggestions, we also add sensitivity results for memory dimension $d_{mem}$, memory length $l_{mem}$ (Link Tab.2), and dataset size (Link Tab.3). In particular, the appendix “Comparison with finite-dimensional Koopman autoencoders”(p. 44) compares DeepKAE, MERLIN linear backbone, and full memory-augmented model, showing that the linear backbone already outperforms the finite-dimensional baseline, while memory term brings further gains. This suggests that the improvement comes from the combination of the functional Koopman backbone and memory closure, rather than from a generic recurrent correction alone. *Regarding the memory realization* in the main text, Appendix Tab.9 (also see Link Tab.5) directly compares two memory models, and the main text reports the best-performing one. Overall, these results suggest that MERLIN is fairly insensitive to the specific memory parameterization across the datasets considered in the main text. We will make this choice explicit in the revision.
>
> **On the interpretation of the latent-dimension transition.** At the dynamical level, once the latent dimension exceeds a small threshold, the learned Koopman observables already capture the dominant modes, suggesting a low intrinsic dimension of the effective dynamics (see also our response to Reviewer Q3B7); the remaining unresolved effects are then handled by the memory term, consistent with our Mori-Zwanzig motivation. This is also supported by Appendix Tab.12: even the smallest setting $D=16$ already achieves low error, while gains for larger $D$ are marginal. As expected, the mild non-monotonicity at larger dimensions may reflect overparameterized training and the fact that only the latent dimension is enlarged while the memory module is kept fixed, but this is orthogonal to the critical transition emphasized in the main text, which concerns the emergence of the dominant predictive dynamics once a minimal latent dimension is reached.
>
> **On presentation, organization, and broader impact.** We appreciate these suggestions and will revise the writing, figure captions, and overall organization accordingly, including moving related work and other key context into the main text. We also agree that “As mentioned” at line 226 is imprecise: the intended reference was Sec. 3.3, where delay embedding is already introduced, and we will rewrite this for clarity. Regarding broader impact, we agree that the current limitation statement is too brief; we will make the practical value more explicit, for example in real sensing scenarios with irregular measurements and in applications where long-horizon forecasting under partial observations is important.

---

> > ### Author Rebuttal · Reviewer_Gr85 · 2026-04-02
> >
> > Thank you for clarifying. I am adjusting my score in anticipation of the reorganization noted. I hope the authors will also consider referencing the appropriate appendix when justification is not present in the main body.

---

> > > ### Author Response · Authors · 2026-04-03
> > >
> > > We sincerely thank the reviewer for the thoughtful follow-up and for adjusting the score. We will carefully revise the paper to address the presentation issues noted above, including improving the organization, discussing related work, limitations, broader impacts, and applications more clearly in the main text, and adding clearer references to the appendix wherever supporting justification is not included in the main body.

---

### Official Review · Reviewer_XM6E · 2026-03-13

**Soundness:** 3
**Presentation:** 3
**Significance:** 2
**Originality:** 3
**Overall Recommendation:** 4
**Confidence:** 4

**Summary:**

The paper proposes MERLIN, which is a framework for learning spatiotemporal dynamics from partial/irregular observations by evolving a koopman-style linear latent state and decoding to full fields at arbitrary query points. A key idea is adding a non-markovian 'memory' closure (inspired by Mori-Zwanzig) to correct drift when the learned observable subspace is not perfectly invariant. The model is designed to be discretization invariant in both encoding and decoding, enabling resolution-free reconstruction. Experiments on synthetic pdes and sea-surface temperature show that it improved long-horizon stability and useful reduced-order interpretability.

**Compliance With Llm Reviewing Policy:**

Affirmed.

**Final Justification:**

The authors addressed my queries during the rebuttal stage and I maintain by 'weak accept' (i.e 4) for the paper.

**Key Questions For Authors:**

1) How sensitive is MERLIN to the specific memory parametrization and training protocol (Phase1->Phase2)? Concretely, which design choices most affect long-horizon stability? Do you see consistent behavior across seeds and dataset sizes?

2) How did you ensure the baseline comparisons are strictly fair under irregular/partial observation settings? For e.g. are all the methods given the same conditioning frames, identical observation masks, and given the same comparable capacity/tuning budgets?

3) Can you provide a clearer diagnostic that shows when the learned observable subspace is "non-invariant", and how much memory term compensates? A quantitative measure of the invariance violation (before/after memory) would help validate M-Z motivation beyond qualitative latent-trajectory plots.

**Limitations:**

Yes.

**Strengths And Weaknesses:**

Soundness: There is a conceptual motivation (Koopman backbone + explicit memory correction) and reasonable empirical support, though the main accuracy table shows top operator baselines can still win on some synthetic benchmarks.

Presentation: Generally coherent, but the training procedure and "what changes between phases" could be made more explicit for reproducibility.

Significance: There is indeed practical value for real sensing scenarios with irregular measurements and long-horizon rollouts. To say whether there is a broader impact depends on the performance across more diverse PDE families and regimes, as examples are not enough right now.

Originality: The novelty is on the integrated combination of funcitonal koopman learning, resolution-free discretization invariance, and mori-zwanzig-inspired non-markovian closure with an interpretability angle.

---

> ### Author Rebuttal · Authors · 2026-03-31
>
> We thank the reviewer for recognizing the motivation and relevance under partial observations. We agree that the training protocol and empirical validation can be clarified further and will revise the main text accordingly. We also note that, beyond main benchmarks, we already stress-test MERLIN on KS and ERA5 (Appendix H.7-H.8), supporting its effectiveness across broader regimes. Below we address the questions and refer to [an anonymous link](https://anonymous.4open.science/r/icml-C1EB) for additional tables and visualizations.
>
> **Fair comparisons.** All methods use the same protocol: 3 conditioning frames, a 10-frame training horizon, and a further 10-frame rollout. Under partial observations, all methods share the same data split and masks under the same random seed, with masks generated once in a common pipeline from the same seed. We followed the public/recommended settings of each baseline in Appendix H.1 and kept the training budget comparable, rather than selectively tuning MERLIN.
>
> **Quantitative diagnostic of non-invariance and memory compensation.** In our latent dynamics, we define the linear closure residual as $r_t^{\mathrm{lin}}:=z_{t+1}-Az_t$, and the residual after memory correction as $r_t^{\mathrm{mem}}:=z_{t+1}-Az_t-e_t$, where $e_t$ is the learned correction. Under exact linear invariance, the linear backbone alone would close the dynamics, so $r_t^{\mathrm{lin}}$ would be negligible. This motivates the normalized invariance-violation measures $\mathrm{InvErr}\_{\mathrm{lin}}:=\sum_t\|r_t^{\mathrm{lin}}\|\_2^2/\sum_t\|z_{t+1}\|\_2^2$ and $\mathrm{InvErr}\_{\mathrm{mem}}:=\sum_t\|r_t^{\mathrm{mem}}\|\_2^2/\sum_t\|z_{t+1}\|\_2^2$. The relative reduction $\mathrm{Comp}:=1-\mathrm{InvErr}\_{\mathrm{mem}}/\mathrm{InvErr}\_{\mathrm{lin}}$ quantifies non-invariant component compensated by memory, while $\mathrm{MEB}:=\sum_t\|e_t\|\_2^2/\sum_t \|A z_t\|\_2^2$ measures whether the correction remains lightweight w.r.t the backbone. Across benchmarks, we observe substantial $\mathrm{Comp}$ with small $\mathrm{MEB}$ (Link Tab.1), supporting a dominant linear backbone plus a low-energy memory correction. To test whether the linear residual has temporal structure, we examine its autocorrelation $\rho(\tau):=\sum_t\langle r_t^{\mathrm{lin}},r_{t+\tau}^{\mathrm{lin}}\rangle/\sum_t\|r_t^{\mathrm{lin}}\|\_2^2$. If $r_t^{\mathrm{lin}}$ were mainly white noise, $\rho(\tau)$ would be close to zero for $\tau>0$. Instead, we observe clear short-lag correlation (Link Fig.1); on NS, the autocorrelation even exhibits an oscillatory decay. This is notable because, although $\mathrm{InvErr}_{\mathrm{lin}}$ is already small, the remaining residual still displays structured memory effects rather than unstructured noise. After adding memory correction, both $\mathrm{InvErr}\_{\mathrm{mem}}$ and autocorrelation decrease, showing that the memory module reduces both residual magnitude and temporal dependence. Together, these diagnostics directly support the MZ interpretation beyond latent-trajectory plots.
>
> **Sensitivity to memory parameterization and training protocol.** First, the 2-phase training is designed to learn a linear Markovian backbone plus a low-energy memory correction, consistent with the Koopman-MZ picture. Joint training conflicts with this decomposition and, in our experience, harms interpretability and long-horizon stability; see also Appendix D.4 and our response to Reviewer Meit.
>
> Regarding memory parameterization, leaky memory model (LMM) and finite memory model (FMM) perform similarly overall on NS and Wave (Appendix Tables 9,12-13). We further test hyperparameter sensitivity to $d_{mem}$ for LMM and $l_{mem}$ for FMM, using the train-to-test loss ratio as an indicator for long-horizon stability. In Link Tab.2, FMM is insensitive to $l_{mem}$, whereas LMM is more sensitive to $d_{mem}$: small $d_{mem}$ leads to underparameterization, lower $\mathrm{Comp}$, and faster error growth, while larger $d_{mem}$ brings LMM close to FMM. Importantly, the linear backbone alone already shows relatively small error growth, while memory mainly reduces absolute error and keeps this growth ratio in a similar range, suggesting that stability is primarily provided by the backbone while memory improves accuracy by compensating the structured non-Markovian residual. The distinction becomes more important in chaotic regimes: on KS, LMM fails while expressive FMM remains effective (Appendix H.7), consistent with the much stronger residual autocorrelation for KS in Link Fig.1, indicating more coherent and longer-range memory which is not well captured by the exponentially decaying ansatz of LMM. Overall, memory parameterization is fairly insensitive on simpler benchmarks, but becomes important in strongly chaotic regimes, where a richer model such as FMM is preferable.
>
> **Consistency across seeds and dataset sizes.** Yes—we observe stable performance across 3 random seeds and varying dataset sizes; see Link Tables 3–4.

---

> > ### Author Rebuttal · Reviewer_XM6E · 2026-04-04
> >
> > I really appreciate the authors response and my concerns are fully addressed. I will maintain my current score.

---

> > > ### Author Response · Authors · 2026-04-04
> > >
> > > Thank you very much for the positive update. We are truly glad to hear that our response has fully addressed your concerns. If you feel that these clarifications have strengthened the paper and further improved your overall assessment, we would be very grateful if this could be reflected in your final evaluation, as appropriate. In any case, we sincerely thank the reviewer for the careful review, constructive feedback, and positive recognition of our work. We are very grateful for the reviewer’s time and consideration, and we would be happy to provide any further clarification if useful.

---

### Decision · Program_Chairs · 2026-04-30

**Decision:**

Accept (spotlight)

**Comment:**

This submission presents a strong and original contribution at the intersection of Koopman learning, reduced-order modeling, and spatiotemporal forecasting. All reviewers found the core ideas technically sound and practically meaningful and 3 of them were further convinced by the rebuttal and increased their score. The experimental results support the main claims convincingly and I encourage the authors to add their clarifications from the rebuttal to the final version.